# Phase information is conserved in sparse, synchronous population-rate-codes via phase-to-rate recoding

Daniel Müller-Komorowska [1,2,5] ✉, Baris Kuru[2,5], Heinz Beck[2,3] &
Oliver Braganza [2,4] ✉

Neural computation is often traced in terms of either rate- or phase-codes. However, most circuit operations will simultaneously affect information across both coding schemes. It remains unclear how phase and rate coded information is transmitted, in the face of continuous modification at consecutive processing stages. Here, we study this question in the entorhinal cortex (EC)- dentate gyrus (DG)- CA3 system using three distinct computational models. We demonstrate that DG feedback inhibition leverages EC phase information to improve rate-coding, a computation we term phase-to-rate recoding. Our results suggest that it i) supports the conservation of phase information within sparse rate-codes and ii) enhances the efficiency of plasticity in downstream CA3 via increased synchrony. Given the ubiquity of both phase-coding and feedback circuits, our results raise the question whether phase-to-rate recoding is a recurring computational motif, which supports the generation of sparse, synchronous population-rate-codes in areas beyond the DG.

One of the most fascinating aspects of neuronal activity is its ability to encode sensory variables, such as place, intensity or context. Yet, understanding how sensory variables can be decoded from spikes of a neural populations and passed on for further processing to a different set of neurons with different properties continues to be an interesting challenge. This challenge is compounded when multiple coding strategies are at play across different interacting regions. Two broad coding strategies have been identified across multiple brain regions. In rate-coding schemes information is encoded in the firing rates of populations of neurons[1]. Temporal coding schemes, on the other hand, describe when information is encoded in the precise timing of spikes relative to a relevant reference signal[2–4]. A prominent temporal coding scheme is phase-coding, where the timing of spikes with respect to an ongoing field-potential oscillation represents information[5,6]. For instance, the precise timing of spikes within theta

oscillations is known to carry spatial information in both the hippocampus[7–9] and its input areas[10,11]. Specifically, cells within these regions display theta-phase precession, meaning that the spatial location of an animal is reliably related to the timing of spikes within the ongoing theta cycle. Interestingly, phase precession is being reported in a steadily increasing number of brain areas, representing spatial as well as non-spatial information[12–14], in both rodents and humans[15,16].

Importantly, phase- and rate-coding schemes are not mutually exclusive and likely often play complementary roles[13,17]. However, it is not well understood how phase- and rate-coded information is modified during successive stages of processing. This is non-trivial because both the firing rate and the temporal relation of action potentials are substantially modified by local circuit motifs. For instance, feedback inhibition is often argued to perform a so called winner-takes-all

[1]Neural Coding and Brain Computing Unit, Okinawa Institute of Science and Technology Graduate University, Okinawa 904-0495, Japan. [2]Institute for Experimental Epileptology and Cognition Research, University of Bonn, Bonn, Germany. [3]Deutsches Zentrum für Neurodegenerative Erkrankungen e.V, Bonn, Germany. [4]Institute for Socio-Economics, University of Duisburg-Essen, Duisburg, Germany. [5]These authors contributed equally: Daniel Müller-Komorowska, Baris Kuru. ✉e-mail: daniel.mueller-komorowska@oist.jp; oliver.braganza@ukbonn.de

computation on neuronal rate-codes, where only the cells with the highest input rates get to fire[18,19]. However, feedback inhibition is likely to also strongly affect information coded via theta phase precession, or other time codes[20]. More generally, it remains unclear how phase- and rate-coded information are transmitted, in the face of continuous modification at consecutive processing stages in the brain

Here, we use computational modeling of the entorhinal cortex (EC)−dentate gyrus (DG)−CA3 system to rigorously examine how the information contained in phase- and rate-codes is modified by an experimentally well-constrained feedback inhibitory circuit. We use this system because (i) phase- and rate-codes are known to carry spatial information in these sequentially connected subfields, and (ii) the DG is a prominent example of strong canonical inhibitory circuits with known physiological and computational properties. Specifically, feedback inhibition in DG supports so called "pattern separation," i.e., the decorrelation of activity patterns, which in turn is thought to improve successful associative storage in CA3 networks, thus enhancing so called "pattern completion"[21]. Notably, computational explorations of pattern separation and completion have overwhelmingly assumed rate-coding schemes (but see Madar et al.[22]), despite evidence for the relevance of spike-timing[22,23].

Our results reveal a coding principle emerging when (i) feedback inhibitory circuits and (ii) phase precessing inputs coincide, termed "phase-to-rate recoding." We show that feedback inhibition draws on EC phase information to create a sparse, yet synchronized, population-rate-code with increased information content. We establish this results using three complementary approaches, namely (i) spatial information analysis[24], (ii) perceptron analysis[25], and (iii) tempotron analysis[26]. Phase-to-rate recoding has two key advantages. First, it allows information within dense phase-codes to be conserved within sparse rate-codes. Second, by increasing DG synchrony, it increases the efficiency of spike-timing-dependent plasticity (STDP) at recurrent CA3 synapses. Together, this suggests that phase-to-rate recoding in the DG may support the canonical hippocampal functions of pattern separation and completion. Given the ubiquity of temporal coding and feedback circuits, these results also raise the question whether brain circuits more generally can implement variations of phase-to-rate recoding to create sparse, synchronous population-rate-codes with increased information content.

## Results

### DG pattern separation of phase- and rate-codes
To investigate how phase- and rate-coded information are transmitted across the EC-DG circuit, we first created a phenomenological model of phase-precessing grid-cell firing in EC. Briefly, we modeled a grid cell population of 200 cells, matching the empirical distribution of grid sizes, phases and orientations[27,28] (Fig. 1a, Supplementary Fig. 1a, b). We then adapted a phase precession model[29] allowing us to create naturalistic phase-precessing spike patterns for trajectories through virtual space. Specifically, we assumed a mouse moving through virtual space in a straight line at a constant velocity (20 cm/s; Fig. 1b), simulating grid cell activity based on a constant theta oscillation at 10 Hz (Fig. 1c, Supplementary Fig. 1c). This reproduced the distinctive phase precession patterns observed empirically, both for average data of EC grid cells[8,10] (Fig. 1d, e) and single trial data of identified EC stellate cells[30,31] (Supplementary Fig. 2).

To explore how phase and rate information from EC inputs was affected by local circuits in DG, we used a previously established model with well-constrained temporal properties (pydentate[32]). Briefly, pydentate reflects a biophysically realistic model, in which we had precisely calibrated the spatial and temporal properties of the complex feedback inhibitory microcircuit output to experimental data[32] (Fig. 1f, g, a local circuit model containing 2000 GC and 108 interneurons, see "Methods").

While there is substantial research on the ability of the DG circuit to perform pattern separation of rate-coded inputs, it has never been explored if it can also perform pattern separation of phase-coded inputs. To address this question, we fed the empirically matched EC-patterns into pydentate, adjusting the synaptic weight of perforant path (PP) inputs to lead to plausibly sparse activity in DG (Fig. 1h, i). To examine if the DG network performed pattern separation on these inputs, we simulated pairs of parallel trajectories with varying distance (Supplementary Fig. 1b).

A common way to define pattern separation is as a decrease in Pearson's correlation ($R$) between activity pattern pairs from the input area (EC) to the output area (DG). To explore pattern separation in both a rate- and a phase-coding domain, we next devised a method to perform an analogous analysis for phase-coded inputs. Specifically, we computed either the mean rate or the mean phase of spikes within individual theta cycles (100 ms time bins), over the course of each trajectory. The information within each theta cycle can be thought to be represented by a theta-vector (Fig. 1j), where, in polar coordinates, the angle is defined by the mean phase of spikes ($\theta$) and the magnitude is defined by the firing rate ($r$)[3,5]. For each GC, a trajectory is thus represented as a sequence of theta-vectors, one per 100 ms bin (Supplementary Fig. 1d, e). In order to isolate phase and rate information for independent analysis, it was necessary to transform individual theta-vectors from polar to Cartesian coordinates ([$r$, $\theta$] into [$x$, $y$]; Fig. 1h). This is because phase-values of cycles without spikes are undefined, precluding correlation calculation, but the corresponding [$x$, $y$]-coordinates are well defined (namely as [0,0]). We then isolated phase-codes by holding rate constant and vice versa (see methods).

This allowed us to compute Pearson's correlations for pairs of EC input patterns ($R_{in}$) and the corresponding pairs of DG output patterns ($R_{out}$), for only phase or only rate information (Fig. 1k, l, respectively). Greater distances between parallel trajectories led to decreasing correlations of EC patterns ($R_{in}$) for both phase- and rate-code. Note that the presently reported maximal input correlation values are not unphysiologically low, but reflect a necessary methodological idiosyncrasy, namely the need to measure correlations based on 100 ms (theta) time-windows. Such short time-windows are known to artificially reduce measured rate-correlation values several-fold[22,32]. Identical spike trains (those of our similar trajectories) lead to correlations of $R_{rate} = 0.4$ when assessed with 100 ms time-windows and $R_{rate} = 0.85$ when assessed with 2 s time-windows. In other words, the actual underlying data covers a similar range as previous studies[22,32] when compared appropriately, i.e., based on similar time-windows (note that, to the best of our knowledge there are no previous studies assessing phase code correlation). Indeed, "behaviorally" identical trajectories (distance = 0 cm) led to input correlations indistinguishable from the shown maximal input correlation (distance = 0.5 cm, Supplementary Fig. 7), suggesting we cover the behaviorally plausible range (given the constraint of a Poisson process).

We found that the network reliably separated more correlated input patterns in both phase- and rate-code (Fig. 1k, l, data points were consistently below unity, where unity indicates that input and output correlations are identical). We quantified total pattern separation as the area between mean output correlations and unity[32] (Fig. 1k, l, insets), revealing that both rate and phase effects are significant (one-sample $t$ test for deviation from 0, $n = 10$, rate: mean area=0.118 ± 0.003, $p = 9.6 \times 10^{-11}$; phase: mean area=0.049 ± 0.003, $p = 1.4 \times 10^{-8}$, Supplementary Table 1). Notice that a direct comparison of pattern separation between phase- and rate-coding is problematic, e.g., because the two codes span different ranges of input similarity. Nevertheless, these data suggest that pattern separation in the DG may operate via both a phase- and a rate-code.

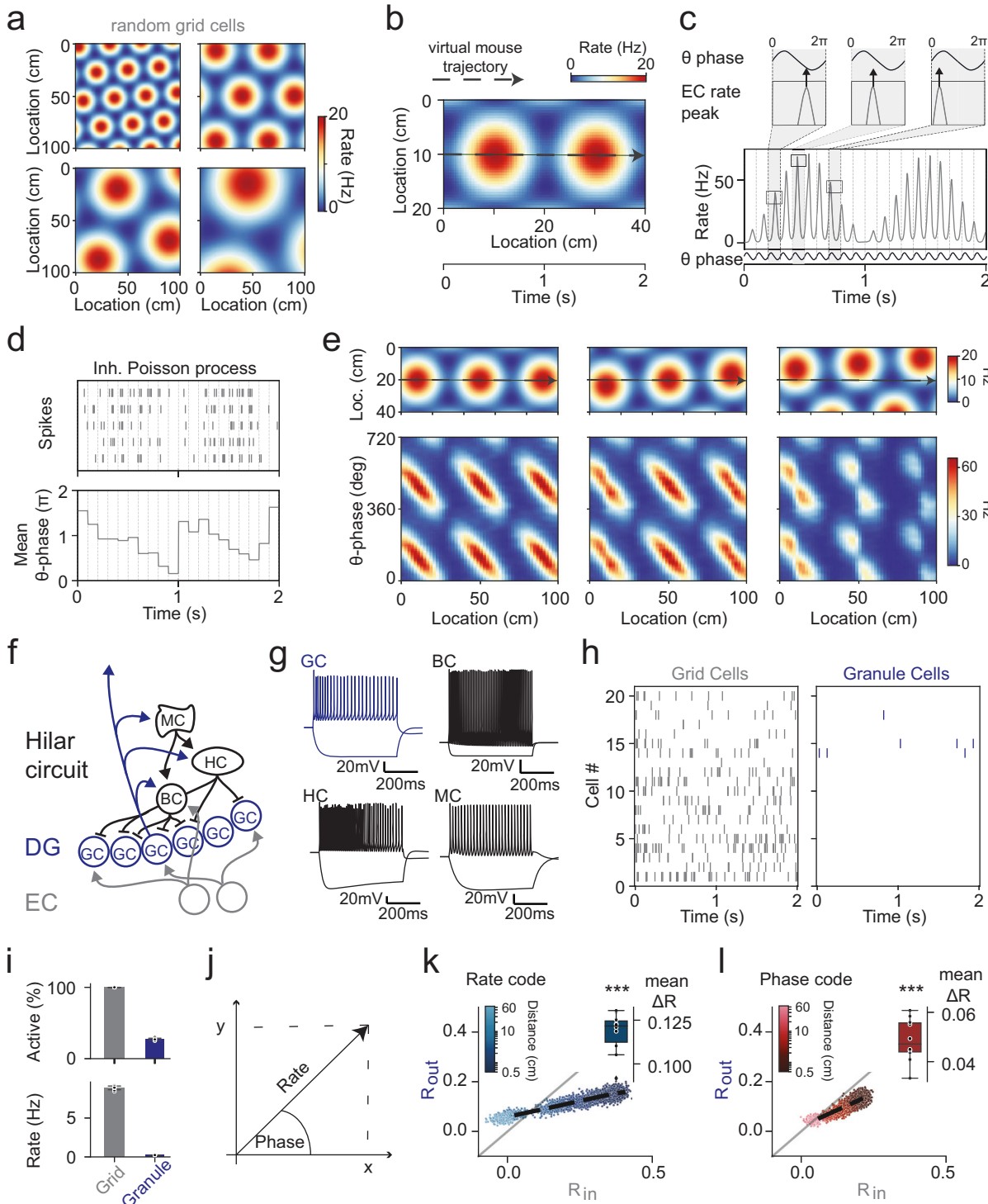

## Impact of EC phase-code on DG rate-code

While pattern separation is thought to aid downstream learning, decorrelation metrics do not directly inform us about the information content of neural activity patterns (in either rate- or phase-codes). Specifically, it is conceivable that DG inhibitory microcircuits increase decorrelation simply by removing information. This is important, because a pattern separation mechanism which simply degrades information might lead to worse rather than improved auto-associative encoding in CA3. We therefore next investigated the theta-phase structure and information content of EC and DG activity, and the latter's dependence on inhibitory microcircuit motifs (Fig. 2).

As expected, phase precession of grid cells (see Fig. 1d, e) led to EC phases distributed relatively broadly across the theta cycle (Fig. 2a, b, gray histograms), consistent with experimental data[10,30,33]. In other words, the EC phase-code makes use of much of the phase-coding space. By contrast, in the intact DG circuit GC spiking was concentrated within the early theta cycle, implying less of the possible coding space was utilized (Fig. 2a, upper left). Strikingly, the peak of activity in DG (the output area) seemed to precede the peak of activity in EC (the input area). Indeed, the mean phase vector of DG was earlier in the theta cycle than that of EC (Supplementary Fig. 3). This highly counterintuitive pattern is consistent with available data in vivo, which also shows (i) a more restricted phase distribution and (ii) an earlier

**Fig. 1 | Pattern separation of both *rate*- and *phase*-codes. a** Examples of spatial rate profiles for four randomly generated entorhinal cortex (EC) grid cells with empirically plausible variation in grid scales and orientations[27,28]. **b** A linear trajectory (20 cm/s, black dashed arrow) of a virtual mouse was assumed to simulate EC population activity. **c** A 10 Hz theta oscillation was assumed to simulate a phase precessing[29] probability distribution used to modulate an inhomogeneous Poisson process. **d** Top: five random Poisson instantiations; bottom: resulting mean phase. **e** Illustration of resulting spatial phase-codes for grid cells with varying orientation (compare Hafting et al.[10]). **f** Schematic of biophysically realistic dentate gyrus (DG) model (pydentate) containing granule cells (GC), basket cells (BC), HIPP cells (HC) and mossy cells (MC). **g** Cell-specific firing patterns of pydentate[32]. **h** Representative example showing the sparse GC spiking pattern that EC grid cell spiking evokes in pydentate (20 of 200 grid and 20 of 2000 GCs). **i** Mean grid and granule cell activity levels (data presented as mean ± SD). **j** Illustration of how "theta-vectors" are transformed from polar [phase, rate] to Cartesian [$x$, $y$] coordinates. For subsequent analyses, rate-code was isolated by holding phase constant and vice versa. **k** Pearson's correlation coefficient between pairs of trajectories of varying distance for EC ($R_{in}$) and DG ($R_{out}$) population-rate-codes. The inset shows the mean area between the data (black dashed line) and unity (gray line), quantifying overall pattern separation (asterisks indicate significance in two-tailed one-sample t-test for deviation from 0, rate: mean area=0.118 ± 0.003, $p = 9.6 \times 10^{-11}$). **l** Same as **k** but for population-phase-codes (phase: mean area=0.049 ± 0.003, $p = 1.4 \times 10^{-8}$). $n = 10$ grid seeds throughout. Box plots show the median, the interquartile range (box) and the data range w/o Tukey-outliers (whiskers). Source data are provided in Source Data.xlsx. Also see Supplementary Table 1 for statistics and Supplementary Fig. 1.

peak for DG than EC[11] (see "Discussion"). Note that the "early" versus "late" theta terminology used throughout this paper is a narrative convenience - since theta is a cyclical phenomenon "late" theta grid cells can also drive "early" theta GCs[11], though in the present case this would imply implausibly long synaptic delays (>50 ms, see "Discussion").

Next, we tried to pinpoint the circuit mechanism responsible for the absence of late-theta GC spikes. To this end, we selectively removed feedforward, feedback, or total inhibition in our model (Fig. 2a). We found that removing feedback, but not feedforward inhibition substantially broadened the DG theta-phase histogram. This suggests that, while the presence of feedback inhibition is likely important for sparse coding and pattern separation, it may interfere with the transmission of phase information from EC to DG.

We hypothesized that feedback inhibition may partially translate EC phase-code into a GC population-rate-code. The reason is that the timing of input spikes is crucial in determining which cells become active and trigger feedback inhibition[19,34]. To test this idea, we devised a shuffling procedure which fully conserves the mean frequency and population-level theta-phase distribution in EC, but removes all phase information about the animal's location, i.e., phase precession (Fig. 2b, d). Interestingly, shuffling slightly increased GC firing rates, despite identical rates in EC (Fig. 2c, see Supplementary Table 2 for statistics). The effect was clearest in the disinhibited network, suggesting phase-coding and intrinsic GC properties interact to slightly increase sparsity. However, since the effect was small, we do not further consider it. Note that the potential effect of increased firing for shuffled data works in the opposite direction of our main results below, rendering these conservative.

We then inspected how shuffling affected pattern separation, including the potential roles of feedback and feedforward inhibition (Fig. 2e–g). Shuffled EC inputs were still significantly decorrelated by the full DG network (Fig. 2e,f; one-sample t-test for deviation from 0, $n = 10$ each, rate: mean area=0.137 ± 0.004, $p = 4.0 \times 10^{-11}$; phase: mean area=0.02 ± 0.001, $p = 2.6 \times 10^{-3}$, Supplementary Table 3). Note that, for shuffled phase-codes the range in input similarity is already reduced to almost zero, precluding a meaningful interpretation of these results. The following thus pertains to the rate-codes, where input correlations are identical (Fig. 2g). First we observed that, consistent with previous results[32], both feedforward and feedback inhibitory microcircuits contributed to pattern separation (Fig. 2g, see Supplementary Table 4 for statistics). This was the case for both shuffled and non-shuffled EC inputs. Interestingly, correlation analysis additionally suggested an improvement of pattern separation of rate codes due to phase shuffling (Fig. 2g). This may be, because GC coincidence detection translates EC phase randomization into a GC population code randomization in a supra-linear way. Inversely, it suggests that phase information in EC per se actually decreases DG pattern separation. However, in this context it is important to reemphasize that Pearson's correlation coefficient is insensitive to the informational content of a signal. For instance, randomizing spikes over spatial bins may improve

decorrelation metrics, simply by removing spatial information[35]. However, before we explore this possibility, it is worth noticing that the apparent improvement of pattern separation due to shuffling was most pronounced in networks containing the feedback inhibitory microcircuit (Fig. 2g, full, no ff). By contrast, in networks in which the feedback inhibitory microcircuit was disabled (Fig. 2g, no fb, disinh), the effect of shuffling was dramatically reduced. This effect of phase-shuffling on rate-correlations suggests a role of the feedback circuit in mediating between the two coding-schemes.

To quantify how well spatial information from EC is retained in the DG, we next performed spatial information analysis[24] (note that this assesses rate- but not phase-codes). If EC phase information is recoded into GC rate codes, then perturbing EC phase information should disrupt GC spatial (rate) information. To directly test this idea, we calculated the mean spatial information within the EC and GC populations with and without EC-phase shuffling. The spatial information content within EC differed slightly between shuffled and non-shuffled data (Fig. 2h left, from 0.317 ± 0.005 to 0.316 ± 0.006, paired t test, $p = 4.3 \times 10^{-3}$, $n = 10$), but given the tiny effect size this difference is not meaningful. By contrast, GC spatial information was substantially decreased by shuffling, across all networks (Fig. 2h right, see Supplementary Table 5 for statistics). This suggests that the apparent improvement in pattern separation due to shuffling was associated with a loss of spatial information. Notably, as for correlation analysis, the effects were most pronounced in networks containing feedback inhibition.

Next, we controlled for changes in mean GC rate, due to removing inhibition. Specifically, we wanted to ascertain that the observed effect did not arise from increased mean GC activity. Changes in sparsity might produce confounding effects for a variety of reasons, e.g., due to non-linear effects in cell recruitment or due to biases in the measurement of spatial information per spike[36]. We therefore ran simulations in which we systematically varied the PP input-strength (synaptic weight), and with it GC sparsity, allowing us to choose PP weights that control for mean GC activity (Supplementary Fig. 4). Our findings remained robust (Fig. 2i): Disrupting EC phase information led to decreases of spatial rate information across networks, with the largest effects occurring in networks with feedback inhibition (Fig. 2i, see Supplementary Table 6 for stats). The relative increase in spatial information with respect to the shuffled data was greatest in the full network (Fig. 2j, see Supplementary Table 7 for statistics) and most saliently reduced in circuits containing no feedback inhibition (Fig. 2j).

Finally, we asked if the observed effects would remain robust to i) neural membrane noise and ii) additional noisy GC inputs. This is important, because time-codes might be highly sensitive to neural noise or potential interference. Furthermore, the presently modeled grid-cell inputs reflect only one of the many functionally and anatomically diverse GC inputs in vivo[33]. Adding noise (or noisy inputs) reflects a parsimonious way of probing if our results would remain robust in the presence of these non-modeled GC inputs. First, we added Gaussian noise to the GC membrane potential, where

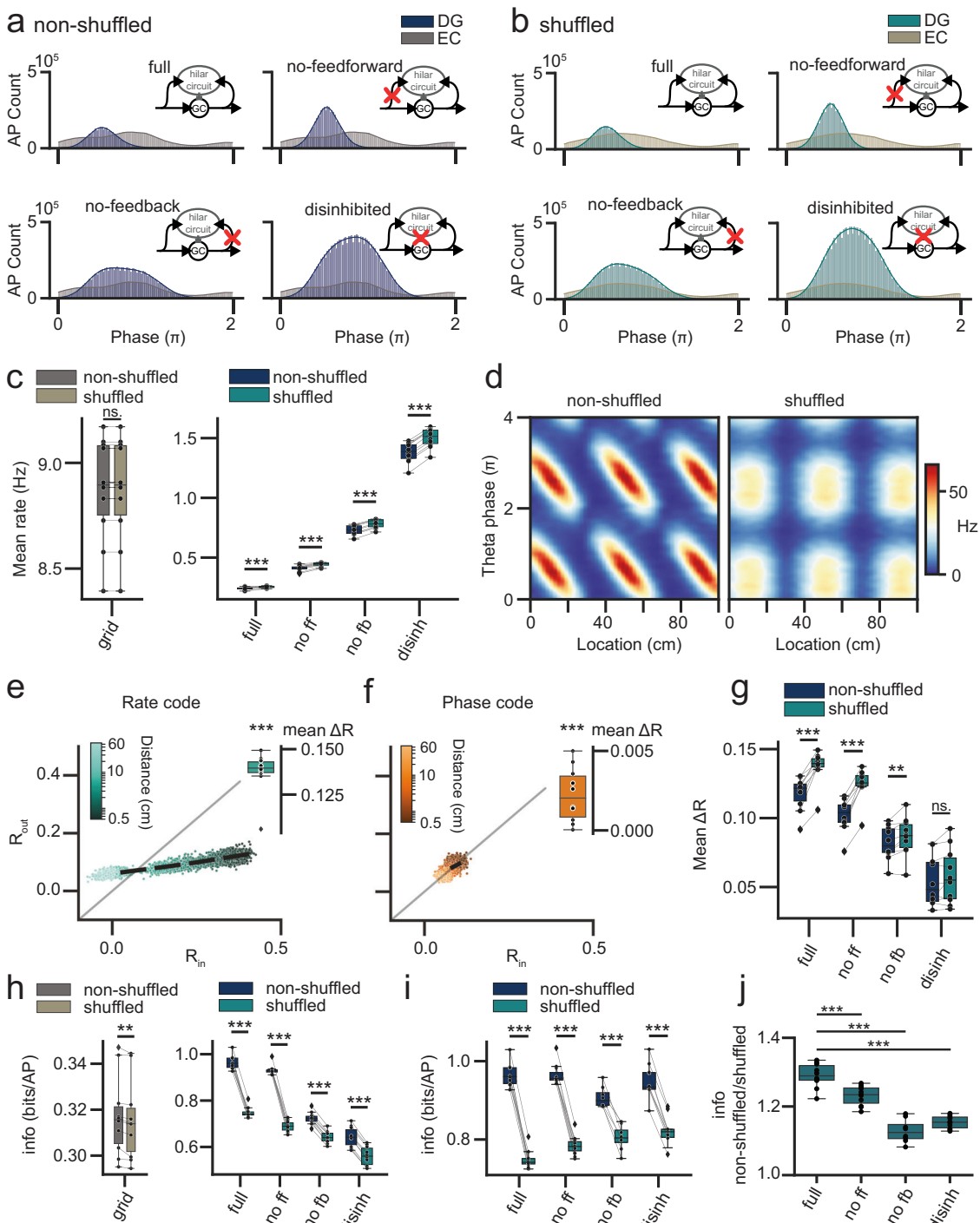

**Fig. 2 | Spatial phase information in EC is translated to spatial rate information in DG. a** Theta-phase histograms for EC and DG spikes, depending on the presence or absence of DG inhibitory circuit motifs (full: full pydentate model; no ff: feedforward inhibition removed; no fb: feedback inhibition removed; disinh.: fully disinhibited network). **b** Theta-phase histograms for data with shuffled EC phases. Note that, since EC phases were shuffled according the probability distribution implied in the non-perturbed theta phase histogram (**a**), the resulting theta-phase distributions are by design minimally affected. **c** Mean firing rates in EC (left) and DG (right) for shuffled and non-shuffled data. **d** Phase precession plots[10], showing the degradation in phase-code for shuffled data despite the conserved theta-phase histograms. **e, f** Pattern separation of rate- and phase-code in the full network as in Fig. 1i, j, but for shuffled data (asterisks indicate significance in two-tailed one-sample *t* test for deviation from 0, *n* = 10 grid seeds, rate: mean area=0.137 ± 0.004,

*p* = 4.0 × 10⁻¹¹; phase: mean area=0.02 ± 0.001, *p* = 2.6 × 10⁻³). **g** Quantification of total pattern separation for shuffled and non-shuffled data across network conditions. **h** Spatial information analysis across conditions. **i** Spatial information analysis across conditions with controlled GC-spike rates (for conditions with impaired inhibition, perforant path input weights were decreased to obtain GC firing rates similar to the full network). **j** Data were normalized to show the fractional improvement of DG spatial information given EC phase information. *n* = 10 grid seeds in all panels. Box plots show the median, the interquartile range (box) and the data range w/o Tukey-outliers (whiskers). Asterisks in **c**, **g**–**j** indicate statistical significance in Bonferroni post-tests following significant two or one-way ANOVA. Source data are provided in Source data.xlsx. Full statistics are summarized in Supplementary Tables 2–7.

the initial amplitude of noise was matched to in vivo recordings[37] (Supplementary Fig. 5a, b). We found that the feedback-inhibition mediated effect of EC phase-shuffling on DG rate codes remained robust even if noise was increased to 5x realistic levels (Supplementary Fig. 5c, d). We also investigated a more specifically defined additional input from the lateral entorhinal cortex (LEC), thought to contain contextual information. The modeled grid cell inputs (which we for brevity referred to as from EC) originate in the medial EC (MEC). LEC inputs are particularly interesting in the present context because they are also theta-modulated, but their modulation appears to be countercyclical to MEC[38]. We added such a counter-cyclically modulated LEC input simulating the same context with either similar (Supplementary Fig. 6) or identical (Supplementary Fig. 7) "context"-input patterns (where the latter contains additional spatial phase and rate information). This led to marked changes in the theta-phase distribution of GC spiking, but the feedback-inhibition dependent effect of MEC phase-coding on DG rate-coding remained robust. For simplicity, we will continue to refer to MEC as EC below.

These results suggest that spatial phase-codes in EC are in part converted to spatial rate-codes in DG, a computation we term phase-to-rate recoding. The data further point to the feedback inhibitory microcircuit as the principal mediator of these effects, and suggest that the computation is highly robust to noise or potential interference from additional inputs.

### Partial degradation of DG phase code

Our previous results suggested that the feedback inhibitory microcircuit leads to a constriction of the utilized phase-coding space from the entire theta cycle to a smaller portion of the theta cycle (Fig. 2a). This suggests the DG in general, and its feedback inhibitory microcircuit in particular, may partially degrade phase-codes arriving from EC. However, so far we have only assessed the effects on rate-coded information[24], leaving unclear how exactly phase-coded information is affected. In this context it should perhaps be emphasized that the presence of phase-to-rate recoding is ultimately logically independent of the degradation of the original phase code - it requires only that upstream phase-codes improve downstream rate-codes. Nevertheless, the question to which degree phase codes are conserved in the present context is important in and of itself (see discussion).

We therefore next used a perceptron approach to systematically investigate the "decodability" of both rate- and phase-codes in DG (Fig. 3). Decodability was measured as the learning speed of a perceptron and is expected to increase with both information content and decorrelation[25]. The perceptron was trained to decode pairs of adjacent trajectories with varying distance on either the grid cell population or the granule cell population given either shuffled or non-shuffled EC codes (Fig. 3a, b).

Learning speeds clearly increased with distance with an apparent saturation beginning at approximately 15 cm, likely because parallel trajectories no longer shared any individual grid fields. Furthermore, as expected, the decodability of the EC rate-codes was not affected by phase-shuffling (Fig. 3c, top; the lines are fully superimposed), while the decodability of EC phase-codes was severely degraded (Fig. 3c, bottom). Notice that, by design, phase shuffling does not perturb the population code allowing the phase-perceptron to still learn even if no phase information is present (i.e., from phase-shuffled EC data, Fig. 3c, bottom: orange trace). Importantly, perceptron learning speed is directly affected by population sparsity, rendering a direct comparison between EC and DG (or different circuit conditions) uninformative (note the different y-axis scales in Fig. 3c–g, see "Methods"). To quantify the impact of EC phase information, we thus normalized the non-shuffled to shuffled data, which have similar sparsity and thus represent an internal control. This revealed an approximately two-fold improvement in phase decodability at 15 cm (Fig. 3c, right; one-sample $t$ test against 1, $n = 10$, mean ratio = 1.950 ± 0.028; $p = 7.2 \times 10^{-11}$). These results

suggest that phase-shuffling selectively removed the phase information from the EC patterns, allowing us to directly ask how such selective removal would affect not only rate- but also phase-coding in the DG.

Perceptron analysis confirmed our previous finding that feedback inhibition partially translates EC phase information into a DG rate code. Removing EC phase information via shuffling significantly interfered with perceptron decoding of DG rate-codes within the full network (Fig. 3d, top). The analysis additionally showed that DG phase-coding was also disrupted by EC-phase shuffling (Fig. 3d, bottom). However, the disruption of phase decodability in DG was far smaller than in EC (compare Fig. 3c, d). This is consistent with the notion that phase information in DG is already impaired by feedback inhibition. To directly test this idea, we again investigated the impact of inhibitory network motifs on rate and phase decodability (Fig. 3e–g).

We first selectively removed feedforward inhibition (Fig. 3e), which led to no marked changes in the relative decodability of rate- and phase-codes as compared to the full network (Fig. 3d). By contrast, removing feedback inhibition (Fig. 3f) almost entirely removed the effect of shuffling for the rate code (Fig. 3f, top), while simultaneously amplifying its effect on the phase-code (Fig. 3f, bottom). This confirms that feedback inhibition simultaneously degrades phase-coding but improves rate coding in the DG. Finally, disrupting both feedforward and feedback circuits led to learning patterns similar to the no feedback or EC networks, namely no difference in rate coding and a significant degradation in phase-coding due to shuffling (Fig. 3g).

These findings further support the idea that feedback inhibition mediates phase-to-rate recoding within the DG. To statistically compare results, we again normalized the non-shuffled data to shuffled data (Fig. 3h–k) to control for effects arising from mere changes in mean GC activity levels. Indeed, when feedback inhibition was present, EC phase information improved decodability by approximately 20% for both rate- and phase-codes (Fig. 3h, i, full, no ff; see Supplementary Tables 8 and 9 for statistics). However, when feedback inhibition was removed, the improvement for rate decodability disappeared, while phase decodability improved to approximately 40% (Fig. 3h, i, no fb, disinhibited).

Next, we again controlled for GC activity rates in an alternative way (by reducing PP input weights), (Fig. 3j, k, see Supplementary Tables 10 and 11 for statistics). In particular, the question arises if achieving the sparse DG code via phase-to-rate recoding has advantages when compared to a similarly sparse code achieved without inhibitory microcircuits. As for spatial information analysis, the results further support the notion of phase-to-rate recoding. While this additional analysis suggested feed-forward inhibition may also play a role, the dominant effect was still for feedback inhibition (compare Fig. 2j and Fig. 3j). Interestingly, the apparent boosting of phase decodability for circuits without feedback inhibition disappeared (Fig. 3k), suggesting it is sensitive to overall GC sparsity.

Finally, we used an alternative information measure, namely positional information[39,40], to directly assess the impact of feedback inhibition on DG phase-information. While the perceptron analysis is suggestive, it still does not unambiguously address the question to which degree phase information is degraded by feedback inhibition. By contrast, positional information analysis allows to directly assess phase information. Briefly, positional information is defined as the reliability of occurrence of a particular phase (or rate) within a particular position across trials[40]. Consistent with previous research[40], we found positional information to be strongly dependent on the employed smoothing scale (Supplementary Fig. 8a, see "Methods"). Nevertheless, overall the results confirmed that the prominent decrease in rate information when eliminating feedback inhibition was accompanied by an increase in phase information (Supplementary Fig. 8b, c). Notably feedback inhibition significantly reduced but did not eliminate phase information in DG. In other words, spatial phase information is partially degraded by the DG feedback circuit.

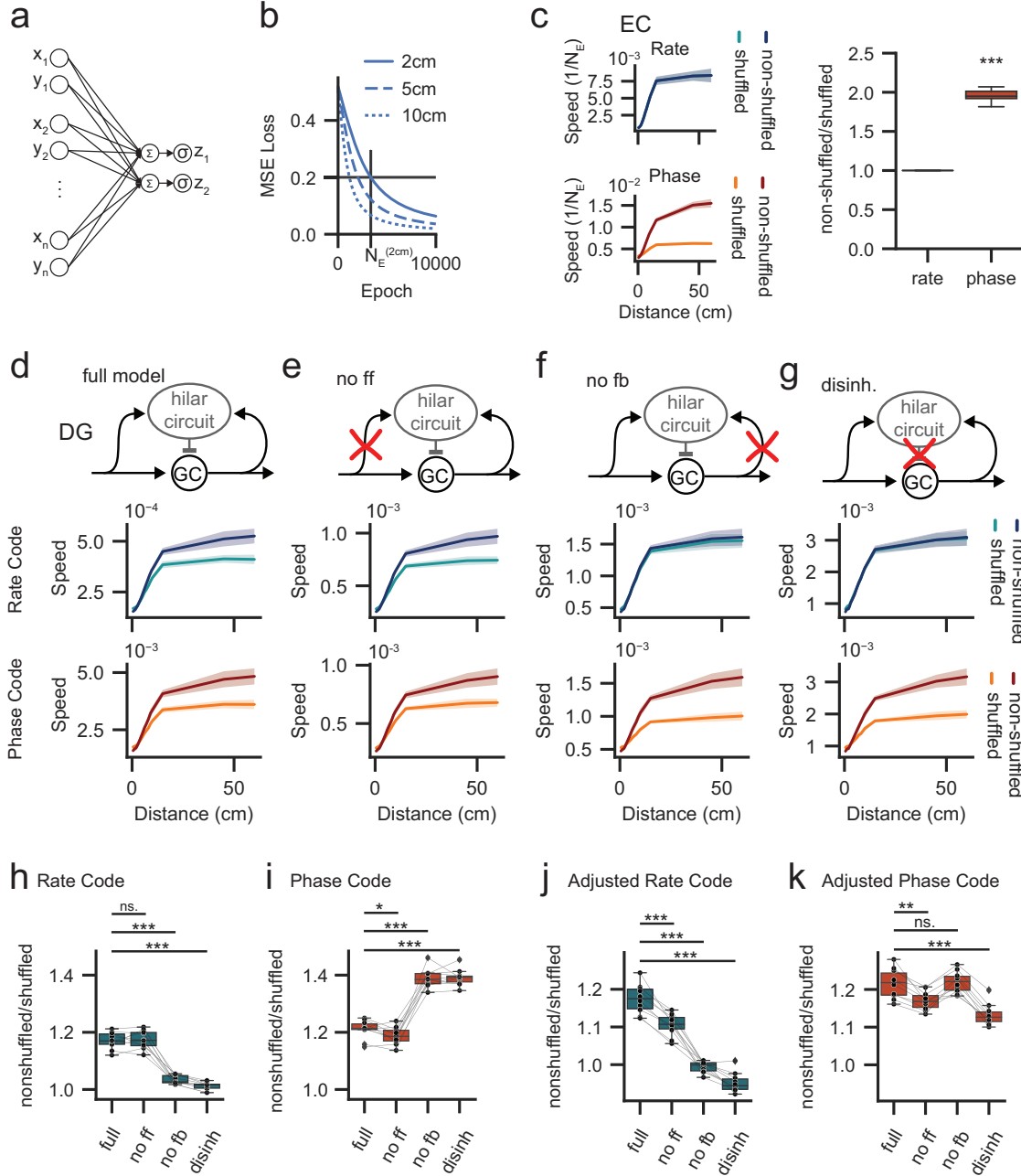

**Fig. 3 | Perceptron decodability of EC and DG rate- and phase-codes confirms phase-to-rate code translation. a** Schematic of perceptron decoder. *X* and *Y* coordinates (see Fig. 1j) for each cell and time bin are fed into the perceptron as a flattened array (full: full pydentate model; no ff: feedforward inhibition removed; no fb: feedback inhibition removed; disinh.: fully disinhibited network). **b** Illustration of perceptron training. Learning speed (decodability) was assessed as $1/N_E$, where $N_E$ indicates the number of epochs required for the Root Mean Squared Error (MSE) loss to cross a threshold of 0.2 (see Cayco-Gajic et al.[25]). **c** Effect of phase-shuffling on *rate-* and *phase-codes* in EC. Left: learning speed ($1/N_E$) for varying distances between parallel trajectories (0–60 cm). Right: non-shuffled results divided by shuffled results for 15 cm distance. Asterisks indicate significance in two-tailed, one-sample *t* test against 1, no multiple comparisons, *n* = 10, mean ratio=1.950 ± 0.028; *p* = 7.2 × 10⁻¹¹. **d–g** Learning speeds for rate (top) and phase

(bottom) codes in DG networks with selectively disabled inhibitory circuit motifs as in Fig. 2 (full model: full pydentate model; no ff: feedforward inhibition removed; no fb: feedback inhibition removed; disinh.: fully disinhibited network). **h** Fractional improvement of DG rate-code learning speed (non-shuffled normalized to the phase-shuffled data) across conditions (at distance = 15 cm). **i** Same as (**h**) but for phase-code. **j** Analogous to (**h**), but where differences in mean GC-firing rates arising from the respective circuit interventions are controlled for by adjusting PP input strength to achieve comparable GC rates. **k** Analogous to (**i**), but with controlled GC-firing rates. *n* = 10 grid seeds in all panels. Box plots show the median, the interquartile range (box) and the data range w/o Tukey-outliers (whiskers). Asterisks indicate significance in (two-tailed) Holm–Sidak post test, following significant one-way ANOVA. Source data are provided in Source data.xlsx. Full statistics are summarized in Supplementary Tables 8–11.

While these results suggest complex interactions of microcircuit function, sparsity and information (also see Supplementary Fig. 4), they unambiguously support the notion of feedback inhibition mediated phase-to-rate recoding.

**Tempotron decodability of combined DG phase-rate code**
We next asked what the combined effect of the changes to DG rate- and phase-codes might be. In particular, the transformation of GC spike-trains to rate and phase vectors relies on various assumptions

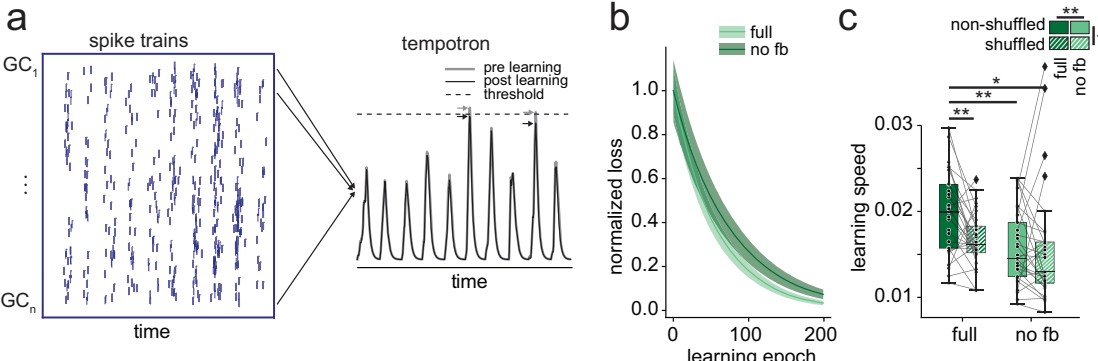

**Fig. 4 | Tempotron decodability of GC spike trains. a** Illustration of tempotron learning of spike trains. Briefly, GC input spike trains (left) were temporally convolved into the tempotron (right, see methods). For learning, input weights of each GC were adjusted such that correct patterns superseded threshold, but incorrect patterns did not. The example illustrates learning for an incorrect pattern, where a false response (gray trace/arrows) is transformed into an accurate non-response (black trace/arrows). **b** Illustration of tempotron learning (normalized loss over learning epochs) for the full and no-feedback (no fb) networks with controlled GC

rates as in Fig. 3j, k (mean ± sd). **c** Tempotron learning speed for the full and no-feedback (no fb) DG networks given both non-shuffled and shuffled EC inputs. Asterisks indicate significance of main effects in two-way repeated measures ANOVA (top) or post-tests (bottom). $n = 29$ grid seeds. Box plots show the median, the interquartile range (box) and the data range w/o Tukey-outliers (whiskers). Source data are provided in Source data.xlsx. Full statistics are summarized in Supplementary Table 12.

(most importantly that the relevant information is contained in the mean rate or phase within a theta cycle). We thus sought a complementary analysis approach which assesses decodability based on unprocessed spike trains.

A minimal approach to assess decodability, analogous to the perceptron, but operating on spike trains is the tempotron[26]. Briefly, the tempotron is a single leaky integrate-and-fire (LIF) neuron driven by exponentially decaying synaptic currents generated by N input spike-trains. During training, the weights of afferents are adjusted such that a simple threshold can distinguish between input patterns (Fig. 4a). Analogous to the perceptron, decodability can then be assessed via the learning speed (Fig. 4b, see "Methods").

To explore how tempotron decodability was affected by phase-to-rate recoding, we trained a tempotron on pairs of trajectories (15 cm distance) for shuffled and non-shuffled data in both the full network and the no-feedback network with controlled GC-rates (as in Fig. 2i, j and Fig. 3j, k). We focus on the feedback as opposed to the feedforward circuit because it shows the most pronounced effects and it is more tightly experimentally constrained[32]. The results were broadly consistent with the perceptron results. Both EC-phase shuffling and the removal of DG feedback inhibition interfered with tempotron decodability (Fig. 4c, Supplementary Table 12 for statistics). Finally, we probed if some general spike-train characteristic could predict the relative contribution of individual GCs to tempotron learning (Supplementary Fig. 6). We found no simple predictive characteristic, suggesting that changes in theta-organized spike times and synchrony at the population-level drive the effects. These results suggest that phase-to-rate recoding improves the overall decodability of combined phase-rate codes in DG spike trains.

## Mechanism of phase-to-rate recoding

Next we asked how phase-to-rate recoding can be explained mechanistically. In general terms, feedback inhibition must be capable of suppressing APs in the GC population in a way that increases spatial information (recall that spatial information measures rate-coded information[24]).

We hypothesized a simple mechanism whereby phase precession induces spatially selective inhibition of spikes: First we reasoned that, in the absence of inhibition, EC phase-precession should be inherited by some GCs (Fig. 5a). For instance, some GC (GC1 in Fig. 5a) is likely to receive supra-threshold inputs in one place (p1) only in the late theta cycle (green), while in another place (p2) the cell would tend to fire

only in the early theta cycle (blue). Some other GC (GC2 in Fig. 5a) will receive inputs that lead to the inverse pattern. If we now consider these two GCs in the presence of the feedback circuit, our previous results suggest that phase specific firing will be translated to spatially selective inhibition of spikes (Fig. 5b). GCs driven to spike in the early theta cycle will recruit inhibition leading to the suppression of GC spikes in the late theta cycle. In the given example this means that GC2 will inhibit GC1 in p1 and GC1 will inhibit GC2 in p2 (Fig. 5b, black panel). By contrast, if either EC phase-coding or DG feedback inhibition is impaired (Fig. 5b, gray panels), spatial information and rate decodability decrease.

To test this hypothesis, we reexamined our modeled GC spike patterns. Our hypothesis predicts that some random subset of GCs should have increased spatial information due to the spatially selective inhibition of late-theta spikes. To identify these cells, we first calculated the cell-wise spatial information for both full and no-feedback networks. Inspecting the 50 cells with the highest spatial-information difference across the whole trajectory confirmed that "inhibited spikes" tended to occur in late theta (Fig. 5c, green dashes in the lower but not upper panel). Spatially selective inhibition becomes most clearly discernible when inspecting the population-code for an individual place in the no feedback condition (Fig. 5d). Specifically, cells 1–25, which otherwise have a clear spatial preference early in the trajectory (Fig. 5c, p1–10) fire numerous "late-theta noise spikes" at this final position of the trajectory when feedback inhibition is absent (Fig. 5d, green arrow). These are almost fully suppressed when feedback inhibition is present (Fig. 5d, top two panels).

Furthermore, for late-theta inhibition to act in a spatially selective manner, an intact EC-phase code is required (Fig. 5d, compare top two panels). Indeed, perturbing the EC-phase code via shuffling led to GC codes with similar late-theta inhibition and of similar sparsity, but without the marked spatially selective AP-inhibition. Specifically, shuffling led to "early-theta noise spikes" (Fig. 5d, blue arrow), precisely as hypothesized (compare Fig. 5b). This confirmed that the effect was not driven by changes in sparsity (also see Supplementary Fig. 10).

Finally, we asked if spatially selective AP-inhibition occurs only in late theta. We have introduced the mechanism as if early theta GC spikes recruit inhibition, but are not themselves affected by it. This simplification helps to illustrate the mechanism, but misleadingly suggests the necessity of total GC suppression at a particular theta-phase. Since the actual recruitment of inhibitory interneurons will occur locally and staggered in time[32,41], we reasoned total suppression

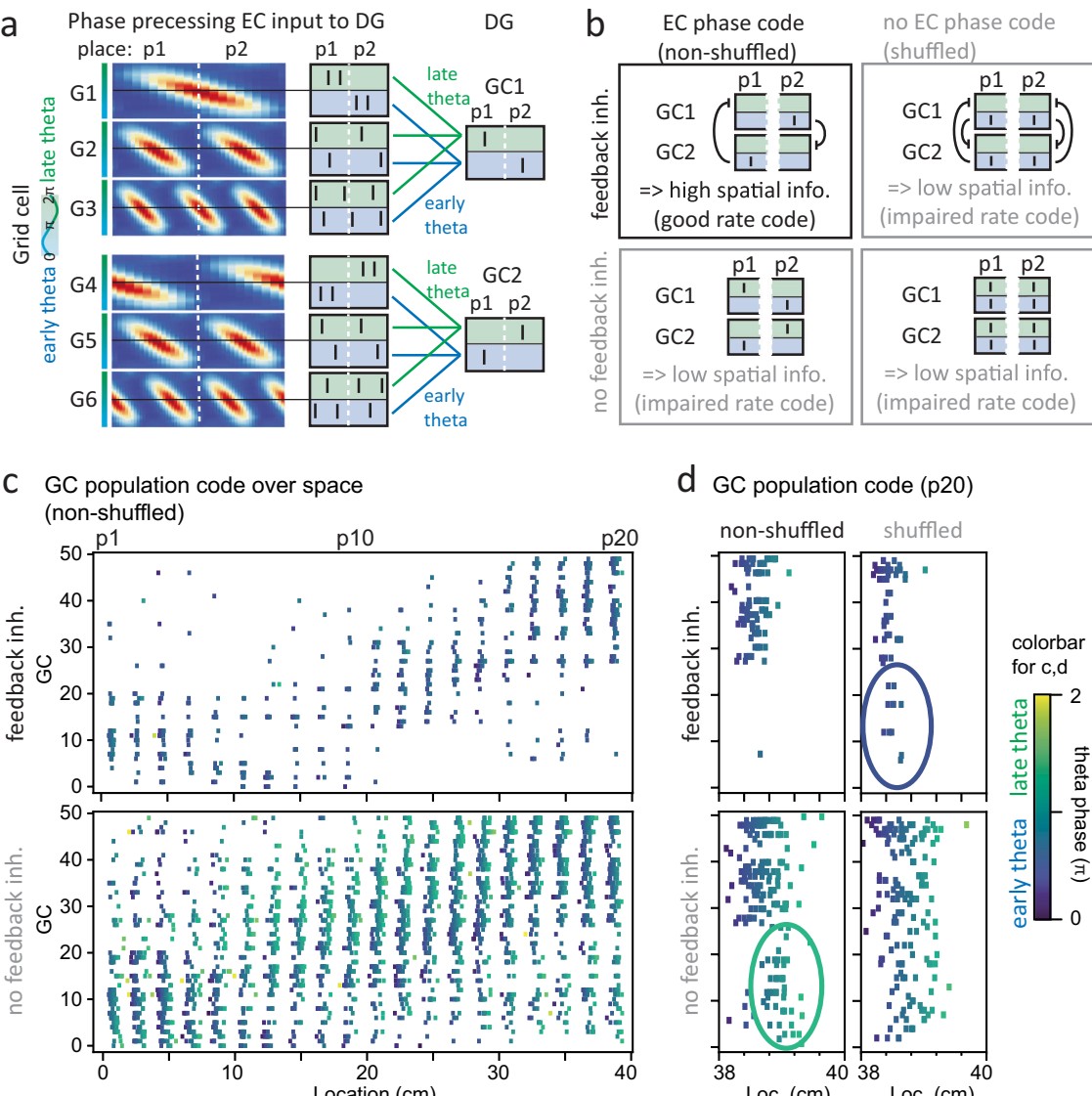

**Fig. 5 | Mechanism of phase-to-rate recoding.** A mechanistic hypothesis (**a**, **b**) and the supporting data (**c**, **d**). **a** Phase-place coupling in DG will be inherited from EC. Consider a random set of EC grid cells (G1-G6) across two places (p1, p2), with typical phase-place relationships (intensity plots in left column). Typical firing patterns of these grid cells across the two places (middle column) will be distributed between early (blue) and late (green) theta cycle. Given convergent input of grid cells on GCs (GC1, GC2; right column), the latter will inherit phase-place coupling, implying spatially selective late-theta firing for some random set of GCs (GC1,2) **b** DG rate code quality under four conditions, namely the presence or absence of EC *phase-coding* (columns) or DG feedback inhibition (rows). Our results indicate that early-theta GC spikes will suppress late-theta GC spikes (arched black lines from blue to green). When both DG feedback inhibition and EC *phase-code* is intact (upper left panel), GC2 will thus inhibit GC1 in p1 and GC1 will inhibit GC2 in p2, leading to high spatial information and good rate-decodability. Without feedback inhibition (lower row), late-theta spikes additionally occur in the "wrong" place (e.g., GC1 will fire in p1 undermining its spatial selectivity). If EC *phase-coding* is impaired (right column), place specific spikes are randomized between early and late theta, leading to a loss of spatial selectivity in inhibition. **c** GC spikes over the full trajectory (aggregated over 20 Poisson seeds), color-coded by theta phase with or without feedback inhibition (the 50 GCs with the largest difference in spatial information between full and no fb networks are shown, sorted by spatial preference). Each theta cycle is mapped to a place, such that the trajectory can be subdivided into place 1 to 20 (p1,...,p10,...p20). Note the sparsity of late-theta (green) spikes in the full network. **d** Magnification of the same 50 cells in p20 for the four conditions in (**b**). Note the presence of "early-theta noise spikes" (blue oval) in the shuffled and "late-theta-noise spikes" (green oval) in the no fb networks, precisely as predicted (**b**).

is not necessary and that phase-to-rate recoding should also occur within ongoing activity in early theta. To test this, we repeated the positional information analysis[13,39] based exclusively on spikes within early theta (0-π, Supplementary Fig. 11). This analysis confirmed that phase-to-rate recoding also shapes ongoing population activity and does not require its cyclic full suppression.

Together with the previous data, this strongly suggests that phase-to-rate recoding operates via a mechanism of spatially selective inhibition of spikes. The mechanism can be summarized by drawing on Robert Gütig's evocative question "To spike, or when to spike?":[4]

The DG feedback circuit decides which GCs are allowed "to spike" based on "when" EC cells spike.

### Phase-to-rate recoding mediates improved STDP at CA3 recurrent synapses

Finally, we asked what the effect of phase-to-rate recoding on STDP in CA3 might be. This is important, because STDP at recurrent CA3 synapses is thought to be a key mechanism underlying the formation of attractors and memory[42–44]. Our previous results suggest that one consequence of the computation is a more synchronous

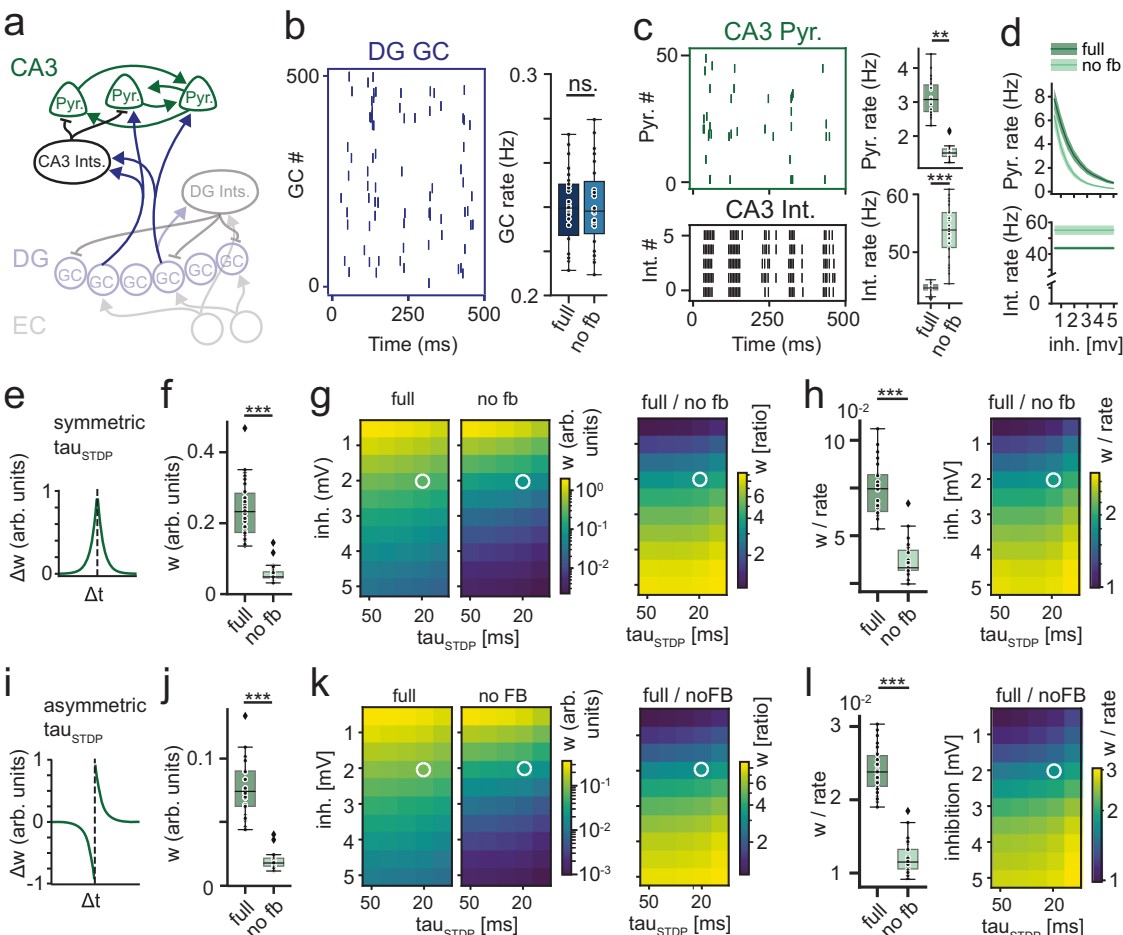

**Fig. 6 | Phase-to-rate recoding improves STDP in CA3. a** Schematic of the simple CA3 model and its input. **b** Representative examples of DG spike trains (left) and mean GC firing rates (right), comparing the full and the GC-rate-controlled no-feedback (no fb) pydentate model. **c** Representative examples of the spike trains and mean firing rates elicited by this input in samples of CA3 pyramidal cells (top) and CA3 feedforward interneurons (bottom). **d** Pyramidal and interneuron rates in response to varying the inhibitory interneuron output (mean ± sd). **e** Illustration of symmetric STDP kernel leading to synaptic potentiation independent of spike sequence (arb. units indicates arbitrary units). **f** Mean synaptic potentiation after 2 s simulation for inhibition=2 mV and tau$_{STDP}$ = 20 ms. **g** Sensitivity analysis of mean synaptic potentiation for variations in inhibitory strength (as in d) and tau$_{STDP}$ for the full (left) and no-feedback (middle) networks, as well as their ratio (right). **h** Left, mean synaptic potentiation (**f**) normalized to the equivalent ratio of pyramidal cell firing rates (as in **c**, **d**). Right, sensitivity analysis for normalized synaptic potentiation. **i**–**l** Same as **e**–**h**, but for asymmetric STDP. Asterisks indicate significance in two-sided *t* tests. Asterisks indicate significance at *p* < 0.0001 in unpaired two-tailed *t* test with Welch's correction; *n* = 30, 23 grid seeds for full and no feedback, respectively. Box plots show the median, the interquartile range (box) and the data range w/o Tukey-outliers (whiskers). Source data are provided in Source data.xlsx. Full statistics are shown in Supplementary Tables 13 and 14.

population code within individual theta cycles in DG (see Fig. 2a). This strongly suggests that the efficacy of STDP in recurrent CA3 networks should be improved. To explore this idea in more detail, we built a minimal CA3 model[45,46] with voltage-based synapses, drawing all relevant parameters from experimental data (Fig. 6a). Briefly, we created a network of 600 recurrently connected LIF CA3 pyramidal neurons, which were randomly connected to GC inputs according to known sparse connectivity rules[47]. We additionally added 60 feedforward inhibitory interneurons, one of the most prominent features of the DG to CA3 connection[48]. In the present context, this circuit is crucial to consider because it adds an additional temporal constraint on the activation of CA3 pyramidal cells: GC inputs to CA3 cells are followed with short latency by powerful feedforward inhibition[49].

To keep sensitivity analysis tractable, we again focused on the key question raised by our previous results, namely if phase-to-rate recoding via feedback inhibition provides advantages over a similarly sparse code with superior phase information but less synchrony (Fig. 6, see Supplementary Tables 13 and 14 for statistics). We thus probed the resulting model with the non-shuffled data of either the full

network or the no-feedback network, controlling for overall GC activity rates as above (Fig. 6b).

We first explored the minimal model abstracting away from numerous complexities of the real system (e.g., mossy-fiber facilitation or recurrent excitation). This minimal model was designed to isolate the impact of DG synchrony on CA3 STDP. We found that the CA3 pyramidal firing rates were significantly greater for the full network, suggesting that phase-to-rate recoding, in DG, and the resulting synchrony, lead to more efficient recruitment of CA3 pyramidal cells (Fig. 6c, top). We noticed that CA3 interneuron rates were inversely affected (Fig. 6c, bottom), suggesting that the difference in pyramidal cell recruitment may be partially driven by decreased inhibition in the full network. However, varying inhibitory output strength revealed that differences in CA3 rates remained even without inhibition (Fig. 6d, top), suggesting the difference in CA3 pyramidal cell recruitment was not primarily driven by differential interneuron recruitment. Cross-correlation analysis confirmed that both GC and CA3 principal cell activity was more synchronous for the full than the no feedback case (Supplementary Fig. 12a). These results suggest that phase-to-rate

recoding improves both the recruitment and synchrony of CA3 pyramidal cells by sparsely firing GC populations.

To investigate how this would affect plasticity in the recurrent CA3 pyramidal network, we adapted a standard STDP mechanism[45,46] to experimental data, namely the non-typical symmetric STDP recently demonstrated for CA3[42] (Fig. 6e). Synaptic weights at recurrent synapses were initialized at zero, allowing us to simply monitor the final synaptic weights as a proxy for the efficacy of STDP. To isolate the effects of DG inputs, we drove CA3 spiking exclusively by GCs (minimal model: recurrent CA3 synapses undergo plasticity but do not affect membrane potential). We found that the full network led to a several-fold increase in synaptic weights as compared to the no-feedback network (Fig. 6f). We next probed if this effect was sensitive to i) inhibitory strength and ii) the time window of STDP. Inhibitory strength was varied as above (from no inhibition to 5 mV inhibition, which almost abolished pyramidal cell activity, see Fig. 6d). The STDP time window ($tau_{STDP}$) was varied between 10 and 50 ms (covering both the canonical 20 ms and broader windows[42]). Both full and no-feedback networks showed robust plasticity over the full range of parameters (Fig. 6g, left). Furthermore, the ratio of weights between full vs no-feedback was strictly $\geq 1$ (Fig. 6g, right), indicating the full network reliably produced greater plasticity levels. The effect vanished only in the absence of CA3 inhibition, suggesting it depends on the CA3 feedforward inhibitory circuit. Within the most plausible range of parameters ($tau_{STDP}$ =20–50 ms, inh. ~1–3 mV, based on plausible CA3 rates[50]), plasticity was improved between 2 and 5 fold.

Next, we asked if the improved plasticity was driven solely by the increased firing rates of pyramidal cells (Fig. 6c, top), or if increased synchrony also made STDP per se more efficient. To assess this question we normalized total weight changes (Fig. 6f) based on the respective mean CA3 rates (which are proportional to the total number of spikes fired, Fig. 6c, d). The resulting normalized weight increases were still significantly more pronounced for the full network (Fig. 6h). This suggests that the temporal coordination of CA3 pyramidal cell firing (Supplementary Fig. 12a) improves plasticity beyond what would be expected from the improvements in pyramidal cell recruitment alone. Sensitivity analysis suggested that, for the most plausible parameter constellations (see above) the improvement was between 1.5 and 2.5 fold.

Next, we asked if these results would remain robust for a more realistic model of CA3 (extended model, Supplementary Fig. 13). Specifically, we (i) exchanged simple voltage-step for current-based synapses, creating more realistic temporal dynamics, (ii) introduced the characteristic powerful mossy fiber facilitation[51], and (iii) added the excitatory drive at the recurrent CA3 synapse based on the weights derived from STDP. We also, (iv), added an experimentally described synaptic scaling mechanism[52] to mitigate biologically implausible synaptic runaway due to the positive feedback loop between firing rate and STDP within individual pyramidal cells. Each of these factors might lead to non-trivial changes in the temporal activity patterns in CA3, and is thus likely to affect the impact of phase-to-rate recoding on CA3 STDP. Nevertheless, our results confirmed the previous findings. The full DG network led to increased CA3 population cross-correlation (Supplementary Fig. 12b) as well as greater firing rates, mean weight increases and normalized weight increases (Supplementary Fig. 13d–h). Together, this suggests that phase-to-rate recoding via the feedback inhibitory microcircuit in the DG leads to significant improvements of CA3 plasticity, due to improvements in both CA3 cell recruitment and STDP. The consequence may be a more efficient formation of CA3 attractors.

Finally, we asked if this effect might generalize to the numerous brain regions with more typical asymmetric STDP (Fig. 6i-l), i.e., if it is robust to an asymmetric STDP kernel (Fig. 6i). For this we returned to the more agnostic minimal model, changing only the STDP mechanism. We found that, while the overall induction of plasticity was

expectably lower (Fig. 6j, k), the more synchronous GC code in the full network as opposed to the (GC-rate controlled) no-feedback network still produced several fold increased plasticity levels. This was the case for overall plasticity (Fig. 6k, right), as well as when accounting for the differential recruitment of pyramidal cells (normalized weight increases, Fig. 6l). Overall, these results suggest that a beneficial consequence of phase-to-rate recoding can be improved STDP in downstream areas, driven by increased synchrony of the population-rate code.

## Discussion

In this paper, we have studied two of the most general coding schemes utilized by neurons: rate- and phase-codes. We address the problem that circuits generating essential characteristics of neuronal activity, such as sparse firing, may differentially affect rate- and phase-codes. How then is phase and rate coded information transmitted through networks that differ in their ability to support the respective coding schemes? Here, we report a candidate solution for a hippocampal circuit. We find that feedback-inhibition translates incoming phase information into a synchronized, high-information rate code, a phenomenon we term *phase-to-rate recoding* (Fig. 7).

Specifically, we studied an extended brain system involved in processing spatial information, the EC-DG-CA3 system. In this system, the dentate gyrus is thought to perform a pattern separation computation in order to decrease the (rate-code) similarity between neural patterns[21,53–56]. It is furthermore well-established that this function relies on specific properties of the underlying circuits, including the local inhibitory microcircuit[32,41,57–59]. Thus, this is an excellent example of a brain region in which the computational function likely mandates the implementation of a circuit with specific properties (feedback inhibition), which modifies not only firing rates but also affects the temporal relations between spikes. Our results suggest, that these temporal effects may actively support computations that were previously theorized in terms of rate coding alone.

Our model highlights the role of feedback inhibition in phase-to-rate recoding, while the role of feed-forward inhibition was less pronounced. The temporal features of feedback inhibition, and its role in assembly competition (through lateral inhibition) are indeed specifically suited to contribute to phase-to-rate recoding. However, we note that the model we have used is experimentally well-constrained only in terms of feedback inhibition[32], and consequently does not permit strong claims about the relative role of feedforward circuits. Under some conditions, feedforward inhibition may also be important in phase-to-rate recoding, though this would likely occur on faster time scales.

One consequence of phase-to-rate recoding was sparser but more synchronous GC activity. This is relevant because GCs project to the CA3 region, where memory engrams are thought to be stored in recurrent networks of CA3 principal cells through associative plasticity[44,60,61]. Importantly, STDP occurs most efficiently on time-scales below 100 ms[42], implying that increased GC synchrony at these time scales should strongly facilitate the formation of attractors. Our results show that increased synchrony can indeed improve plasticity several fold, both for the symmetric STDP described in CA3 as well as for asymmetric STDP, common in most other brain areas (Fig. 6).

Numerous previous studies have hypothesized *trajectory encoding*, or the *compression of temporal sequences* as the primary function of phase precession[9]. Briefly, a rodent's trajectory through adjacent place fields, given phase precession, naturally leads to the encoding of spatial trajectories within individual theta cycles. This temporal compression from behavioral to theta-timescales is thought to allow the encoding of the spatial trajectories via STDP, for instance in area CA3[42,61].

The present study proposes another function of phase precession, which may operate in particular where codes are required to be

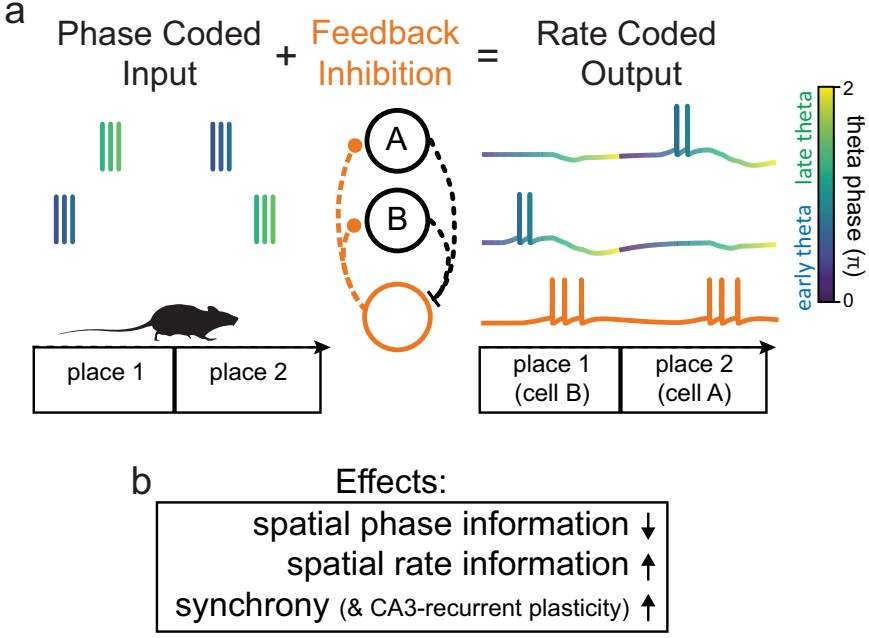

**Fig. 7 | Schematic summary of phase-to-rate recoding. a** Schematic of phase-to-rate recoding mechanism. A phase-coded input (mean rates, but not phases, of inputs are identical across space) coincides with feedback inhibition leading to a rate-coded output (only cell B fires in place 1 and only cell A fires in place 2). **b** Summary of effects of phase-to-rate recoding in the dentate gyrus–CA3 system.

sparse and synchronous. It is noteworthy that the two functions may compete at the local level, since the inhibition of late-theta spikes should impede trajectory encoding. Accordingly, our results suggest that DG is ill suited to trajectory encoding, due to its powerful feedback inhibitory circuits[32]. However, downstream area CA3 will nevertheless simultaneously receive an "intact" theta-phase code directly from EC. How the conjunction of these functionally differentiated inputs in CA3 might affect its operation remains to be explored.

More generally, the question arises if feedback inhibition necessarily entails a partial degradation of phase codes. This "trade-off scenario" predicts that areas with an intact phase-code should have little, or weak feedback inhibition. Interestingly, Tingley and Buzsaki[40] indeed report the efficient encoding of position through phase- but not rate-codes in lateral septum, where no local inhibitory circuit is documented[62] (note that other studies do report spatial rate coding in lateral septum[63]). According to the trade-off scenario, temporal codes in brain areas with local feedback inhibitory circuits[9,10,12], exist despite feedback inhibition. Feedback inhibition may for instance be sufficiently weak to preclude a full phasic inhibition, but to the degree that it acts, will nevertheless impair phase coding. Our analysis of early-theta phase to rate recoding supports this notion in the DG, since it showed moderate phase-code degradation even in the absence of full phasic inhibition (Supplementary Fig. 11). However, this does not imply that phase codes are necessarily degraded by feedback inhibition. Specifically, feedback inhibition should only interfere with phase codes if the time-course of the former is matched to the time-course of the latter. If, for instance, time-codes exist on much longer timescales than inhibition then inhibition could sparsify activity without much impacting temporal information. If on the other hand the millisecond timing of spikes carries information, while feedback inhibition only unfolds on slower time scales, then sparsification may again occur without much impact on time codes. The ultimate impact of feedback inhibition on rate and phase codes will depend on numerous factors, including the precise spatial and temporal properties of inhibition, the overall strength of inhibition, and the type and oscillatory structure of incoming temporal codes.

A prediction of our model in vivo is that the theta phase-preference distribution of GCs should be narrower than for grid cells. Indeed, already Skaggs et al.[9] noted a characteristic "lack of activity at the end of the theta-cycle" of GCs in vivo. Additional work[11] allows a direct comparison between the theta phase distributions in DG and EC, and overall also suggests a more constricted range of phase preferences in DG than EC (Supplementary Fig. 3; see below for some caveats). Interestingly, these in vivo data display another pattern, which appeared highly counterintuitive to its discoverers: The mean phase vector of DG (the output area) precedes that of EC layer 2 (the input area) within the theta cycle (at least assuming plausible synaptic delays of <50 ms). The EC activity-peak is in turn followed by a conspicuous silence in DG activity. This highly counterintuitive pattern precisely matches the emergent pattern in our model, and reveals powerful DG feedback inhibition as a potential explanation (Fig. 2a). It should be noted that differences in EC cell types and projections provide an additional, though not mutually exclusive, explanation[33,64,65] (Supplementary Fig. 3c).

Nevertheless, it must be emphasized that our model reflects a dramatic simplification compared to the in vivo circuit. For instance, in vivo GCs receive numerous other inputs which may or may not themselves contain time codes. Indeed, while there are some discrepancies in the literature[66–68], recent research suggests that perhaps only 25% of DG-projecting EC stellate cells are grid cells[33]. We presently chose to nevertheless focus our analysis on these inputs, because only they allowed to specify a well-defined time code (addressing the role of additional inputs in various robustness analyses).

These caveats may explain a number of differences between in vivo data and our model. For instance, Mizuseki et al.[11] find that (i) in vivo, EC cells are more strongly theta-modulated than DG cells and that (ii) phase precession appears at least as pronounced in DG as in EC (a finding potentially at odds with the presently suggested degradation of phase information in DG). Relatedly, (iii) the DG cells in our model appear more strongly theta-modulated than in vivo[11,69]. Interestingly, concerning (i), recent research suggests that many strongly theta-modulated EC cells recorded in Mizuseki et al.[11] may not have been grid

cells[33] and may not project to DG[65] (they may be pyramidal cells). Moreover, concerning (ii), it remains unclear to which degree the DG cells recorded in Mizuseki et al.[11] reflected mossy cells, rather than GCs, since extracellular recordings are known to be biased towards the more active mossy cells[69,70]. Finally, the less stringent theta-modulation of GCs in vivo (iii) suggests the impact of additional inputs.

Other features omitted in the present model include MF-CA3 LTP and short term depression at PP synapses. While we have performed a number of analyses suggesting considerable robustness to a variety of factors (Supplementary Figs. 5–7 and 11–13), future research must show in how far the presently proposed computation might be enhanced, impaired or altered by factors not yet considered.

In conclusion, our results suggest a translation of phase to rate codes in the EC-DG system by local feedback inhibitory circuits. The consequence is a sparse, synchronized population-rate code, with increased rate information content. All three properties, sparsity, synchrony and information-content, may support the associative storage of non-overlapping engrams in downstream area CA3. The essential ingredients for phase-to-rate recoding are (i) phase-coded inputs and (ii) local feedback inhibitory microcircuits. Notably, these ingredients are widespread in the mammalian CNS. Temporal coding patterns such as phase precession are being discovered in more and more brain structures[12–14] and local feedback circuits are ubiquitous. Our results thus raise the question whether phase-to-rate recoding may be a conserved computational motif occurring in many brain areas where the two ingredients coincide. Indeed, there is no reason why the computational motif should not similarly work for other temporal coding schemes, as long as feedback circuits systematically leverage temporal information to improve population-rate codes. The latter could support the formation of precise auto-associative attractor networks across the brain. More generally, this work also raises the question which other recoding mechanisms may exist, that maintain information at subsequent stages of processing, by redistributing information among different coding schemes[40].

## Methods
### Simulation of theta-phase-precession in EC grid cells

To simulate grid cell firing, we combined a grid cell model by Solstad et al.[28] and a phase precession model by Bush and Burgess[29,71]. To create a grid cell firing function $g_w(x,y)$, where $\mathbf{l} = [x, y]$ are spatial coordinates, the sum of three sinusoidal gratings specified by their wavevectors $\mathbf{k}$, with 60° and 120° angular differences were used[28] (Supplementary Fig. 1a), according to Eq. (1):

$$g_w(x,y) = \frac{2}{3}\left(\frac{1}{3}\sum_{i=1}^{3}\cos(\mathbf{k}_i(\mathbf{l}-\mathbf{l}_0)) + \frac{1}{2}\right) \quad (1)$$

Two hundred grid cells, with empirically matched parameters[27], were created ($\mathbf{k}_i$ computed as in Solstad et al.[28], based on (i) spacing, (ii) orientation and (iii) spatial offset): Spacing drawn from a skewed normal distribution between 15 and 120 cm with the median of 43 cm; orientation drawn from uniform distribution between 0 and 60°; and spatial offset drawn from a uniform distribution between 0 and 100 cm in both $x$ and $y$ directions. The random number generators were seeded for each grid cell population to allow data reproducibility. Throughout this manuscript, a "grid seed" corresponding to one grid cell population (or mouse) will be the unit for statistical comparison. We then simulated a mouse traveling on a linear trajectory at 20 cm/s for 2 s. We modeled straight parallel trajectories at variable distances (0.5, 1, 1.5, 2, 2.5, 3, 4, 5, 6, 7, 8, 9, 10, 15, 45, 60 cm; Supplementary Fig. 1b).

To model phase precession, we adapted an approach using the linear distance of the mouse to the closest grid vertex and the direction of travel[29], combining it with the generated grid profiles[28]. To this end, we used the formula previously used to obtain $g_w(x,y)$ to also calculate the relative linear distance to the closest grid vertex d$(x,y)$ using Eq. (2).

$$d(x,y) = \arccos\left(\frac{3}{2} \times \frac{g_w(x,y)}{g_w^{max}} - \frac{1}{2}\right) \times \left(\frac{\lambda\sqrt{6}}{4\pi}\right) \quad (2)$$

where $g_w^{max}$ is the maximal firing rate of grid fields, and $\lambda$ is the spacing of a given grid cell (see above).

A virtual trajectory at 20 cm/s now corresponds to a mapping of time ($t$) to position ($x,y$), such that we could calculate a spatial rate profile $r_G(t)$ from $g_w(x,y)$ and d$(t)$ from d$(x,y)$ for each grid cell over the course of the trajectory. To make phases precess monotonically during the traversal of a grid field, d$(t)$ was multiplied by −1 if the mouse was leaving a grid field.

This allowed us to calculate phase precession patterns within each grid field[29]. Briefly, the relative linear distance d$(t)$ was first transformed into a preferred theta phase φ$(t)$, according to Eq. (3).

$$\varphi(t) = k_1\pi\left(\frac{d(t)}{\lambda} + 0.5\right) \quad (3)$$

and then used to calculate a phase-code function $r_\varphi(t,\theta)$ using Eq. (4).

$$r_\varphi(t,\theta) = \exp(\cos(\theta(t) - \varphi(t)) \times k_2) \quad (4)$$

where $k_1 = 1$ and $k_2 = 1.5$ are constants which we chose to match experimental data[10,71] and $\theta(t)$ was an assumed global theta oscillation at 10 Hz. Note, that in Bush and Burgess[29] cells display phase precession but the overall firing probability stays constant, i.e., firing probability does not increase towards the middle of a field, whereas in the present model firing probability increases towards the middle of a field.

Finally, the phase precessing spike rate profile ($r_{\varphi,G}$) for each grid cell was generated according to Eq. (5).

$$r_{\varphi,G} = r_\varphi(t,\theta) \times r_G(t) \quad (5)$$

scaled to obtain instantaneous firing rates consistent with experimental observations[10]. Individual spike trains were generated by using an inhomogeneous Poisson process based on the calculated spike probability profiles (Elephant package[72]). To produce the phase precession plots shown in Fig. 1 individual trajectories were simulated with random Poisson seeds 1000 times.

### Isolation of phase- and rate-code

To isolate phase- and rate-code in EC or DG, we defined a separable coding scheme based on the mean rate and phase within individual theta cycles. An arising difficulty is that in theta cycles where a cell does not fire the phase-code is undefined. Since any arbitrary substitution of, e.g., phase = 0 for such undefined cycles impacts both correlation measures and perceptron learning in non-trivial ways, we chose to first transform the phase-rate code from polar to Cartesian coordinates. Specifically, where the original phase defined the angle, and the rate the length, of a vector (polar coordinates) we calculated the corresponding $x$ and $y$ coordinates (Cartesian coordinates), which could be readily learned by the perceptron (Fig. 1j). To isolate phase information, we held firing rate (i.e., vector length) constant (1 Hz), effectively eliminating the information contained in differential firing rates across grid cells and space (other constants will rescale perceptron inputs, but cannot reintroduce information contained in differential firing rates). To isolate rate-information we held mean firing phase (i.e., vector angle) constant at $\pi/4$ (this ensured that rate information was equally distributed across both Cartesian coordinates, but phase information played no role). Subsequent analyses (Pearson's correlation, spatial information, perceptron learning) were then performed

on the flattened [time-bin × cell × Cartesian coordinate] arrays (20 time-bins of 100 ms each, 200 or 2000 cells, 2 Cartesian coordinates).

### EC phase shuffling

In order to remove the information contained in theta phase precession in a minimally disruptive way, we devised a per-cell and per-theta-cycle shuffling procedure, which conserves the overall theta-phase distribution. Specifically, we recorded the emergent theta-phase distribution of EC cells in our model (Fig. 2a, gray histograms), and then used it as an inhomogeneous Poisson template, to redraw EC spike times within individual theta cycles of individual cells (Fig. 2b, gray histograms). This fully conserves which cells fire in which theta cycles, perturbing only the precise timing of the spikes. It also conserves the overall theta-phase distribution, which is important to avoid unspecific effects deriving from the unequal recruitment of downstream cells due to mere threshholding.

### GC membrane noise and LEC inputs

To ensure our results are robust when additional GC inputs are active we simulated either GC membrane noise (modeling completely random inputs) or LEC inputs (modeling a partially informative contextual input), each with increasing strengths. GC membrane noise was simulated through a continuous varying current injection at the soma of each GC. Instantaneous noise current was drawn from a normal distribution of mean 0 and different standard deviations (50, 250, 500 pA). As baseline standard deviation, we chose 50 pA, qualitatively matching in vivo recordings[37].

LEC inputs were created from an inhomogeneous Poisson process, to create a counter-cyclical theta modulation, but with no further spatial or phase information (Supplementary Fig. 6). They were randomly connected to the GC population at the distal dendrites, but with at most one LEC input per GC. The counter-cyclical modulation profile was generated by smoothing a saw-tooth wave (between 0 and 30 Hz) to match Deshmukh et al.[38]. To simulate a situation with "similar" contextual input we assume the same ensemble of 20 LEC cells was active, but with random spike trains (different Poisson seeds). To increase their relative impact on DG, we systematically increased the number of output synapses per LEC cell (Supplementary Fig. 6: 100 in B, 150 in C and 200 in D). To simulate a situation with "identical" contextual inputs (i.e., containing rate and phase information, Supplementary Fig. 7) we let the same ensemble of 20 LEC cells repeat the same spike train (same Poisson seed).

### Spatial information

To calculate spatial (rate) information ($I_S$) we used the standard spatial information measure (Skaggs et al.[24]). Briefly, $I_S$ in bits/spike was calculated for each cell according to Eq. (6).

$$I_s = \sum_s^S \left( p \frac{r_s}{r} \log_2 \left( \frac{r_s}{r} \right) \right) \tag{6}$$

where $s$ indexes spatial bins, $p$ is the occupancy of a bin, $r_s$ is the mean firing rate within the bin, and $r$ is the mean rate of the cell over all bins. Note, that as we consider linear trajectories and uniform speed, a spatial bin is simply a distance and occupancy is always the same. To avoid spuriously high information values due to sparsity, we used 5 cm spatial bins, aggregated spikes over 20 Poisson seeds, and included only cells with at least 8 spikes overall. Total information per spike was then calculated either on a per cell basis or by averaging over cells.

### Positional Information

To calculate positional (rate or phase) information we used the information measure introduced by Olypher et al.[39], as adapted by Tingley and Buzsaki[40]. This measure allows to measure phase information arising from phase precession, given multiple uni-directional traversals of the same field (e.g., on a circular track)[40], since in such a setup positions will be associated with specific phases across trials. $I_{pos}(x_i)$ was calculated for each position and cell according to Eq. (7).

$$I_{pos}(x_i) = \sum_{K \geq 0} \left( P_{k|x_i} \log_2 \left( \frac{P_{k|x_i}}{P_k} \right) \right) \tag{7}$$

where $P_{k|xi}$ is the probability of a specific phase (or rate) $k$ at a particular position $x_i$ (assessed over trials or, in our case, Poisson seeds) while $P_k$ is the probability of $k$ over all positions. To calculate this, rate or phase values for each cell and position bin were discretized in to 7 data bins[40] and then smoothed with a box filter over progressively larger sets of position bins (1–20 bins corresponding to 2–40 cm). As above, cells with <8 spikes over all 20 Poisson seeds were excluded from analysis. For the phase code the circular mean was used and bins without spikes (NaNs) were ignored. To obtain information per spike, data were divided by the mean spike rate and to obtain a summary measure, data was averaged across all positions and cells.

### DG model

The biophysically realistic model of DG (pydentate) was adapted from Santhakumar et al.[73] as previously described in Braganza et al.[32]. Briefly, we model 2000 GCs, 24 BCs, 24 HCs and 60 MCs, all with biophysically realistic membrane conductances, and experimentally calibrated synaptic properties. Specifically, the spatiotemporal properties of net-feedback inhibition delivered to GCs was precisely tuned to match experimental observations. The term "net-feedback inhibition" emphasizes that the inhibition arriving at GCs derives from multiple interconnected interneuron types within the inhibitory microcircuit (for an overview see Supplementary Table 1 in ref. 32). We connected every EC grid cell to 100 randomly chosen GCs and 1 BC, adjusting the synaptic weight to obtain plausible activity levels. Specifically, the PP weight was decreased from 1 to 0.9 nS in order to obtain an activity of around 2% of GCs per 100 ms theta cycle[32,37,74], corresponding to 0.2–0.3 Hz. DG circuit interventions were performed as in Braganza et al.[32], by setting the appropriate subset of synaptic weights to 0. For GC rate-controlled data in the DG circuit interventions (e.g., Fig. 2i), PP weights were reduced further such that GC rate stayed within the range (0.2–0.3 Hz) independent of which or whether inhibitory circuits were present (see Supplementary Fig. 4).

### Perceptron

A perceptron model was trained to compare the decodability of both rate- and phase-code. The perceptron was implemented using PyTorch[75]. At the output two sigmoid units were used to distinguish two trajectories ([1,0] and [0,1]). The input to the perceptron was the vectorized cells-by-bins matrix containing either rate or phase information (Supplementary Fig. 1e). To allow statistical comparison, the perceptron was trained on EC and GC data given 10 distinct grid cell populations (corresponding to 10 virtual animals, i.e., samples). For each grid seed the perceptron was trained to classify two trajectories with varying distance (see above), where a trajectory was presented in the form of 20 random spike trains (independent inhomogeneous Poisson processes, given $r_{\varphi,G}$), or more specifically, the 20 resulting phase-rate vectors. Perceptron weights were adjusted via stochastic gradient descent, for 10,000 epochs, during each of which all 40 phase-rate vectors of a trajectory pair were presented. Learning rate had to be adjusted between conditions due to different population sparsity (within conditions it was constant at either $10^{-3}$ or $10^{-4}$). The loss function used was the Root Mean Square Error (MSE). Decodability was then quantified as $1/N_E$, where $N_E$ is the number of epochs until MSE reached the threshold of 0.2 (as in Cayco-Gajic et al.[25]).

## Tempotron

To investigate the decodability of complete spike trains in DG without making assumptions about phase and rate coding windows, we used the Tempotron as described in Gütig and Sompolinski[26].

We trained the tempotron for 200 epochs on the granule cell spike patterns (2000 GCs, 2 s) to distinguish trajectories 15 cm apart. Other tempotron parameters were $V_{rest} = 0$ mV, $\tau = 10$ ms, $\tau_s = 2$, learning rate = $10^{-3}$. The classification threshold was set for the spiking patterns of each grid seed by calculating the maximum output for each pattern and averaging across the maxima. To quantify the learning speed, we calculated the loss at each epoch as the sum of absolute differences between the maximum output and the threshold, for each incorrectly classified spike pattern. We then fit an exponential decay function to the loss across epochs and defined the learning speed of a grid seed as $\frac{1}{\tau_{decay}}$. We found that this procedure, while analogous to the threshold approach taken for the perceptron, led to less noisy results. Our tempotron implementation can be found at https://github.com/danielmk/tempotron and is based on an implementation by Dieuwke Hupkes.

## CA3 model

To investigate the potential consequences of phase-to-rate recoding on CA3 plasticity, we created a minimal CA3 model (minimal CA3 model) within the Brian2 simulation environment[76], adapting an STDP mechanism from Goodman and Brette[45] based on Song et al.[46]. Specifically, we created a CA3 pyramidal cell network of 600 recurrently connected leaky integrate and fire (LIF) cells[45],

defined by Eq. (8).

$$\frac{dv}{dt} = \frac{E_l - v + v_{syn}}{\tau_m} \qquad (8)$$

where $v$ is membrane voltage and all other parameters are as in Table 1. This $Pyr_{CA3}$ population was then randomly connected to the 2000 GC afferents with probability $p = 0.35$, leading to ~70 GCs per CA3 cell and 21 CA3 cells per GC[47]. We additionally modeled a population of 60 feedforward interneurons ($IN_{CA3}$), one of the most prominent features of the GC-CA3 connection[48], according to Eq. (5), but with a refractory period of 5 ms to prevent firing rates >200 Hz. This is important, given the likely role of the disynaptic latency in the feedforward-circuit on CA3 input synchrony detection. Inhibitory potentials were modeled with a latency of 5 ms to match the experimentally determined range of 2–7 ms[77]. STDP was modeled as in Goodman and Brette[45], but was made symmetric to match experimental findings in CA3[42]. If not otherwise specified, the STDP time-window was 20 ms. To assess the efficacy of STDP, all recurrent CA3 synapses were initialized with a weight of zero, allowing to simply monitor the mean final weights as a measure of total plasticity. In this minimal model CA3 spiking is exclusively driven by GCs and not affected by recurrent CA3 inputs in order to isolate the plasticity effects of DG inputs.

Parameter values were closely constrained by experimental data (see references in Tables 1 and 2). Briefly, electrophysiological properties were taken to be the mean values for the respective cell types from neuroelectro.org (Table 1). Synaptic connectivity values were aggregated from the literature (Table 2). Large postsynaptic effects of GCs were set to match the empirically observed ability of even single GC discharges to sometimes elicit APs as well as disynaptic inhibition in CA3 pyramidal cells. To probe sensitivity to plausible STDP windows we probed STDP time constants up to 50 ms. To probe sensitivity to feedforward inhibitory strength, we probed values from 0 mV (no inhibition) to −10 mV (where $Pyr_{CA3}$ activity is almost completely suppressed). For parameter sweeps (Fig. 6 g, h, k, l) individual simulations were run on 4 grid-seeds and 4 Poisson seeds, and only mean values are plotted.

To check robustness for a more realistic and more complex CA3 model (extended CA3 model), we made three alterations (parameters specific to the extended model in Tables 1 and 2 are cursive). First, we replaced the original voltage-based for a current-based synapse modeled as shown in Eq. (9).

$$\frac{dv}{dt} = \frac{E_l - v + R_{in}(I_e - I_i)}{\tau_m} \qquad (9)$$

where $R_{in}$ is the input resistance and $I_e$ and $I_i$ are excitatory and inhibitory synaptic currents, which are in turn defined by their amplitude $\Delta I$ and synaptic decay time constant $\tau_S$ (see Table 2)[78]. Second,

### Table 1 | Properties of CA3 leaky integrate and fire cells

| Cells | | | References |
|---|---|---|---|
| $Pyr_{CA3}$ | | | |
| $N$ | 600 | Number of cells | CA3 pyramidal cell from neuroelectro.org |
| $\tau_m$ | 45 ms | Membrane time constant | |
| $v_{thr}$ | −48 mV | AP threshold | |
| $v_r$ | −54 mV | Reset voltage after AP | |
| $E_l$ | −67 mV | Resting potential | |
| $R_{in}$ | 150 MΩ | Input resistance | |
| $IN_{CA3}$ | | | |
| $N$ | 60 | Number of cells | CA3 basket cell from neuroelectro.org |
| $\tau_m$ | 14 ms | membrane constant | |
| $v_{thr}$ | −37 mV | AP threshold | |
| $v_r$ | −54 mV | Reset voltage after AP | |
| $E_l$ | −52 mV | Resting potential | |
| $R_{in}$ | 112 MΩ | Input resistance | |

Italic entries refer to the extended CA3 model (Supplementary Fig. 13).

### Table 2 | Synaptic connectivity and properties within the CA3 model

| Synapses | | | References |
|---|---|---|---|
| | Connection prob. | Postsynaptic effect | |
| $Pyr_{CA3}$ => $Pyr_{CA3}$ | $p = 0.01$ | $v_{syn} = 0$ or $\Delta I = w \times 40$ pA, $\tau_S = 15$ ms | 85 |
| GC => $Pyr_{CA3}$ | $p = 0.35$ | $v_{syn} = 15$ mV or $I_e = (TM)$ pA | 47,78,86 |
| GC => $IN_{CA3}$ | $p = 0.7$ (40–50 $IN_{CA3}$ per GC) | $v_{syn} = 5$ mV or $I_e = 40$ pA, $\tau_S = 15$ ms | 48,78,86 |
| $IN_{CA3}$ => $Pyr_{CA3}$ | $p = 0.1$ | $v_{syn} = -0$ to −5 mV (default = −2 mV) or $I_i = 0$ to 80 pA (default 40 pA) | 77 |
| Spike-based homeostasis | All input synapses of a Pyr. cell | $\Delta w = 0.05$ | Brian2 docs: synaptic scaling |

"TM" refers to the facilitating Tsodyks–Markram synapse. Italic entries refer to the extended CA3 model (Supplementary Fig. 13).

we added mossy fiber facilitation modeled by a Tsodyks–Markram (TM) synapse to match Toth et al.[79] namely $\tau_{inact} = 30$ ms, $\Delta I_{max} = 5$ nA, release_fraction = 0.03, $\tau_{recovery}$ 130 ms, and $\tau_{facilitation} = 530$ ms. Third, we added the recurrent excitatory drive at CA3-CA3 pyramidal synapses, which was determined by the weights resulting from STDP. Note that, since symmetric CA3 STDP[42] can only increase synaptic weights (unlike the more typical asymmetric STDP), this setup quickly leads to runaway excitation and weight-increases. In order to avoid biologically implausible runaway weight increases, we additionally added a simple cell-level synaptic scaling mechanism[52], which decrements all incoming synaptic weights of a cell by a factor of 0.05 for each outgoing action potential. Such a scaling mechanism is plausible for CA3 pyramids[80], but of course also affects mean weight increases.

## Statistical analysis

Statistical analysis was performed in GraphPad Prism 4 or 7 and the full details are given in Supplementary Tables S1–S14. Briefly, we performed *t*-tests or ANOVAs as appropriate. In the case of an ANOVA, asterisks indicate significance in the post-test, given that the overall ANOVA was significant. The unit of statistical comparison was always a grid seed (modeling a virtual animal, with a new random network anatomy). For analyses in Figs. 1–3, we were only interested in very large effects, and thus chose a sample size of 10. From Fig. 4, i.e., the tempotron, we increased sample size to 30, since tempotron classification is known to be noisy. For GC rate-controlled analyses network seeds where GC rate was outside the permissible range of 0.2–0.3 Hz were excluded. Box plots show the median, the interquartile range (box) and the data range w/o Tukey-outliers (whiskers), where Tukey-outliers are defined as exceeding the 1.5× interquartile range.

## Reporting summary

Further information on research design is available in the Nature Portfolio Reporting Summary linked to this article.

## Data availability

The pre-simulated spike data generated in this study have been deposited in zenodo under accession code https://doi.org/10.5281/zenodo.8280121 (https://doi.org/10.5281/zenodo.8280121). The processed data underlying the figures are provided with this paper as Source data file. Source data are provided with this paper.

## Code availability

All code required to reproduce the present data figures is available at https://doi.org/10.5281/zenodo.8278084 or https://github.com/barisckuru/phase-to-rate. Our models relied on Python 3.9.12 and standard software packages, namely NumPy 1.22.4[81], elephant 0.11.1[72], neo 0.10.2[82], seaborn 0.11.2[83], PyTorch 1.12.1[75], as well as the NEURON 8.0.0-2-g0e9a0517+[84], and Brian2 2.5[76] modeling environments.

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

## Acknowledgements

We thank Tatjana Tchumatchenko for helpful comments on the manuscript. This study was funded with the help of BonFor 2019-1B-01 to O.B. as well as DFG SPP 2041 and SFB 1089 Project C4 to H.B.

## Author contributions

O.B. and H.B. devised the study. B.K., D.M.-K., and O.B. performed the simulations and analyzed the data. All authors contributed to writing the manuscript.

## Funding

## Competing interests

The authors declare no competing interests.
