## [Peer Review File · Nature Communications]

Phase information is conserved in sparse, synchronous population-rate-codes via phase-to-rate recodingREVIEWER COMMENTS

Reviewer #1 (Remarks to the Author):

Kuru and colleagues take on an intriguing aspect of neuronal coding, namely the handling of phase and rate coded information by different circuits that may have a differential ability to represent/handle phase and rate information. They use 3 distinct and independent circuit models of entorhinal cortex (EC), Dentate Gyrus (DG) and CA3 to primarily examine (1) how location and direction, coded as rate and phase information, from the EC model is transformed by the DG circuit, specifically by feedforward and feedback inhibition, and (2) how the DG model output is processed by a CA3 model which includes inhibition. Starting with the EC, the authors simulate grid cells by simulating animals moving at various trajectories. These spiketrains encode location and direction information as rate, phase, and phase precession. These neural codes are then parsed split into discrete rate and phase information to be utilized as inputs into the DG model. Rather curiously, the processing of the combined information and whether it has an advantage over the individual modalities is not explicitly tested. By using identical seeds for each simulated mouse, they ensure that spike train variability arises from the simulated trajectories travelled. The grid cell spike trains generated have rate information, phase information on direction, as well as phase precession information. They evaluate how information in EC output spike trains is processed by the DG using 3 approaches (1) an assessment of information content to preferentially assess rate coding (2) using “decodability” as number of trials needed for a machine learning perceptron classifier to learn to discriminate input patterns and (3) using a tempotron ML classifier which is designed to process sequential timing information. The simulations are carefully constructed and the use of multiple complementary approaches to assess EC to DG transformation of phase and rate information is a strength. The authors conclude that DG feedback inhibition is necessary and sufficient to convert EC phase code to rate code while maintaining spatial information. By feeding the DG output to a CA3 model, they show that compression of the EC phase into a rate code occupying half of the theta cycle in the DG enhances downstream CA3 plasticity (STDP). The results and approach are interesting and, if they hold up, will provide novel insights into DG information processing. However, there are some conceptual concerns that need to be addressed.

Specific Comments:

1. While each individual model is firmly based in prior studies, the manuscript has limited information about the original models, especially the EC and phase precession models and how these were integrated. For instance, the equation defining $g_w(x,y)$ is not included forcing the readers to look up original papers. Better illustration of the phase precession as part of this model in supplemental figures would be useful. Illustration of how EC phase information components (phase of trajectory and phase precession) are developed would be helpful. Also, it is confusing to introduce the DG model before the EC in fig 1. Illustrating the EC model and its output before the DG model is recommended.

2. The polar to cartesian transformation illustrated in Fig 1H does not isolate rate from phase information as the transformation of the vector based on rate and phase would inherently retain both modalities. Thus, the assumption that x is rate and y is phase on the transformed vector is flawed. If this is true, the analysis in Fig I & J are not meaningful. The authors do introduce a more valid vector to cartesian transformation in the methods for the perceptron model in Fig 3. If the same strategy was used in fig 1 and 2 it needs to be introduced upfront. If not, the analysis and interpretation of Fig 1 I & J need to be reconsidered.

3. On a related note, why are there negative R_{in} valued in Fig 1I and 2E, what does this mean and why does it output a positive R_{out} ? Similarly, what happens when the distance between parallel trajectories are the same ($R_{in} = 0$)? Fig 1I shows $R_{in} = 0$ yields $R_{out} > 0$.

4. The authors need to explain why the DG R_{in} phase is low? Is this a function of EC grids and/or trajectories simulated being highly variable. Are a broader range of input variabilities, including more highly correlated inputs, expected in vivo?

5. In processing of EC phase information in DG, can the authors discuss whether DG feedback inhibition reduces EC phase information as suggested in line 165 or could it compress it to half a cycle? Considering this aspect in relation to pattern separation of phase code, Fig 1J implies more separability of phase information in dentate output (assuming point #2 is not an issue) than in the EC output, is that possible if phase information is degraded?

6. The relevance of the shuffled data, where EC phase information is retained but phase precession information is removed (Fig 2E&F), is not fully flushed out. The conclusion from this analysis is that lack of phase component decreases pattern separation in the rate domain – but this is only the spike precession component of phase code? Removal of this information reduces “phase code R_{in} ” to zero. Does that imply that only phase precession contributes to differences/decorrelation in phase component of EC firing? Wouldn't the broad theta distribution of spikes contribute to decorrelation of EC spike train output in the phase domain? In this regard, the independent assessment of information content reveals that removing phase precession reduces information content. The perceptron data (3C) indicate that there is phase information beyond the precession in the EC spike trains. This needs to be clarified/discussed.

7. Since perceptron learning speed improves with both decorrelation and information content, and shuffling increases decorrelation (Fig 2G) it is difficult to assess whether the differences in “decodability” relate to information content (as proposed) or merely reflects decorrelation. This confound needs to be addressed. Moreover, while learning speed was shown to correlate with input decorrelation (Cayco-Gajic et al., 2017) the authors need to justify the use of learning speed as a metric to assess the

information content of the input rather than just a reflection of the classifier performance on a particular class of inputs.

8. In the discussion, the authors suggest that phase information in EC input may be converted to a “population” code in DG with distinct granule cell subsets responsive to early and late phase EC activity. Is there any evidence for this in the simulation data? Also, how does this comport with the compression of data within half the theta cycle in Fig. 2.

9. The statement in the abstract “through computational modeling of the entorhinal cortex (EC)-dentate gyrus (DG)- CA3 system” is misleading as the study adopts 3 models which are not actively synced rather output from each model is used as input to the subsequent model. This needs to be clarified.

Overall, the fact that shuffling to remove phase precession reduces decodability of rate information content using both perceptron and tempotron, even when adjusted for rate supports the conclusion that some phase information is transformed to rate information by DG feedback inhibition. Clarifying specific issues raise above would strengthen the study.

Minor:

- Line 31: “This challenges is compounded” → “This challenge is compounded when...”
- Line 38 - 41: “For instance, the precise...and its main input area EC” should be reworded
- Line 64: “ ‘pattern completion’ ” → “ ‘pattern completion’ ”
- Line 249: “measures” → “measured”
- Line 258 / Line 634: loss uses MSE in the caption but RMSE in methods – it’s the same but for consistency
- Line 289: “Fig3C and D” → “Fig. 3C and D”
- Figure 5B: Y-axis formatting is overlapping
- Line 589 Eqn4: $\log_2(r/r)$ – numerator not defined. Is it r_s/r ?
- Line 656: “, where v is membrane...” → delete comma

Reviewer #2 (Remarks to the Author):

In this study, Kuru et al investigate how the common circuit motif of feedback inhibition interacts with phase coding, using simulations based on entorhinal-cortex (EC) to dentate gyrus circuits. This is an interesting and important question as phase coding is a widely observed phenomenon in the brain, and feedback inhibition is a ubiquitous feature of brain circuits. The choice to focus on the entorhinal-dentate gyrus system is sensible as the circuit is well characterized, a lot is known about information coding in this system *in vivo*, and it is widely thought that feedback inhibition in this circuit is performing a well-defined computational function of pattern separation – i.e. decorrelating similar input patterns while preserving their information content. The authors have previously worked on characterizing feedback inhibition in this circuit using a combination of experimental and computational methods so are well placed to address this question.

Based on their analysis of simulated data the authors conclude that when the dentate circuit with feedback inhibition receives phase coded input from EC, information contained in the phase of input spiking is recoded into spatial patterns of firing rate in the output, a process they term phase-to-rate recoding. This occurs because dentate granule cells that are driven to spike by input arriving early in the theta cycle recruit strong feedback inhibition that suppresses firing later in the cycle. Therefore it is not simply which granule cells are receiving input but also when that input occurs in the theta cycle that determines the spatial pattern of activity in the granule cell population. A consequence of these dynamics is that the range of theta phase over which the granule cells spike is much narrower than that over which the EC input activity occurs.

Though the study is well motivated, there are two issues which would need to be addressed convincingly to make a strong case that the conclusions are relevant to the real brain system.

The first concerns the nature of the code in the entorhinal-hippocampal system and what function phase precession is actually serving. Phase precession in the hippocampus is thought to enable each theta cycle to encode a short trajectory through space, rather than a single location, with cells firing early in the theta cycle representing the start of the trajectory and those firing late in the cycle the end of the trajectory. Though phase precession in EC is less well characterized, the model of phase precession on which the authors base their simulated EC activity (Bush and Burgess 2020) also assumes such trajectory coding, and shows that the direction of motion can be decoded from simulated EC activity by decoding a series of locations from activity at different phases and fitting a vector through them. The encoding of trajectories by multiplexing rate codes for different locations into different phases of the theta cycle appears fundamental to information processing in this system. However it is largely ignored in the current work, where analyses assume that each theta cycle represents a single position. This is problematic because the proposed phase-to-rate recoding mechanism appears likely to severely degrade the ability of the system to encode trajectories. This is because granule cells recruited early in the theta cycle by input representing the start of the trajectory, will recruit feedback inhibition

preventing input later in the theta cycle, representing later steps of the trajectory, from driving spiking. I feel that given the importance of trajectory encoding by sequences of activity across the theta cycle in the real system, to address the effect of feedback inhibition on coding in this system requires characterizing how it affects such codes.

The second issue is whether the proposed mechanism is consistent with in vivo activity. A strong prediction of the work is that the output spiking of dentate granule cells should be concentrated into a much narrower range of theta phase than the input received from EC, and should show much less phase precession. Whether this is in fact the case is therefore critical to determining whether the simulations are likely an accurate reflection of what is happening in the real system. The only discussion of this I could find in the manuscript was:

"As expected, the broad phase tuning of grid cells ... led to EC phases well distributed across the full theta cycle. By contrast, in the intact DG circuit GC spiking was concentrated within the first half of the theta cycle. ... This appears to be consistent with experimental data (Mizuseki et al., 2009)."

From my reading of Mizuseki et al it is not at all clear that the in vivo data are consistent with the simulations. For example Mizuseki figure 3D shows that EC layer 2 (EC2) spiking (which provides the input to the dentate gyrus) is overall more strongly modulated by the theta rhythm than dentate gyrus (DG), while Figure 3D shows that the phase of maximum spiking probability is narrower in EC2 than in DG. Figures 6 and 7 show that, if anything, phase precession is stronger in DG than in EC2, and show an example DG neuron which phase precesses across nearly the entire theta cycle. These data appear inconsistent with the prediction from the simulations that the output of DG is more tightly synchronized, and constrained to a narrower range of theta phase, than the input received from EC. The authors need to engage much more seriously with the question of whether their simulations are in fact consistent with in vivo data.

Other points:

Though the manuscript was generally clearly written, there was one aspect of the analysis that would benefit from clearer explanation. Figure 1I and J look at how the network pattern separates inputs separately for rate and phase codes, but important information about what is done differently in figures I and J to isolate rate and phase coding seems to be omitted. The accompanying text (lines 113-120) indicates that the activity of each neuron was first converted into a 'theta-vector' defined by the firing rate and mean phase of the spikes in each theta cycle. These vectors are then converted into X and Y coordinates to deal with cycles where no spikes were emitted (where phase is undefined), and these coordinates used to examine pattern separation for phase and rate codes. However, I could not find information about what was actually done to the X and Y coordinates to extract the 'Phase code' and 'rate code'. I assume that this was done by setting either the phase or firing rate component of the

theta-vectors to a constant value before computing the X and Y coordinates, but it was not clear from the manuscript.

Reviewer #3 (Remarks to the Author):

Summary of the findings:

This is an elegant and well-written study of how a model of the dentate gyrus (DG) processes spatial information coded in the rate or phase of MEC grid cells spiking activity. Using a conductance-based model of DG they previously published (Braganza et al. 2020), the authors first showed that DG performs pattern separation of the EC inputs when considering both the rate-code and the phase-code. Thanks to a clever shuffling procedure, they show that the phase-tuning of input spikes, which contains spatial information, affects the rate code of DG output cells: removing that phase-coded spatial information (but not the rate-coded info) increases granule cells (GCs) firing rate and pattern separation of the spatial rate code, and decreases GCs rate-coded spatial information. Those effects are mediated in large part through the feedback hilar circuit, suggesting that net feedback inhibitory dynamics are important for converting the EC phase-code into a rate-code in DG. A perceptron decoding approach suggests that although DG may still perform some phase-coding, the phase-coded spatial information from EC is partially converted in a rate-code. A tempotron decoding approach further shows that the phase information from the inputs is used by DG and is important for preserving spatial information in the spike times and synchrony of the GC population. Manipulations of the feedback inhibitory circuit suggests that it may mediate, at least in part, the use and conversion of the phase information contained in the inputs. Finally, the authors investigated a potential effect of this information processing on CA3, the network downstream of DG: they report that the improved DG synchrony mediated by the feedback circuit helps recruit CA3 pyramidal cells and promotes long-term potentiation of the CA3 recurrent synapses.

General comment:

This study addresses an interesting and refreshing question (how DG processes phase-coded information from its inputs) and it does it with state-of-the-art modeling techniques and rigor. Some questions asked by the authors are not fully answered and the study could be improved and expanded a bit to become even more interpretable and useful. However, I realize that I made a lot of comments and suggestions in my review. Not all points have the same importance, I'm not expecting the authors to follow all my analyses suggestions and it would be perfectly appropriate to address some concerns simply with textual edits.

Targeted comments:

1-Model validation and realism

Line 60: Theta phase precession in DG principal cells is actually not well documented. Skaggs et al 1996 and Mizuseki & Buzsaki 2009 did not distinguish between GCs and Mossy Cells (MCs), and their conclusions are likely drawn from a majority of MCs because active cells in DG are biased to MCs (cf Goodsmith & Knierim 2017 and Senzai & Buzsaki 2017), plus it's harder to study precession in low-firing neurons like GCs. Nevertheless, these papers could be used to further validate the model in regard to phase coding. How does phase modulation and precession in MCs and GCs of the model compare to those experimental reports? Validating the model using data segregating GCs and MCs would be best, but there is a gap in the literature, as far as I know. However, the data might exist: Senzai & Buzsaki 2017 dataset is available with some mice running in linear tracks I think (<https://dandiarchive.org/dandiset/000003/0.210812.1448>).

Line 99-100 (empirical basis of modeled inputs): The authors assume that the inputs to DG are from grid cells. This is a common assumption in modeling (although see Kim & Royer 2020), but a discussion of the limitation of this approach should at least be included: 1) the data used to constrain the modeled inputs are not guaranteed to come from MEC layer 2 and reach DG, 2) other spatial but non-grid neurons exist in MEC, potentially with phase information. The percentage of such cells in the DG inputs is not well known, 3) LEC is of course a major input ignored. LEC does not have much phase modulation (Deshmukh & Knierim 2010) but it's not obvious how robust the computations investigated here would be when adding inputs with little phase and no spatial information.

The DG model used here was reported in Braganza et al 2020 to include facilitation of feedback inhibition. That's still the case here, correct? What about the short-term dynamics at other synapses, feedforward excitation and inhibition? It feels like PP-GC synaptic dynamics themselves could have large effects on rate or phase coding. Also, the conclusion on the role of the feedback circuit vs feedforward could be biased if short-term dynamics are not included for the feedforward circuit (as those dynamics likely provide more opportunities for interesting emergent computations).

Regarding the robustness of the results, an important question related to any form of temporal coding is whether and how it resists to noise. The impact of neural noise on phase coding in particular could be non-trivial. The determinism of the models used here is another limitation of the study.

2-Activity patterns and pattern separation

Line 103: more details needed on virtual spatial trajectories: linear or random walk? how long? Are arrows in Fig 1E examples of trajectories?

Line 108-120: Need more details (here or in methods) to understand exactly what patterns of activity are considered and how similarity is computed (population vector of binwise rate or phase code with several bins? how many bins?). Line 113 (defining “theta-vectors”) was confusing at first because I was coming with “population vectors” in mind, or time-vectors (where each time bin is a coordinate).

How were correlation between the matrices of population activity computed? (Average across neurons or across time bins or matrices as flattened arrays?)

Showing the trajectories of (rate, phase) theta-points in the 2D cartesian space, for both inputs and outputs, would be helpful. For DG output it should show a bunch of dots at (0,0) and a different cluster of points somewhere else. If phase-coding is not important in output (i.e. staying constant for a given GC and perhaps across GCs) it will be strikingly different from the cluster of inputs.

Line 119: It was not clear from the text or Fig 1H how the phase-code similarity was computed using the cartesian coordinates and there was no section on pattern separation calculations in the methods. I realized later that the explanation was in the Perceptron paragraph of the methods. The explanation (i.e. making all rate values the same before converting to cartesian coordinates) should be stated earlier. A more detailed illustration and legend for Fig 1H would help too. However, I’m concerned that this manipulation of the data before computing the similarity alters the measure of pattern separation for the phase code (indeed, silent periods are now not silent...). An alternative approach could be to preserve the true points in the 2D cartesian space (with silent periods at [0 0]), compute the similarity between sequences of population of “theta-vectors” (i.e. neural trajectories in the population rate-phase space) and subtract the similarity of the neural trajectories based on rate code only (i.e. matrices of population vectors of rates).

Fig 1I, J: The scope of the results is impaired because of the limited range of R_{in} values (both for the rate and phase codes). Could a larger range of the R_{in} , perhaps using different virtual trajectories or distances between trajectories, be explored? It would be especially important to know how more similar inputs are treated, given the theoretical role of DG pattern separation in supporting discrimination for similar experiences.

A few less important comments about the choice of similarity metric to measure pattern separation:

- The Pearson's correlation coefficient R includes bins without spikes (which must be numerous in the DG population) but the normalized dot product (NDP) does not. Would using NDP (like was done in Braganza et al 2020) from the similarity analysis change results?
- Considering the single-cell average phase of spike times in 100ms bins is fine, but just to make sure it makes sense to average the phase values: what is the variability of phases per bin? It should be fairly constrained in the inputs since that's how they are constructed but what about GCs? Or is it not relevant because there are usually not that many spikes per 100ms in single GCs?
- Could a more native metric of spike-phase similarity be used? A non-binned metric? The limitations of the binned approach are raised by the authors in the tempotron section of the results. However, the tempotron analysis doesn't address the question of pattern separation directly (although it does provide a measure of separability). I am not aware of spiketrain similarity metrics specifically designed to deal with the spike-phase relationship, but similarity metrics for "marked" point processes could be used (see Hino et al 2015 <https://journal.r-project.org/archive/2015/RJ-2015-033/index.html>).
- Since there are several theta cycle (time bins) per trajectory, it would be interesting to show how pattern separation evolves over time, akin to what has been done in the olfactory system (Friedrich & Laurent 2004; Gschwend et al 2015). This approach could consider simple population vectors of mean rate or phase for each theta cycle, or could consider spiketrains contained in those theta cycle and use spiketrain similarity metrics mentioned above.

3-Phase-tuning distribution and phase-to-rate recoding

Fig 2A-B: the broadening of phase tuning in DG after shuffling seems correlated with an increase in firing, i.e. with lack of sparsity. Is it an actual trade-off? Is the broadening due only to increased firing or does inhibition affect phase coding in other ways? Using data from Fig 2I (lower input weights to maintain sparsity) with or without the feedback circuit: is the broadening staying or going back to the phase distribution of the full network? A regression plot between different sparsity levels and variance of the phase-tuning distribution could also be telling.

Understanding how sparsity levels affect the computations tested in the study would be valuable in general. Expansion recoding and sparsity have repeatedly been shown to help pattern separation of rate codes at various timescales (e.g. Severa and Aimone 2016, Cayco-Gajic 2019; Madar et al 2021), but it's not obvious how it would affect pattern separation of phase codes or recoding of that info into rates. Comparing 2H and 2I it seems that a more active DG leads to increased phase-to-rate recoding, but the effect of sparsity (and the separate effect of expansion) could be studied more systematically.

The narrow phase distribution of DG outputs seems at odds with phase-precession in GCs, which superficially seems inconsistent with Skaggs 1996 and Mizuseki 2009, unless we consider they mostly measured MCs.

Importantly, this narrow distribution compared to inputs also seems in contradiction with the pattern separation of the phase-code of figure 1: if there is a narrowing of the phase-tuning of the output spikes, doesn't that mean that the DG outputs are more similar (in terms of phase) than the inputs?

Line 237-40 and Fig 2E: As hinted by the authors, their results suggest a fundamental trade-off between pattern separation (improved when no phase info in inputs) and conservation of information. This is quite an interesting finding, which is reminiscent of experimental observations from Madar et al. 2019a (Scientific Reports). This brings up the question of the computational mechanisms of pattern separation in the model. Doesn't the result suggest that pattern separation, in this model and perhaps in the real DG as well, is largely based on random processes (which tend to degrade information)? Clarifications on how exactly pattern separation of rate codes (and phase codes) is achieved (in terms of spiketrain characteristics) would be welcome.

4- Perceptron

Could the authors provide a rationale for using a perceptron for the decoder? In Cayco-Gajic 2017, the reason given was that analogies exist between cerebellum computations and that of the perceptron. Can the same be said for DG? I understand that both cerebellum and DG are hypothesized to perform pattern separation by expansion recoding, but here the pattern separation of phase-coded information is quite different. Other types of decoders might provide a better estimate of spatial information contained in EC and DG activity.

In methods it is said that perceptrons were initialized on EC data. Does that mean that the training on DG started on a perceptron already trained to recognize the same trajectories from EC data? Or were the perceptrons naive when starting DG learning?

The perceptron and tempotron were used as classifiers to distinguish 2 trajectories, which mixes issues of spatial information content and separability of representations. Why not try to decode virtual position directly? Removing the discrimination/classifying aspect of the analysis and focusing on decoding the spatial info from phase or rate codes might be easier to interpret, and it would more directly answer the question asked by the authors: is phase-coding still informative in DG, and to which extent is it degraded?

This last point brings me back to the relationship between pattern separation and random processes (cf line 151: "it is conceivable that DG inhibitory microcircuits increase decorrelation simply by removing information"). Intuitively, if activity patterns representing different trajectories have been "pattern

separated”, they should be easier to classify by the perceptron or tempotron, no? This observation is either in contradiction with the idea of pattern separation benefiting from (or causing) a degradation of information, or it undermines the idea that the classifiers of trajectories provide a pure measure of spatial information. In fact, in the framework of the index theory of episodic memory, it’s possible that neural trajectories are separated by DG (and thus distinguishable by a classifier) but without preserving spatial information per se, by creating an abstract index for each trajectories.

Line 288 (Fig 3C and D comparison): the authors just said above (line 282) that learning speed could not be compared across conditions. If absolute speed values cannot be compared, how can the effect size be compared across conditions? Since such comparisons are the basis of the whole analysis, some rephrasing and caution in the interpretation might be needed. Perhaps only the sparsity-adjusted networks should be compared?

Line 296: Not being able to compare the effect sizes of different conditions (due to different sparsity levels), it’s hard to go that far in the interpretation. Comparing No fb vs Full is necessary. Assuming we can, I’m not sure concluding that feedback inhibition “improves rate-coding” is correct either.

Fig 3K: The authors tried to mitigate the issue raised above by using the ratio of shuffle/non-shuffle learning speeds to compare across conditions. To answer the question of which element of the network is involved in phase and rate coding, being able to compare the absolute quantity of spatial info in DG phase or rate codes between conditions (with non-shuffle EC inputs and adjusted sparsity levels) would be easier to interpret. I feel like the shuffle/nonshuffle comparison gets at a different question: whether spatial phase info is used/relevant to a given part of the circuit.

Line 314-5 (sensitivity of the analysis to sparsity levels): I think this result highlighted by the authors should incite them to a bit more caution in the preceding interpretations. Overall, figure 3 does not make it clear to me which part of the circuit is responsible for degradation of the spatial phase information. It seems distributed across feedforward and feedback circuits and there are interactions with sparsity levels which are enforced in some capacity by both.

5-Tempotron

Line 318 (“We next asked what the combined effect of the changes to DG rate- and phase-codes might be”): I am not sure that the tempotron analysis answers that question. The approach avoids binning assumptions, but it also doesn’t rely at all on phase information since the tempotron inputs are spike-trains with no information about the underlying theta rhythm. The analysis fits in the story only because of the comparison between shuffled vs nonshuffled phase data. So, it provides information on how phase-info preserved in the input spike timing is used by DG. It’s not about phase-coding but about

theta-organized spike times and synchrony... which might actually be more relevant to read-out neurons.

As is, it's not clear to me what the tempotron analysis adds to the story. More analysis on what the difference between shuffle and nonshuffle data (for both EC inputs and DG outputs) would be useful to interpret the results: shuffling is said to lead to slightly higher firing rates in GCs, but what about inter-spike-intervals (ISI), burstiness and synchrony of spiketrains? What's the ISI of neurons weighted heavily by the tempotron? What about the synchronicity between simultaneous spiketrains (cf different metrics from the Kreuz lab)?

6- Effect of DG phase-to-rate recoding on CA3

Line 412 ("we normalized total weight changes (Fig. 3F) to the respective mean CA3 rates"): Is that normalization appropriate? The unit would be in seconds... Another way to get at the question asked (i.e. separating the effect of DG synchrony and CA3 recruitment) could be to adjust the excitability of CA3 (in terms of percent of active cells as well as firing rates) to keep it constant and vary the synchrony of inputs (or vice-versa).

Line 350 "we asked what the effect of phase-to-rate recoding on plasticity and the formation of attractors in CA3 might be": The analysis does not get at attractor dynamics in CA3. The link to autoassociative memory is only based on the conclusion of Mishra et al 2016 that symmetric STDP facilitates storage and retrieval. But the models used in the Mishra study were quite different from the one in the present study. Moreover, it's not because symmetric STDP is better for autoassociations that a blanket increase in potentiation with that plasticity rule would translate into better storage and recall as well. Actually, a higher recruitment of CA3 pyramidal (i.e. a less sparse network) could lead to more overlap between stored population patterns and thus impair recall. If the authors want to say something about how plasticity promoted by DG inputs improves attractor dynamics, they should test recall in a model that includes PP inputs to CA3.

Overall, I appreciate that the authors attempted to widen the scope of the study and investigate the effect of phase-to-rate recoding in DG on the downstream network, but many other functional consequences other than plasticity at CA3 recurrent collaterals could have been investigated. For instance, when the authors say, line 353, "This [synchrony of DG inputs] strongly suggests that the efficacy of STDP in recurrent CA3 networks should be improved.", the same thing could have been said of the short-term facilitation at MF-CA3 synapses. Choosing specifically the recurrent synapses feels arbitrary.

Conversely, it's unclear how much weight to give the results of that analysis because some critical features of the EC-DG-CA3 dynamical system have not been modeled, like the short-term (and long-term) dynamics of mossy fibers synapses in CA3 or the interaction between DG inputs and EC inputs in CA3.

Here, I am thinking of the hypothesis developed by Hainmueller and Bartos in their 2020 review that the role of DG inputs to CA3 during encoding and consolidation could be to promote plasticity at PP-CA3 and recurrent CA3 synapses in order to make them strong enough to trigger recall without DG inputs. Why not test that hypothesis (which would require testing pattern completion in the CA3 network)?

But before investigating the effect of recoded DG inputs on heterosynaptic plasticity (like the authors did), a more immediate question would have been to test the effect on homosynaptic plasticity. Do recoded DG inputs favor evasion from the feedforward inhibition thanks to short-term facilitation? And what about the role of the peculiar and mysterious non-NMDA-dependent LTP at MF synapses? do the recoded inputs help with that?

Finally, since the guiding thread of this study is the spatial phase code and that the authors widened the scope to CA3, it begs the question of how CA3 gets its phase-code back. Is CA3 recoding back some of the DG spatial rate info into phase through local computations? Or is it getting the phase info from EC, and if so, why is it not degraded like in DG? And, more generally, why would phase-coding be used in EC or CA? Why does the brain bother with phase coding if rate-coding + synchrony is all you need? Is the phase-info necessary to get synchrony in DG?

I want to stress that I do not suggest that the authors should attempt to answer all the questions raised above with new modeling or analyses. They are merely suggestions to improve their story in the direction they see fit, and to discuss the questions that cannot be answered yet.

7 – Mechanism of phase-to-rate recoding in DG

Line 486-8 (“phase specific firing will be translated to spatially selective inhibition (Fig. 6B). GCs driven to spike in the early theta cycle will recruit inhibition leading to the suppression of GC spikes in the late theta cycle”): In the discussion, the authors tackle the question of how DG recodes phase info into rate. They make a perfectly sound hypothesis that would answer an important question. They should analyze their model to test whether it's true.

Minor comments:

Line 24 (“question if”): The grammar feels off. I would replace “if” by “of whether”. Same issue in a few other instances (lines 78, 526, 531)

Line 31: challenge, no s

Line 40: add Mizuseki and Buzsaki 2009?

Line 66: Reference not listed in the bibliography.

Line 122 (“compare to Braganza 2020”): what should we compare?

Line 124: I just re-read Braganza 2020 and get what was done, but as a self-standing sentence, it could be more detailed.

Fig 3C and D: It took me a long time to understand why Fig 3C and 3D (full model) were different. Stating “EC” or “DG” in the figure would help the reader.

Line 249: measured

Line 282: “different” instead of “differential”

Line 376 “sensitivity analysis”: not sure what this means

Point to Point response (Responses to the reviewers will be highlighted in blue; line numbers refer to the manuscript with 'tracked changes' in 'simple markup' mode):

REVIEWER COMMENTS

Reviewer #1 (Remarks to the Author):

Kuru and colleagues take on an intriguing aspect of neuronal coding, namely the handling of phase and rate coded information by different circuits that may have a differential ability to represent/handle phase and rate information. They use 3 distinct and independent circuit models of entorhinal cortex (EC), Dentate Gyrus (DG) and CA3 to primarily examine (1) how location and direction, coded as rate and phase information, from the EC model is transformed by the DG circuit, specifically by feedforward and feedback inhibition, and (2) how the DG model output is processed by a CA3 model which includes inhibition. Starting with the EC, the authors simulate grid cells by simulating animals moving at various trajectories. These spiketrains encode location and direction information as rate, phase, and phase precession. These neural codes are then parsed split into discrete rate and phase information to be utilized as inputs into the DG model. Rather curiously, the processing of the combined information and whether it has an advantage over the individual modalities is not explicitly tested. By using identical seeds for each simulated mouse, they ensure that spike train variability arises from the simulated trajectories travelled. The grid cell spike trains generated have rate information, phase information on direction, as well as phase precession information. They evaluate how information in EC output spike trains is processed by the DG using 3 approaches (1) an assessment of information content to preferentially assess rate coding (2) using "decodability" as number of trials needed for a machine learning perceptron classifier to learn to discriminate input patterns and (3) using a tempotron ML classifier which is designed to process sequential timing information. The simulations are carefully constructed and the use of multiple complementary approaches to assess EC to DG transformation of phase and rate information is a strength. The authors conclude that DG feedback inhibition is necessary and sufficient to convert EC phase code to rate code while maintaining spatial information. By feeding the DG output to a CA3 model, they show that compression of the EC phase into a rate code occupying half of the theta cycle in the DG enhances downstream CA3 plasticity (STDP). The results and approach are interesting and, if they hold up, will provide novel insights into DG information processing. However, there are some conceptual concerns that need to be addressed.

We thank the reviewer for these positive comments

Specific Comments:

1. While each individual model is firmly based in prior studies, the manuscript has limited information about the original models, especially the EC and phase precession models and how these were integrated. For instance, the equation defining $g_w(x,y)$ is not included forcing the readers to look up original papers. Better illustration of the phase precession as part of this model in supplemental figures would be useful. Illustration of how EC phase information components (phase of trajectory and phase precession) are developed would be helpful. Also, it is confusing to introduce the DG model before the EC in fig 1. Illustrating the EC model and its output before the DG model is recommended.

Thank you for this helpful comment. We have now added Supplementary Fig. 1, which more clearly explains how phase precession in the EC grid-cell population was simulated. We have also further clarified this in the methods section, now including the equation for $g_w(x,y)$, and making explicit how the Solstad et al.¹ and the Bush and Burgess² models were combined

(eq.5, lines 613-653). In addition, we have modified Fig.1 to more clearly illustrate how phase precession is introduced (specifically as insets in Fig. 1C). Finally, we now introduce the EC model output before the DG model in both Fig1 and the main text, as recommended by the reviewer (lines 79-92).

2. The polar to Cartesian transformation illustrated in Fig 1H does not isolate rate from phase information as the transformation of the vector based on rate and phase would inherently retain both modalities. Thus, the assumption that x is rate and y is phase on the transformed vector is flawed. If this is true, the analysis in Fig I & J are not meaningful. The authors do introduce a more valid vector to Cartesian transformation in the methods for the perceptron model in Fig 3. If the same strategy was used in fig 1 and 2 it needs to be introduced upfront. If not, the analysis and interpretation of Fig 1 I & J need to be reconsidered.

We are very sorry for this descriptive oversight. The reviewer is correct in assuming that the same isolation procedure as used for the perceptron was used, and we now introduce this at the appropriate position (line 110-111). We have also added a dedicated methods section immediately following the introduction of the EC model (lines 654ff).

3. On a related note, why are there negative R_{in} valued in Fig 1I and 2E, what does this mean and why does it output a positive R_{out} ? Similarly, what happens when the distance between parallel trajectories are the same ($R_{in} = 0$)? Fig 1I shows $R_{in} = 0$ yields $R_{out} > 0$.

The distribution of R_{in} values around zero and the existence of small negative values is simply due to noise given the Poisson process. It has no real meaning, other than perhaps providing some measure of this noise. The keen observation that R_{out} is always positive is correct, and was something that also puzzled us at first. Upon investigation we found that it is also not physiologically meaningful, but instead a result of using random connectivity between EC and DG. Specifically, we found that when implementing the connections between EC and DG without further constraints, then some GCs came to receive fewer ‘anatomical’ inputs than others. These tend to fire less than GCs with more ‘anatomical’ inputs, no matter the simulated input-activity pattern. Because these non-firing cells are consistent between patterns, results appear as a small offset on the y-axis (i.e. all values of R are increased by the same small offset). We decided not to further address this issue, since the effects are small and if anything lead to an underestimation of our pattern separation effect. Furthermore, properly addressing the issue would have required us to change the connectivity scheme between EC and DG from simple random connectivity. Since this could have had nontrivial effects on pattern separation we opted against it.

4. The authors need to explain why the DG R_{in} phase is low? Is this a function of EC grids and/or trajectories simulated being highly variable. Are a broader range of input variabilities, including more highly correlated inputs, expected in vivo?

The reason that the observed range of input correlations for the *rate code* is low is primarily a result of using 100ms windows, which was necessary for the analog definition of phase and rate codes. However, it is markedly different from the typically much longer time windows of seconds or even minutes (e.g. Leutgeb, 2007³). Correlation measures are lower for such very short windows, since the contained code is inevitably much, much sparser. It is important to recognize that this is not a physiological finding, but a well-known (and difficult to avoid) methodological artefact⁴. In other words, the small input correlations we show are consistent with, and indeed equivalent to, the higher input correlations shown in other studies (e.g. computing R_{in} for the full two seconds on our data produces maximal $R_{in}=0.86$, $R_{out}=0.78$). We have attempted to make this more clear in lines 115ff: “Note that the relatively low

maximal input correlation is a methodological consequence of defining correlations based on phases and rates of sparse activity within individual theta cycles, and is consistent with higher values described for longer periods⁵.”

Concerning the low input correlations for the phase code, we believe they are plausible in vivo. In our model, the values are low due to the inhomogeneous Poisson instantiation. Specifically, the ‘signal’ in our model is given only by the Poisson ‘inhomogeneity’, and the actual spikes are created by a random Poisson process. This implies substantial variability in spike times (a homogeneous Poisson process would imply pure noise). Neural spike trains in vivo are well described by Poisson processes, suggesting spike time variability and resultant low phase-code correlations are biologically plausible.

5. In processing of EC phase information in DG, can the authors discuss whether DG feedback inhibition reduces EC phase information as suggested in line 165 or could it compress it to half a cycle? Considering this aspect in relation to pattern separation of phase code, Fig 1J implies more separability of phase information in dentate output (assuming point #2 is not an issue) than in the EC output, is that possible if phase information is degraded?

On the first point: FB inhibition can really only remove spikes within a single bin and there is no plausible mechanism we could envision by which it could ‘compress’ a sequence. We now include an additional analysis which confirms the mechanism proposed in the original Fig 6, which we have now integrated into a revised novel Fig 5. Specifically, these new results confirm that spatial information is increased by the spatially selective removal of late-theta spikes.

On the second point, this seems to be a misunderstanding. If anything, there is less pattern separation in the phase than the rate-code: output correlations are *higher* (closer to unity). But as already noted in the manuscript (line 181ff), since the input correlations are already degraded, the respective difference between rate and phase code must be interpreted with caution. Note that the previous Fig. 1J is now Fig. 1L.

6. The relevance of the shuffled data, where EC phase information is retained but phase precession information is removed (Fig 2E&F), is not fully fleshed out. The conclusion from this analysis is that lack of phase component decreases pattern separation in the rate domain – but this is only the spike precession component of phase code? Removal of this information reduces “phase code R_{in} ” to zero. Does that imply that only phase precession contributes to differences/decorrelation in phase component of EC firing? Wouldn’t the broad theta distribution of spikes contribute to decorrelation of EC spike train output in the phase domain? In this regard, the independent assessment of information content reveals that removing phase precession reduces information content. The perceptron data (3C) indicate that there is phase information beyond the precession in the EC spike trains. This needs to be clarified/discussed.

This is a helpful comment – we now try to be more clear in the manuscript: The only way we introduced phase-coding is via phase precession. Thus in our model, shuffling removes all phase information, and this is by design. This is consistent with the reviewer’s inference that ‘only phase precession contributes to differences/decorrelation in phase component of EC firing’. The remaining theta-phase preference in the shuffled data does not carry spatial information. We have now made this more clear, e.g. in line 170f (‘removes all phase information’).

The reviewer also correctly notes that in Fig. 3C the perceptron with shuffled EC-phases still learns. However, this does not indicate that there ‘is phase information beyond the precession’. Instead, the reason that there is still learning is that, in order for there to be phase-coding, some cells in the population must fire. Because of this, both rate and phase-codes still contain population-code information. It is this population code that the perceptron picks up. We now make this more clear by writing in line 289f: ‘phase shuffling does not perturb the *population code* allowing the phase-perceptron to still learn even if no phase information is present’.

Finally, we should perhaps clarify that we do not mean to imply that there is no phase information beyond phase-precession *in vivo*. However, phase precession is well-studied and examining potential additional sources of phase information would be quite speculative.

7. Since perceptron learning speed improves with both decorrelation and information content, and shuffling increases decorrelation (Fig 2G) it is difficult to assess whether the differences in “decodability” relate to information content (as proposed) or merely reflects decorrelation. This confound needs to be addressed. Moreover, while learning speed was shown to correlate with input decorrelation (Cayco-Gajic et al., 2017) the authors need to justify the use of learning speed as a metric to assess the information content of the input rather than just a reflection of the classifier performance on a particular class of inputs.

The reviewer correctly observes the difficulty to disentangle these factors. However, our data already clearly indicate the relevance of *information content*. Specifically, phase-shuffling *increases decorrelation* but *decreases rate-learning speed*. This is the opposite of what the reviewer correctly observes should be expected. In other words, since the presence of phase coding *decreases* decorrelation, the increase in perceptron rate-learning is likely to be due to an increased information content.

Concerning the question whether perceptron learning speed is also an appropriate measure of information content, or only of decorrelation, we would argue that the former is actually the standard interpretation. Decodability (of any decoding approach) is a way to assess information content⁶. We chose perceptron learning speed because it is one of the simplest and most general ways to assess decodability. Of course the reviewer is correct that *any specific decoding approach* will only reflect performance contingent on i) the strengths and weaknesses of that decoder and ii) the specific input data/ classification task. It is precisely for this reason that we combined our perceptron approach with i) spatial information analysis and ii) tempotron analysis, in our view rendering the conclusions strong.

8. In the discussion, the authors suggest that phase information in EC input may be converted to a “population” code in DG with distinct granule cell subsets responsive to early and late phase EC activity. Is there any evidence for this in the simulation data? Also, how does this comport with the compression of data within half the theta cycle in Fig. 2.

Thank you for this valuable comment. We have now traced the evidence for our inferred mechanism (described in the discussion Fig 6 of the initially submitted manuscript) within the data. Indeed, in combination with this additional analysis, we found the Figure was better placed as a main Results, Fig and have consequently moved it into the results (changing Fig. numbering, the revised Fig. is now Fig. 5). As mentioned above, and now described in detail in lines 372ff, these analyses do not support the notion of ‘compression’. Instead late-theta spikes are ‘deleted’ in a spatially selective manner.

9. The statement in the abstract “through computational modelling of the entorhinal cortex (EC)- dentate gyrus (DG)- CA3 system” is misleading as the study adopts 3 models which are not actively synced rather output from each model is used as input to the subsequent model. This needs to be clarified.

We have changed the respective sentence to “Here, we study this question in the entorhinal cortex (EC)- dentate gyrus (DG)- CA3 system using three distinct computational models” (line 18f).

Overall, the fact that shuffling to remove phase precession reduces decodability of rate information content using both perceptron and tempotron, even when adjusted for rate supports the conclusion that some phase information is transformed to rate information by DG feedback inhibition. Clarifying specific issues raise above would strengthen the study.

Minor:

- Line 31: “This challenges is compounded” \diamond “This challenge is compounded when...”
- Line 38 - 41: “For instance, the precise...and its main input area EC” should be reworded
- Line 64: “ ‘pattern completion’ ” \diamond “ ‘pattern completion’ ”
- Line 249: “measures” \diamond “measured”
- Line 258 / Line 634: loss uses MSE in the caption but RMSE in methods – it’s the same but for consistency
- Line 289: “Fig3C and D” \diamond “Fig. 3C and D”
- Figure 5B: Y-axis formatting is overlapping
- Line 589 Eqn4: $\log_2(r/r)$ – numerator not defined. Is it r_s/r ?
- Line 656: “, where v is membrane...” \diamond delete comma

Thank you very much for these helpful comments. We have made respective corrections.

Reviewer #2 (Remarks to the Author):

In this study, Kuru et al investigate how the common circuit motif of feedback inhibition interacts with phase coding, using simulations based on entorhinal-cortex (EC) to dentate gyrus circuits. This is an interesting and important question as phase coding is a widely observed phenomenon in the brain, and feedback inhibition is a ubiquitous feature of brain circuits. The choice to focus on the entorhinal-dentate gyrus system is sensible as the circuit is well characterized, a lot is known about information coding in this system in vivo, and it is widely thought that feedback inhibition in this circuit is performing a well-defined computational function of pattern separation – i.e. decorrelating similar input patterns while preserving their information content. The authors have previously worked on characterizing feedback inhibition in this circuit using a combination of experimental and computational methods so are well placed to address this question.

Based on their analysis of simulated data the authors conclude that when the dentate circuit with feedback inhibition receives phase coded input from EC, information contained in the phase of input spiking is recoded into spatial patterns of firing rate in the output, a process they term phase-to-rate recoding. This occurs because dentate granule cells that are driven to spike by input arriving early in the theta cycle recruit strong feedback inhibition that suppresses firing later in the cycle. Therefore, it is not simply which granule cells are receiving input but also when that input occurs in the theta cycle that determines the spatial pattern of activity in the granule cell population. A consequence of these dynamics is that the range of theta phase over which the granule cells spike is much narrower than that over which the EC input activity occurs.

Though the study is well motivated, there are two issues which would need to be addressed convincingly to make a strong case that the conclusions are relevant to the real brain system.

The first concerns the nature of the code in the entorhinal-hippocampal system and what function phase precession is actually serving. Phase precession in the hippocampus is thought to enable each theta cycle to encode a short trajectory through space, rather than a single location, with cells firing early in the theta cycle representing the start of the trajectory and those firing late in the cycle the end of the trajectory. Though phase precession in EC is less well characterized, the model of phase precession on which the authors base their simulated EC activity (Bush and Burgess 2020) also assumes such trajectory coding, and shows that the direction of motion can be decoded from simulated EC activity by decoding a series of locations from activity at different phases and fitting a vector through them. The encoding of trajectories by multiplexing rate codes for different locations into different phases of the theta cycle appears fundamental to information processing in this system. However, it is largely ignored in the current work, where analyses assume that each theta cycle represents a single position. This is problematic because the proposed phase-to-rate recoding mechanism appears likely to severely degrade the ability of the system to encode trajectories. This is because granule cells recruited early in the theta cycle by input representing the start of the trajectory, will recruit feedback inhibition preventing input later in the theta cycle, representing later steps of the trajectory, from driving spiking. I feel that given the importance of trajectory encoding by sequences of activity across the theta cycle in the real system, to address the effect of feedback inhibition on coding in this system requires characterising how it affects such codes.

This is a valid observation. The relation between trajectory-coding and phase-to-rate coding is intriguing but complex. It seems to us as if ‘encoding of trajectories by multiplexing rate codes’ represents a form of *rate-to-phase* recoding, i.e. a mirror image of *phase-to-rate* recoding. We have now included a section explicitly discussing the issue (lines 558ff, section titled “Relation to trajectory encoding”).

Our general answer to the question ‘what function phase precession actually serves’ is that there may be multiple functions, which could be simultaneously served in different brain areas. In other words, we believe both types of computation likely serve a system- and task-dependent purpose.

Concerning the specific observation that DG feedback inhibition would severely impair trajectory encoding three things come to mind. First: our finding of the behaviour of DG feedback inhibition was not guided by computational theory, but by confronting experimentally observed EC input patterns with experimentally observed DG inhibition. To the degree that the finding contradicts a computational theory, we are inclined to trust the data over the theory. However, it is not clear to us that the computational theory is contradicted, but it rather seems to be complemented. This brings us to the second issue: the observed effect may be specific to the DG and may still allow trajectory encoding in CA3. Specifically, CA3 cells may simultaneously receive an ‘intact’ theta-phase code from EC and the ‘degraded’ phase-code from DG. It is possible that the conjunction of these codes provides particular benefits, but also that the relevance of codes and inputs varies in a state and/or task dependent manner. Third, there is no reason to believe that the trajectory encoding within the first half of the theta cycle would be degraded. Trajectory encoding could thus still occur in a limited manner in the DG. However, in our view DG is ill-suited for trajectory encoding simply due to its sparsity. This does not undermine the potential role of trajectory encoding in other brain

regions, where feedback inhibition may be weaker or have different spatiotemporal properties, and we have tried to now make this clearer (lines 567ff).

The second issue is whether the proposed mechanism is consistent with *in vivo* activity. A strong prediction of the work is that the output spiking of dentate granule cells should be concentrated into a much narrower range of theta phase than the input received from EC, and should show much less phase precession. Whether this is in fact the case is therefore critical to determining whether the simulations are likely an accurate reflection of what is happening in the real system. The only discussion of this I could find in the manuscript was:

"As expected, the broad phase tuning of grid cells ... led to EC phases well distributed across the full theta cycle. By contrast, in the intact DG circuit GC spiking was concentrated within the first half of the theta cycle. ... This appears to be consistent with experimental data (Mizuseki et al., 2009)."

From my reading of Mizuseki et al it is not at all clear that the *in vivo* data are consistent with the simulations. For example, Mizuseki figure 3D shows that EC layer 2 (EC2) spiking (which provides the input to the dentate gyrus) is overall more strongly modulated by the theta rhythm than dentate gyrus (DG), while Figure 3D shows that the phase of maximum spiking probability is narrower in EC2 than in DG. Figures 6 and 7 show that, if anything, phase precession is stronger in DG than in EC2, and show an example DG neuron which phase precesses across nearly the entire theta cycle. These data appear inconsistent with the prediction from the simulations that the output of DG is more tightly synchronized, and constrained to a narrower range of theta phase, than the input received from EC. The authors need to engage much more seriously with the question of whether their simulations are in fact consistent with *in vivo* data.

This is of course a key point. We now make our reasoning concerning the *in vivo* data more clear in the main text (lines 157ff) and have added a dedicated section in the discussion (lines 576ff). We have further added several novel analyses (Supplementary Figs. 2,4&5) addressing this issue. Briefly, Supplementary Fig 2 shows population mean phase vectors with striking similarity to Mizuseki et al.. Supplementary Fig. 4 shows that our mechanism is highly robust to noise, demonstrating in a somewhat general manner that it is plausible *in vivo*. Supplementary Fig. 5 shows that our mechanism is robust given additional countercyclical LEC inputs, even if these have marked impacts on the theta-phase distribution GC firing.

Before we proceed, it is important to point out an important limitation of available *in vivo* data (or a "gap in the literature" as noted by reviewer 3). Specifically, studies such as Mizuseki et al.'s do not distinguish between dentate mossy cells and GCs. Given the higher mean activity of mossy cells, this implies that they (and likely most other studies prior to 2017) may have reported mostly properties of mossy cells^{7,8}. Indeed, since particularly more sophisticated analysis procedures require minimum numbers of spikes, it seems highly likely that a cell such as that presented in Mizuseki's Fig. 6 is a mossy cell rather than a GC. Nevertheless, recent data suggest that mossy cells have similar theta-modulation to GCs *in vivo*⁸. While there are significant differences between the cell-types, these are quite small (Rev-Fig. 1, adapted from Senzai & Buzsaki 2017⁸).

Rev-Fig. 1, Theta phase modulation of dentate GCs (yellow) and mossy cells (magenta), adapted from Senzai and Buzsaki 2017. While there are formally significant differences between cell-types, effect sizes are small, and the overall distributions similar. GCs have slightly stronger phase modulation, suggesting the Mizuseki data below are conservative with respect to the present question. Note that the measurement of the reference theta-oscillation differs between studies, leading to a variable x-offset.

Notwithstanding this confound, we would argue that the Mizuseki data are consistent with our model, and indeed are so in a rather striking manner.

First, concerning the reviewer's correct observation that individual EC cells are on average more strongly phase modulated (Mizuseki's Fig 3D), it is important to recognize that this has no implication for the population level theta modulation. Specifically, it does not imply that *the phase preferences of individual cells cohere to a preferred population phase*. This can e.g. be clearly seen in EC5 in Mizuseki's Fig 3C,D (black data in Rev-Fig.2), where 'mean modulation per cell' is comparable to DG (Rev-Fig. 2A), but there is no discernible population-level modulation (Rev-Fig. 2B). Technically, the difference is whether a *conventional mean* or a *circular mean* is calculated, where the panel in question (Rev-Fig. 2A) shows the conventional mean.

Rev-Fig. 2, Theta-phase modulation in vivo adapted from Mizuseki et al., 2009⁹. **A)** Mean modulation strength of individual cells (normal mean, not circular mean). **B)** EC5 as example where circular mean is 0 but normal mean is ~ 0.2 (as depicted in A). Each circle represents a cell. **C,D)** Distribution of preferred theta-phases for DG cells (C) and EC2 cells (D). Note that EC2 follows DG in the theta cycle.

In the DG, inspection of the underlying phase preferences reveals that they are almost exclusively between 0° and 180° (**Rev-Fig. 2C**, green), while EC2 cells display preferred phases around the entire cycle (**Rev-Fig. 2D**, light blue), albeit with a prominent peak at $\sim 150^\circ$. What is striking about the EC2 peak is that it *follows the DG peak in the theta cycle*. The mean population vector (the circular mean, displayed as a black line in Mizuseki's Fig 3C or Rev-Fig. 2C,D) shows this clearly. In our modelled data, we find the same pattern of the population mean vector in EC following that of DG in the theta cycle, which we now show in Supplementary Fig. 2a,b. The peak of EC2 activity is itself followed by a conspicuous silence in DG activity, in Mizuseki's data and our model. The same pattern is borne out in Mizuseki's Fig. 8, reproduced below in **Rev-Fig. 3**, juxtaposed to our simulated data).

Rev-Fig. 3, Theta-phase modulation in our model vs in vivo. Depiction of one theta-cycle, i.e. 0° to 360° . DG cells are blue and EC2 cells are grey. **A)** Theta phase modulation in our model (Fig.2A of the manuscript). **B)** Population mean data traced from Mizuseki et al., 2009, Fig. 8 (originally rendered as dashed black lines).

In our view, the similarities are striking, particularly since in both cases they are quite counter-intuitive (why would the input population *follow* the output population in the theta cycle?). Indeed, Mizuseki et al. spend considerable time puzzling over it in their discussion, where they essentially infer that DG cells follow EC cells with delays of over half a theta-cycle, a duration that is “considerably longer [...] than would be expected by axon conduction velocity, synaptic delay, and neuronal integration”. Our model provides a more plausible explanation, namely that the peak in EC does not induce a consecutive peak in DG because it coincides with powerful feedback inhibition from a population of GCs already recruited earlier in the cycle (before MEC reaches its peak).

Other studies, e.g. Skaggs (1996)¹⁰ or Ahmadi (2022)¹¹ also find a narrow phase distribution in plausible GCs. Skaggs (1996) explicitly note that a “lack of activity at the end of the theta cycle was characteristic of the granule cells recorded in this study”. Cells in Ahmadi (2022) are also consistent (their Fig. S3a; cell 1 and 5 are plausible GCs as they have only one place field⁷). However, overall the limitations concerning e.g. the identification of GCs make these similarities difficult to interpret with confidence.

Given the limitations of available *in vivo* data, we strove to further assess the plausibility of our mechanism *in vivo* by conducting additional robustness analyses. Specifically, since *in vivo* numerous additional input or noise sources might interfere with our mechanism, we probed the robustness of phase-to-rate recoding to i) GC membrane noise (Supplementary Fig. 4) and ii) additional GC inputs from LEC (Supplementary Fig. 5). These analyses suggested that the mechanism was highly robust, and thus plausible under more realistic conditions.

Though the manuscript was generally clearly written, there was one aspect of the analysis that would benefit from clearer explanation. Figure 1I and J look at how the network pattern separates inputs separately for rate and phase codes, but important information about what is done differently in figures I and J to isolate rate and phase coding seems to be omitted. The accompanying text (lines 113-120) indicates that the activity of each neuron was first converted into a ‘theta-vector’ defined by the firing rate and mean phase of the spikes in each theta cycle. These vectors are then converted into X and Y coordinates to deal with cycles where no spikes were emitted (where phase is undefined), and these coordinates used to examine pattern separation for phase and rate codes. However, I could not find information about what was actually done to the X and Y coordinates to extract the ‘Phase code’ and ‘rate code’. I assume that this was done by setting either the phase or firing rate component of the theta-vectors to a constant value before computing the X and Y coordinates, but it was not clear from the manuscript.

Thank you. The assumption is correct and we are sorry for this being unclear/misleading. We have now clarified this in both the methods (lines 654ff) and results (line 110f).

Reviewer #3 (Remarks to the Author):

Summary of the findings:

This is an elegant and well-written study of how a model of the dentate gyrus (DG) processes spatial information coded in the rate or phase of MEC grid cells spiking activity. Using a conductance-based model of DG they previously published (Braganza et al. 2020), the authors first showed that DG performs pattern separation of the EC inputs when considering both the rate-code and the phase-code. Thanks to a clever shuffling procedure, they show that the phase-tuning of input spikes, which contains spatial information, affects the rate code of DG output cells: removing that phase-coded spatial information (but not the rate-coded info) increases granule cells (GCs) firing rate and pattern separation of the spatial rate code, and decreases GCs rate-coded spatial information. Those effects are mediated in large part through the feedback hilar circuit, suggesting that net feedback inhibitory dynamics are important for converting the EC phase-code into a rate-code in DG. A perceptron decoding approach suggests that although DG may still perform some phase-coding, the phase-coded spatial information from EC is partially converted in a rate-code. A tempotron decoding approach further shows that the phase information from the inputs is used by DG and is important for preserving spatial information in the spike times and synchrony of the GC population. Manipulations of the feedback inhibitory circuit suggests that it may mediate, at least in part, the use and conversion of the phase information contained in the inputs. Finally, the authors investigated a potential effect of this information processing on CA3, the network downstream of DG: they report that the improved DG synchrony mediated by the feedback circuit helps recruit CA3 pyramidal cells and promotes long-term potentiation of the CA3 recurrent synapses.

General comment:

This study addresses an interesting and refreshing question (how DG processes phase-coded information from its inputs) and it does it with state-of-the-art modeling techniques and rigor. Some questions asked by the authors are not fully answered and the study could be improved and expanded a bit to become even more interpretable and useful. However, I realize that I made a lot of comments and suggestions in my review. Not all points have the same importance, I'm not expecting the authors to follow all my analyses suggestions and it would be perfectly appropriate to address some concerns simply with textual edits.

Thank you for these positive comments and the helpful clarification.

Targeted comments:

1-Model validation and realism

Line 60: Theta phase precession in DG principal cells is actually not well documented. Skaggs et al 1996 and Mizuseki & Buzsaki 2009 did not distinguish between GCs and Mossy Cells (MCs), and their conclusions are likely drawn from a majority of MCs because active cells in DG are biased to MCs (cf Goodsmith & Knierim 2017 and Senzai & Buzsaki 2017), plus it's harder to study precession in low-firing neurons like GCs. Nevertheless, these papers could be used to further validate the model in regard to phase coding. How does phase

modulation and precession in MCs and GCs of the model compare to those experimental reports? Validating the model using data segregating GCs and MCs would be best, but there is a gap in the literature, as far as I know. However, the data might exist: Senzai & Buzsaki 2017 dataset is available with some mice running in linear tracks I think (<https://dandiarchive.org/dandiset/000003/0.210812.1448>).

We concur with the reviewer that there appears to be a gap in the literature. Given the absence of data which allows to directly compare EC and GC theta-phase distributions, the suggestion to concentrate on the comparison of mossy cells and GCs is an interesting workaround. Unfortunately, upon closer consideration, two issues arise. Firstly, the data by Senzai and Buzsaki suggest that mossy cells and GC have a very similar phase preference distributions (notwithstanding a formally significant but tiny shift in preferred phase by $\sim 5\text{-}10^\circ$, reproduced from Senzai and Buzsaki in **Rev-Fig.1**). Secondly, our model is not particularly well constrained concerning mossy cells. In our DG model, the spatiotemporal properties of net-feedback inhibition¹² are carefully matched to experimental data. This did not include a careful calibration of inputs to mossy cells¹³. We would thus interpret any finding of the relative theta-phase coupling between GCs and mossy cells within our model with caution.

Line 99-100 (empirical basis of modeled inputs): The authors assume that the inputs to DG are from grid cells. This is a common assumption in modeling (although see Kim & Royer 2020), but a discussion of the limitation of this approach should at least be included: 1) the data used to constrain the modeled inputs are not guaranteed to come from MEC layer 2 and reach DG, 2) other spatial but non-grid neurons exist in MEC, potentially with phase information. The percentage of such cells in the DG inputs is not well known, 3) LEC is of course a major input ignored. LEC does not have much phase modulation (Deshmukh & Knierim 2010) but it's not obvious how robust the computations investigated here would be when adding inputs with little phase and no spatial information.

This is a very valid point. We focussed our analysis on phase-precessing grid-cells because these promised the clearest view on the hypothesized computation. However, *in vivo*, they are of course just one input among many, raising the question whether our mechanism would remain robust given additional inputs. We now address this crucial question in two ways. First, we have added a random membrane noise component in GCs (New Supplementary Fig. 4). We find that phase-to-rate recoding remains robust even under substantial additional noise, where the noise could reflect any number of additional influences on membrane potential *in vivo*. Second, we have added an LEC input without any spatial information and with phase modulation that is anticyclical with regard to MEC, as observed *in vivo* (Deshmukh & Knierim)¹⁴ (New Supplementary Fig. 5). We found that our mechanism remained robust, even when the additional LEC inputs led to marked changes in overall GC firing times within the theta cycle (e.g. Supplementary Fig. 5D).

The DG model used here was reported in Braganza et al 2020 to include facilitation of feedback inhibition. That's still the case here, correct? What about the short-term dynamics at other synapses, feedforward excitation and inhibition? It feels like PP-GC synaptic dynamics themselves could have large effects on rate or phase coding. Also, the conclusion on the role of the feedback circuit vs feedforward could be biased if short-term dynamics are not included for the feedforward circuit (as those dynamics likely provide more opportunities for interesting emergent computations).

Yes, the synapses still contain facilitation, since it was a hallmark feature required to match the experimental properties of FB inhibition. We agree with the reviewer, that the extremely complex patterns of short-term facilitation and depression at any of the participating synapses could affect our computations. However, investigating the potential impact of each such feature and in particular their combinations is unfortunately beyond the scope of this study. We now explicitly and prominently acknowledge this limitation in line 587-592.

Regarding the robustness of the results, an important question related to any form of temporal coding is whether and how it resists to noise. The impact of neural noise on phase coding in particular could be non-trivial. The determinism of the models used here is another limitation of the study.

This is a very valid concern. As mentioned above, we now include two dedicated additional analyses (Supplementary Figs. 4, 5), in order to test the robustness of phase-to-rate recoding to noise. The injection of random noise for individual trials arguably removes determinism. Both analyses revealed a remarkable robustness to noise. Indeed, the degree of robustness surpassed our expectations – we certainly did not expect the computation to remain robust to 5x realistic noise levels.

The reason is perhaps that our approach was already designed to assess coding in light of noise. Specifically, the use on an inhomogeneous Poisson process assumes maximal temporal noisiness given the phase-precession constraint.

2-Activity patterns and pattern separation

Line 103: more details needed on virtual spatial trajectories: linear or random walk? how long? Are arrows in Fig 1E examples of trajectories?

Thank you for this helpful comment. We now more clearly state early on that “we assumed a mouse moving through virtual space in a straight line at a constant velocity (20 cm/s)” (line 84). We also made the illustration in Fig1 clearer (Fig. 1b, black dashed arrow). We have also made this clearer in the methods (e.g. line 623ff). Finally, we have included a depiction of the entire set of trajectories simulated to assess pattern similarity in supplementary Fig. 1b.

Line 108-120: Need more details (here or in methods) to understand exactly what patterns of activity are considered and how similarity is computed (population vector of binwise rate or phase code with several bins? how many bins?). Line 113 (defining “theta-vectors”) was confusing at first because I was coming with “population vectors” in mind, or time-vectors (where each time bin is a coordinate).

How were correlation between the matrices of population activity computed? (Average across neurons or across time bins or matrices as flattened arrays?)

Thank you for this helpful comment. Correlations were calculated on the flattened arrays of *time-bin x cell x Cartesian coordinate* (20 time-bins of 100ms each, 200 or 2000 cells, 2 Cartesian coordinates). We now more clearly state how exactly this was calculated in the methods (lines 668), and have added an additional clarification in supplementary Fig. 1e.

Showing the trajectories of (rate, phase) theta-points in the 2D Cartesian space, for both inputs and outputs, would be helpful. For DG output it should show a bunch of dots at (0,0)

and a different cluster of points somewhere else. If phase-coding is not important in output (i.e. staying constant for a given GC and perhaps across GCs) it will be strikingly different from the cluster of inputs.

An interesting suggestion. We have now included the suggested plot in supplementary Fig. 2a,b, which bears out both the reviewers prediction and the data in Fig.2a. Specifically, GC points are clustered primarily in the first and second Cartesian quadrants, while the EC points stretch over all quadrants.

Line 119: It was not clear from the text or Fig 1H how the phase-code similarity was computed using the cartesian coordinates and there was no section on pattern separation calculations in the methods. I realized later that the explanation was in the Perceptron paragraph of the methods. The explanation (i.e. making all rate values the same before converting to cartesian coordinates) should be stated earlier. A more detailed illustration and legend for Fig 1H would help too. However, I'm concerned that this manipulation of the data before computing the similarity alters the measure of pattern separation for the phase code (indeed, silent periods are now not silent...). An alternative approach could be to preserve the true points in the 2D cartesian space (with silent periods at [0 0]), compute the similarity between sequences of population of "theta-vectors" (i.e. neural trajectories in the population rate-phase space) and subtract the similarity of the neural trajectories based on rate code only (i.e. matrices of population vectors of rates).

We are sorry for this oversight; the relevant methods description should clearly have come earlier, and this has now been corrected (lines 654ff). We have also made this more clear in the Fig.1 legend (line 140f) Concerning the reviewers concern that setting the rate to one in order to isolate the phase coding component will alter the similarity measure, we must clarify that *silent cells remain silent*. Otherwise we would have had to make an additional assumption about the phase of firing, which would have been problematic. This is indeed the reason why even in the shuffled data (Fig. 3c) a phase-perceptron can still learn (it uses the information *which* cells fire, i.e. the population code).

We hope this addresses the reviewers concern. The proposal to calculate phase-code-correlations by subtracting rate-code-correlations from correlations of the unmanipulated data seems problematic to us. It tacitly assumes that Pearson's R behaves in a linearly additive manner, which is not the case.

Fig 1I, J: The scope of the results is impaired because of the limited range of Rin values (both for the rate and phase codes). Could a larger range of the Rin, perhaps using different virtual trajectories or distances between trajectories, be explored? It would be especially important to know how more similar inputs are treated, given the theoretical role of DG pattern separation in supporting discrimination for similar experiences.

We were at first also concerned that the scope of results would be impaired because of the limited Rin range, and indeed undertook several attempts to achieve higher Rin values. However, Rin similarity of even *identical* trajectories (with different Poisson seeds) proved to still only yield the maximal similarities presently reported. In other words, one factor limiting maximal input similarity is Poisson noise, which we regard as a physiologically plausible and parsimonious modelling choice. This in fact harks back to the reviewer's previous valid point, that it is crucial to consider the role of noise, particularly when investigating time-codes.

Another factor, which leads to an arguably artefactual underestimation of our input similarities is the use of 100ms windows, which was necessary for the analogous definition of

phase and rate codes. It is important to recognize that this is not a physiological finding, but a mathematical artefact (as e.g. shown in Madar 2019⁴). If R of the exact same traces are calculated on longer time windows, then maximal similarities are much larger and indeed comparable to other studies (e.g. $R_{in} = 0.85$, $R_{out} = 0.79$ for non-shuffled data).

Concerning the low input correlations for the phase code, we believe they are also plausible in vivo. In our model, the values are again low due to the inhomogeneous Poisson instantiation. Specifically, the ‘signal’ in our model is given only by the Poisson ‘inhomogeneity’, and the actual spikes are created by a random Poisson process. This implies substantial variability in spike times (a homogeneous Poisson process would imply pure noise). Neural spike trains in vivo have been shown to be well described by such Poisson processes, suggesting spike time variability and resultant low phase-code correlations are biologically plausible. We now make this more clear in lines 115ff.

A few less important comments about the choice of similarity metric to measure pattern separation:

- The Pearson’s correlation coefficient R includes bins without spikes (which must be numerous in the DG population) but the normalized dot product (NDP) does not. Would using NDP (like was done in Braganza et al 2020) from the similarity analysis change results?

Using alternative metrics does not change the results, as can be seen below in Rev. Fig. 4A-C. Though there is a caveat with unadjusted NDP (Rev. Fig. 4D): since our isolation of rate-code assumes a constant angle of 45° , the obtained Cartesian values for x and y are always positive and identical. This leads to an offset, which is notable particularly for the EC-rate-code (Rev. Fig. 4D, note the rightward shift). The reason is that the EC inputs display a very large mean (almost all firing rates are positive). For our analysis it is thus arguably appropriate to adjust NDP by mean subtraction (Rev. Fig. 4B). Note however, that the adjusted NDP is mathematically identical to Pearson’s R. We believe that a detailed account of this would be quite confusing to the average reader, and have therefore included it as a reviewer figure only. We hope it addresses the reviewer’s question.

Rev.-Fig. 4: Alternative similarity metrics. Alternative similarity metrics similarly show pattern separation in both rate and phase code. Each panel shows both non-shuffled and shuffled data. In all cases similarity was computed on the flattened arrays just as in the main Fig. **A:** Pearson's R, for comparison, as shown in the main Figures. **B:** Adjusted (i.e. mean-subtracted) NDP or cosine similarity. **C:** Spearman's rho (rank-ordered correlation). **D:** Cosine similarity or NDP (without mean-subtraction).

- Considering the single-cell average phase of spike times in 100ms bins is fine, but just to make sure it makes sense to average the phase values: what is the variability of phases per bin? It should be fairly constrained in the inputs since that's how they are constructed but what about GCs? Or is it not relevant because there are usually not that many spikes per 100ms in single GCs?

The single cell average phase is of course a simplification which reduces the information of the underlying spike train. We now additionally show raster-plots of individual cells in Fig. 5 and Supplementary Fig. 7 showing that there is indeed substantial variability (spikes accumulated over 20 Poisson seeds). But in a way the question to which degree it 'makes sense' to simplify the description of a spike-train into 'mean phases' or 'mean rates' is precisely the point of defining a coding scheme. In other words, we would argue that the fact that defining the 'phase code' in the way we did allows meaningful analysis proves that it makes some sense. However, since information about the underlying spike trains is invariably obscured (i.e. it to some degree makes no sense, as the reviewer correctly observes), we also added the tempotron analysis.

- Could a more native metric of spike-phase similarity be used? A non-binned metric? The

limitations of the binned approach are raised by the authors in the tempotron section of the results. However, the tempotron analysis doesn't address the question of pattern separation directly (although it does provide a measure of separability). I am not aware of spiketrain similarity metrics specifically designed to deal with the spike-phase relationship, but similarity metrics for "marked" point processes could be used (see Hino et al 2015 <https://journal.r-project.org/archive/2015/RJ-2015-033/index.html>).

Using modern spike train similarity measures to quantify pattern separation and comparing them to traditionally used binned metrics is an interesting idea. However, we do not see a way to meaningfully quantify phase similarity separately from rate. This is also the case for the tempotron, where both timing and the number of spikes factor into the output. One of the most interesting methods to quantify spike pattern similarity for pattern separation would be SpikeShip (<https://www.biorxiv.org/content/10.1101/2020.06.03.131573v2>). SpikeShip deals with different firing rates by assigning mass to single spikes and calculating the optimal cost to make the weighted spike trains identical. While this is an excellent way to deal with different average rates, the rate still factors into the total optimal cost through the mass and therefore the method does not only quantify phase similarity.

We also have a more fundamental concern with spike train similarity measures, when it is also possible to do population-based analysis. The use of spike train measures is of course well established, and *given consecutively recorded single cells*, it is the best possible option. However, we would argue that it is merely a proxy for what one is actually interested in, namely the population-code patterns in short time windows. The reason is simply that neurons integrate over space (i.e. from the population of input neurons at a relatively short moment in time) more than over time (i.e. from a pair of input neurons over long time periods). It is e.g. crucial to recognize that the mean of a population of pairwise spike-train correlations is not the same as the mean (over time bins) of pairwise population correlations. Imagine for instance that the actual information (signal) is coded by 10% of neurons, which fire identical spike-trains at say 10Hz mean, and the rest of the population contains only random noise spikes at say 1Hz. The mean spike-train correlation in this scenario will be ~0.1, while the mean population correlation will approach 1. Which value is more physiologically relevant? In our view it is clearly the latter. The very low correlation for the spike-train measure results from the relative amplification of noise due to cell-wise rate normalization. We have included an excel sheet (CorrelationAnalysis.xlsx) for the reviewer's interest to illustrate these patterns. Regardless, in our view it is not fruitful to investigate spike-train similarity when the more relevant population similarity is methodologically assessable (as it is for our modelled data).

- Since there are several theta cycle (time bins) per trajectory, it would be interesting to show how pattern separation evolves over time, akin to what has been done in the olfactory system (Friedrich & Laurent 2004; Gschwend et al 2015). This approach could consider simple population vectors of mean rate or phase for each theta cycle, or could consider spiketrains contained in those theta cycle and use spiketrain similarity metrics mentioned above.

We are also intrigued by these studies and their relation to our study. But their question is somewhat different. They observe how a circuit dynamically increases the separation given a *single static input* due to internal processes. By contrast, ours is a *dynamically changing input*. Firstly, our circuit contains no internal self-sustaining mechanism that could support a divergence over time of a single static input. Secondly, even if we modelled such a mechanism (e.g. recurrent MC-GC circuits), its effects would overlap with the online changes of inputs. A comparison between our and their studies is thus very difficult and hinges on

several unknowns (e.g. how the dynamics of changing inputs and internal network dynamics are coordinated within the hippocampus¹⁵).

3-Phase-tuning distribution and phase-to-rate recoding

Fig 2A-B: the broadening of phase tuning in DG after shuffling seems correlated with an increase in firing, i.e. with lack of sparsity. Is it an actual trade-off? Is the broadening due only to increased firing or does inhibition affect phase coding in other ways? Using data from Fig 2I (lower input weights to maintain sparsity) with or without the feedback circuit: is the broadening staying or going back to the phase distribution of the full network? A regression plot between different sparsity levels and variance of the phase-tuning distribution could also be telling.

We assume the reviewer meant to write “the broadening of phase tuning in DG after inhibition” here (we did not observe a broadening after shuffling). It is indeed the case that the sparsity adjusted DG theta-phase distribution remains broader. We now show this in Supplementary Fig. 3e.

Understanding how sparsity levels affect the computations tested in the study would be valuable in general. Expansion recoding and sparsity have repeatedly been shown to help pattern separation of rate codes at various timescales (e.g. Seiver and Aimone 2016, Cayco-Gajic 2019; Madar et al 2021), but it's not obvious how it would affect pattern separation of phase codes or recoding of that info into rates. Comparing 2H and 2I it seems that a more active DG leads to increased phase-to-rate recoding, but the effect of sparsity (and the separate effect of expansion) could be studied more systematically.

This is indeed an interesting issue. We had more systematically explored the impact of sparsity to make sure that our findings are not due to a confound in this respect, and now include these data as Supplementary Fig. 3a-d. We would argue that “comparing 2H and 2I” if anything suggests that *a less active DG leads to increased phase-to-rate recoding*: The difference between non-shuffled and shuffled for e.g. ‘no fb’ is bigger in 2I (the less active case) than 2H. The more systematic analysis in Supplementary Fig. 3 also bears this out. The difference between non-shuffled and shuffled grows larger for sparser networks. The Fig. also shows that the difference remains robust for the plausible range of DG activity (<0.5Hz), and indeed the non-*phase*-shuffled data appear to maintain higher spatial *rate*-information for the entire range tested.

There are doubtlessly a number of additional interesting phenomena visible in this dataset, relating e.g. to the variance between grid seeds. However, these remain difficult to interpret without further analysis, which we feel goes beyond the scope of the present manuscript.

The narrow phase distribution of DG outputs seems at odds with phase-precession in GCs, which superficially seems inconsistent with Skaggs 1996 and Mizuseki 2009, unless we consider they mostly measured MCs.

While the reviewer is correct to caution whether the recorded cells are GCs, we are unsure why she/he thinks these data at odds with our findings. E.g. Skaggs 1996 explicitly note that a “lack of activity at the end of the theta cycle was characteristic of the granule cells recorded in this study” and their Fig 5 suggests that DG cells indeed fire over a much shorter portion of the theta cycle than CA1 cells. As outlined above (see Rev-Fig.2) the Mizuseki data also bear this out.

Importantly, this narrow distribution compared to inputs also seems in contradiction with the pattern separation of the phase-code of figure 1: if there is a narrowing of the phase-tuning of the output spikes, doesn't that mean that the DG outputs are more similar (in terms of phase) than the inputs?

This seems plausible at first. However, two things should be noted. First, Pearson's R (just as e.g. NDP) entails a normalization for scale (specifically variance in the case of R), which adjusts for the differences in theta-range. Second, to the degree that the reviewers reasoning might nevertheless be correct, our data clearly show that some other mechanism must be compensating for it, perhaps expansion recoding.

Line 237-40 and Fig 2E: As hinted by the authors, their results suggest a fundamental trade-off between pattern separation (improved when no phase info in inputs) and conservation of information. This is quite an interesting finding, which is reminiscent of experimental observations from Madar et al. 2019a (Scientific Reports). This brings up the question of the computational mechanisms of pattern separation in the model. Doesn't the result suggest that pattern separation, in this model and perhaps in the real DG as well, is largely based on random processes (which tend to degrade information)? Clarifications on how exactly pattern separation of rate codes (and phase codes) is achieved (in terms of spiketrain characteristics) would be welcome.

This is no coincidence, as our understanding of this matter was influenced by Madar2019a (Scientific Reports). However, it is worth emphasizing that the notion of *pattern separation by noise* is distinct from the original conceptualization of the computation (expansion recoding), where information is conserved. While the 'fundamental trade-off' applies to pattern separation by noise, there is no such trade-off for pattern separation via expansion recoding. Indeed, since the initial submission of this manuscript, a preprint explicating this matter has been posted¹⁶. The preprint assumes what is arguably the canonical view on the matter, namely that pattern separation should in principle conserve information.

But we agree with the reviewer that there is no fundamental reason why pattern separation by noise could not also be useful in vivo, for instance for indexing unique episodes. Ultimately the question boils down to whether or not it is physiologically desired that identical inputs to DG will lead to identical outputs. If this is not desired (e.g. because pattern similarity is assessed elsewhere), then pattern separation by noise would be a perfectly useful mechanism. Notions that the DG (with its neurogenesis) could e.g. continuously supply novel random IDs for novel experiences fit into this understanding.

Our results could be interpreted as indicating a combination of both mechanisms (i.e. pattern separation by noise and pattern separation via 'deterministic' recoding). The fact that the various decoding approaches can learn to decode trajectories, as well as the information analysis, demonstrates information-conservation. At the same time, shuffling led to decorrelation by 'noise', i.e. at the cost of information content (compare Fig. 2G and I). But it should be made clear, that more generally, our approach conforms with the information conservation perspective, i.e. for us decodability is predicated on the conservation of information. Note that this is distinct from ¹⁷, who utilize a perceptron to learn pure noise, i.e. their perceptron performs exclusively 'overfitting' and would have no predictive value out-of-sample.

As mentioned above, we also have some general reservations concerning the use of spike-train analysis, when population-based analysis is also possible (such as in the present case). This is in no way meant to imply that spike-train analysis does not have its place: in the absence of population data it is the best available approach.

4- Perceptron

Could the authors provide a rationale for using a perceptron for the decoder? In Cayco-Gajic 2017, the reason given was that analogies exist between cerebellum computations and that of the perceptron. Can the same be said for DG? I understand that both cerebellum and DG are hypothesized to perform pattern separation by expansion recoding, but here the pattern separation of phase-coded information is quite different. Other types of decoders might provide a better estimate of spatial information contained in EC and DG activity.

We chose a perceptron decoder as arguably the simplest and most general approach with well-defined learning speed, allowing to assess decodability. Furthermore, the general theory used to motivate this approach, which is cited by Cayco-Gajic¹⁷ (namely¹⁸), is what the same authors in a recent review term Marr-Albus theory of pattern separation¹⁹, which indeed explicitly draws analogies between cerebellum and DG (and olfactory circuits). However, the reviewers concern remains valid: it is difficult to finally establish which types of decoders are best suited to particular applications. This is indeed one of the major reasons we followed up our perceptron analysis with the tempotron analysis, which confirmed the main results. The consistency between our spatial information analysis and decoding approaches provides further confidence in our results. While we did not attempt to use additional types of decoders, the scope to do so is of course vast, and it seems highly likely that additional insights could be gleaned.

In methods it is said that perceptrons were initialized on EC data. Does that mean that the training on DG started on a perceptron already trained to recognize the same trajectories from EC data? Or were the perceptrons naïve when starting DG learning?

Perceptrons were naïve when starting DG learning. What we meant to say is that the data for both DG and EC were generated given distinct grid-seeds. We have now made this clearer (line 621ff).

The perceptron and tempotron were used as classifiers to distinguish 2 trajectories, which mixes issues of spatial information content and separability of representations. Why not try to decode virtual position directly? Removing the discrimination/classifying aspect of the analysis and focusing on decoding the spatial info from phase or rate codes might be easier to interpret, and it would more directly answer the question asked by the authors: is phase-coding still informative in DG, and to which extent is it degraded?

Phase precession, which underlies all phase coding in the present study, is not well-defined for individual positions but depends on the trajectory through space. For instance, depending on if a mouse passes through a grid field from left or right, the same position will be coded by completely different phases. The present analysis is thus not possible for positions. Intuitively, the question we ask is closer to if the mouse will be able to separate episodes (which are analogous to spatial trajectories) rather than individual positions.

This last point brings me back to the relationship between pattern separation and random

processes (cf line 151: “it is conceivable that DG inhibitory microcircuits increase decorrelation simply by removing information”). Intuitively, if activity patterns representing different trajectories have been “pattern separated”, they should be easier to classify by the perceptron or tempotron, no? This observation is either in contradiction with the idea of pattern separation benefiting from (or causing) a degradation of information, or it undermines the idea that the classifiers of trajectories provide a pure measure of spatial information. In fact, in the framework of the index theory of episodic memory, it’s possible that neural trajectories are separated by DG (and thus distinguishable by a classifier) but without preserving spatial information per se, by creating an abstract index for each trajectories.

As noted above, the notion of pattern separation by noise (e.g. quantified by Pearson’s R and many other distance metrics) is probably most clear in the extreme case where neuronal population activity is completely random, as in ¹⁷. For random neuronal activity, pattern separation is maximized, because output patterns will be completely uncorrelated. This might be relevant to create an abstract index (similar to a unique ID, or a hash code in computer science) that can uniquely identify an episodic memory, but randomness does not contain spatial information. We cannot judge the degree to which this is what the DG may do *in vivo*, but think it is certainly plausible that it plays a role.

In our perceptron learning paradigm, the understanding of pattern separation is closer to the original theory, and arguably the canonical understanding. The key difference is that classification is not about uniquely identifying a single episode (a single Poisson seed) but deciding whether the neuronal activity was generated while traversing the same trajectory (which was used to generate 20 spike-trains, each with a separate Poisson seed). In other words, the perceptron must learn to disregard the noise (variability between Poisson seeds), by isolating the information (what about spike-patterns is conserved). We have added a citation to make this clearer (line 192).

Line 288 (Fig 3C and D comparison): the authors just said above (line 282) that learning speed could not be compared across conditions. If absolute speed values cannot be compared, how can the effect size be compared across conditions? Since such comparisons are the basis of the whole analysis, some rephrasing and caution in the interpretation might be needed. Perhaps only the sparsity-adjusted networks should be compared?

This is a valuable comment, as our phrasing was indeed misleading. What we meant is that the *absolute learning speeds* cannot be compared. It is for this reason that we normalized the learning speeds to the respective shuffled data, which have similar sparsity and thus represent an internal control. We decided to include the absolute, normalized (to shuffled), and the sparsity-adjusted data, i.e. two different ways to control for sparsity. The reason is that sparsity can have complex consequences of its own. For instance, sparsity can lead to an artefactual inflation of decorrelation measures, because in Pearson’s R noise-firing of a constant absolute frequency will be amplified more when overall firing is sparser. It is difficult to predict how such effects might influence perceptron learning, so it seemed most appropriate to include both controls. We have modified the text to make this more clear (line 291ff).

Line 296: Not being able to compare the effect sizes of different conditions (due to different sparsity levels), it’s hard to go that far in the interpretation. Comparing No fb vs Full is

necessary. Assuming we can, I'm not sure concluding that feedback inhibition "improves rate-coding" is correct either.

As outlined above, we believe that the normalization to shuffling and the sparsity adjustment allow us to compare the conditions, which is indeed core to our results. We should also emphasize that at this stage of the manuscript the result is already confirmatory, i.e. it is consistent with the previous 'spatial information' result, allowing increasing confidence in the conclusion that feedback inhibition "improves rate-coding".

Fig 3K: The authors tried to mitigate the issue raised above by using the ratio of shuffle/non-shuffle learning speeds to compare across conditions. To answer the question of which element of the network is involved in phase and rate coding, being able to compare the absolute quantity of spatial info in DG phase or rate codes between conditions (with non-shuffle EC inputs and adjusted sparsity levels) would be easier to interpret. I feel like the shuffle/nonshuffle comparison gets at a different question: whether spatial phase info is used/relevant to a given part of the circuit.

This was also our first intuition. Indeed concerning spatial rate information we display the data the reviewer requests in Fig.2I (Skaggs information for full vs. sparsity-adjusted noFB). A statistical comparison of these two groups confirms our main finding, showing significantly lower information in the noFB case (t-test: $p=0.0006$). Unfortunately, we are unaware of a simple way to assess spatial *phase* information in the way the reviewer would like to see (there is no equivalent to Skaggs spatial information in the phase domain). It is precisely for this reason that we introduced the shuffling procedure, which we maintain is an ideal way to isolate the effect of phase information for the function of each network element.

Concerning the question "which element of the network is involved in phase and rate coding", we would define "involvement" as a causal interaction between i) phase information and ii) the circuit element, in light of the readout of interest (i.e. spatial rate information). Given this definition, 'whether spatial phase info is used/relevant to a given part of the circuit', as assessed by an impact on spatial rate information, seems to ask about 'involvement'.

Line 314-5 (sensitivity of the analysis to sparsity levels): I think this result highlighted by the authors should incite them to a bit more caution in the preceding interpretations. Overall, figure 3 does not make it clear to me which part of the circuit is responsible for degradation of the spatial phase information. It seems distributed across feedforward and feedback circuits and there are interactions with sparsity levels which are enforced in some capacity by both.

We agree with the reviewer's analysis of the complexity of the issue and now make this clearer in the manuscript (lines 331ff), stating "...these results suggest complex interactions of microcircuit contributions, sparsity and information (see Supplementary Fig. 3)...". Nevertheless, we would argue our main result holds in spite of this complexity for two reasons:

First, the question whether phase code is destroyed and if so by which circuit, does not really affect our main claim, i.e. that phase-code is translated to rate-code. What we consistently demonstrate is that phase-shuffling degrades rate coding, *i.e. phase code is translated to rate code*. A translation between two codes does not require the source code to be destroyed.

Second, we believe that overall our results still strongly suggest a dominant role for feedback inhibition, and a destruction of phase information. We do not believe the rate-adjusted analysis (Fig.3k) is inherently more interpretable than the non rate-adjusted analysis (Fig.3i).

For instance, if the improvement in phase-decodability for nofb in Fig.3i is simply due to decreased sparsity, why then is decodability significantly decreased for noff in the same panel, although removing feedforward inhibition also decreases sparsity.

Meanwhile, a potential explanation why the nofb-improvement of phase-decodability does not appear in the sparsity-adjusted data (Fig.3K) may be saturation effects: Perhaps phase information can simply not be utilized below a certain sparsity level, where decoding becomes mainly dependent on population-coding. Indeed, we suspect this is the case and are thus inclined to trust Fig.3i over Fig3k. However, note that we make no strong claims about the potential role of the feedforward circuit and explicitly write this in a dedicated section of the discussion (lines 540ff).

5-Tempotron

Line 318 (“We next asked what the combined effect of the changes to DG rate- and phase-codes might be”): I am not sure that the tempotron analysis answers that question. The approach avoids binning assumptions, but it also doesn’t rely at all on phase information since the tempotron inputs are spike-trains with no information about the underlying theta rhythm. The analysis fits in the story only because of the comparison between shuffled vs nonshuffled phase data. So, it provides information on how phase-info preserved in the input spike timing is used by DG. It’s not about phase-coding but about theta-organized spike times and synchrony... which might actually be more relevant to read-out neurons.

In our view, ‘phase-coding’ is *precisely* and *fully* captured by ‘theta-organized spike times and synchrony’. The fact that we, as experimenters use the concept of *phase* to better understand the phenomenon should not detract from the fact that individual neurons will only ever perceive input spike patterns. In our view, it is unlikely that neurons explicitly use the actual field-potential theta-oscillation in any relevant computational sense, and that is certainly not what we mean to imply by the term phase-coding. Since we agree with the reviewer that the tempotron is well-suited to assess ‘theta-organized spike times and synchrony’, we would thus argue that the tempotron necessarily also well suited to assess the combination of rate and phase-coding. We have now made this clearer in the manuscript (lines 368ff).

As is, it’s not clear to me what the tempotron analysis adds to the story. More analysis on what the difference between shuffle and nonshuffle data (for both EC inputs and DG outputs) would be useful to interpret the results: shuffling is said to lead to slightly higher firing rates in GCs, but what about inter-spike-intervals (ISI), burstiness and synchrony of spiketrains? What’s the ISI of neurons weighted heavily by the tempotron? What about the synchronicity between simultaneous spiketrains (cf different metrics from the Kreuz lab)?

This is an interesting point. As outlined above, we reasoned that the tempotron would inform us on improvements in ‘theta-organized spike times and synchrony’ at the population-level, but had not considered the possibility that simple mean changes in spike train characteristics could drive our effects. We thus now include additional analysis (Supplementary Fig. 6, lines 366ff), to explore if three key spike-train characteristics can predict the relative contribution of individual GCs to tempotron learning, namely Skaggs information (per GC), mean ISI and the Fano Factor. We found no such correlation, consistent with our interpretation that it is about theta organized spike times and synchrony of populations of GCs.

In our view the tempotron (and our CA3 model) are far better suited to assess biologically relevant transient patterns of synchrony than abstract measures of spike train synchronicity between GC pairs. Such measures are far inferior approximations, because they abstract away from the population, thus losing the information ‘when synchrony matters’ for ‘which combination of GCs’. For instance, a measure of synchronicity for several pairs of spike trains could be very low if the GCs have continuous sparse random ‘noise spikes’ but where all GCs reliably spike in synchrony at one specific time-point during the trajectory during which they drive tempotron learning or CA3 cell activity/plasticity.

Concerning the more general question what the tempotron analysis contributes to our story, we would refer back to the reviewer’s point that any decoder will have particular strengths and weaknesses, and none provide a universal measure of information or decodability. In our view the tempotron complements the perceptron as an assessment of decodability, which is ideally suited to assess the “combined effects of phase and rate code” because it assesses “theta-organized spike times and synchrony”.

6- Effect of DG phase-to-rate recoding on CA3

Line 412 (“we normalized total weight changes (Fig. 3F) to the respective mean CA3 rates”): Is that normalization appropriate? The unit would be in seconds... Another way to get at the question asked (i.e. separating the effect of DG synchrony and CA3 recruitment) could be to adjust the excitability of CA3 (in terms of percent of active cells as well as firing rates) to keep it constant and vary the synchrony of inputs (or vice-versa).

Thank you for this comment. We have now phrased this more precisely (line 497). It is more appropriate to think about this as unit-less. Either the weight-change should also be conceived as per unit time, or the total number of spikes (rather than rate) should be considered. Since all these differ simply by a constant factor, the difference is not relevant.

Concerning the reviewer’s suggestion to artificially vary the synchrony, we considered the same idea, specifically to take GC spike trains of the full network and then to artificially decrease synchrony by reshuffling theta-phase based on the no-FB distribution. However, this would destroy spike relations which would not necessarily be impaired by the absence of FB inhibition. In our view the analysis provided tests the role of FB inhibition much more precisely, since the synchrony decrease is less artificially imposed but reflects the actual pattern to be expected without FB inhibition.

Line 350 “we asked what the effect of phase-to-rate recoding on plasticity and the formation of attractors in CA3 might be”: The analysis does not get at attractor dynamics in CA3. The link to autoassociative memory is only based on the conclusion of Mishra et al 2016 that symmetric STDP facilitates storage and retrieval. But the models used in the Mishra study were quite different from the one in the present study. Moreover, it’s not because symmetric STDP is better for autoassociations that a blanket increase in potentiation with that plasticity rule would translate into better storage and recall as well. Actually, a higher recruitment of CA3 pyramidal (i.e. a less sparse network) could lead to more overlap between stored population patterns and thus impair recall. If the authors want to say something about how plasticity promoted by DG inputs improves attractor dynamics, they should test recall in a model that includes PP inputs to CA3.

We did not mean to imply that we investigated the formation of attractors, and have made the respective phrasing more clear (lines 435f).

Overall, I appreciate that the authors attempted to widen the scope of the study and investigate the effect of phase-to-rate recoding in DG on the downstream network, but many other functional consequences other than plasticity at CA3 recurrent collaterals could have been investigated. For instance, when the authors say, line 353, “This [synchrony of DG inputs] strongly suggests that the efficacy of STDP in recurrent CA3 networks should be improved.”, the same thing could have been said of the short-term facilitation at MF-CA3 synapses. Choosing specifically the recurrent synapses feels arbitrary.

Conversely, it's unclear how much weight to give the results of that analysis because some critical features of the EC-DG-CA3 dynamical system have not been modeled, like the short-term (and long-term) dynamics of mossy fibers synapses in CA3 or the interaction between DG inputs and EC inputs in CA3.

In our view, the recurrent CA3-CA3 synapse is by far the most important starting point for investigation, because it is the canonical substrate of ‘auto-associative memory’. It is arguably a model for many of the comparable computations in other brain areas. That being said, the reviewer’s points are of course true: short-term facilitation at MF-CA3 synapses is likely to similarly enhance CA3 cell recruitment and plasticity and any number of features that were not modelled will play a role and interact with the mechanism we describe. There can also be no doubt that modelling each of these mechanisms and in particular the complex interactions between EC and DG inputs to CA3, are extremely interesting questions. However, the present model of CA3 was explicitly designed to test the very specific hypothesis that phase-to-rate recoding might improve CA3-recurrent plasticity. In our view the model achieves this, thus confirming a specific hypothesis about a possible function of phase-to-rate recoding. Given the space limitations of a journal paper (we had considerable difficulties shortening the article to the required length), we would argue that the presently raised questions, though all extremely interesting, are simply beyond the scope of the present paper.

Here, I am thinking of the hypothesis developed by Hainmueller and Bartos in their 2020 review that the role of DG inputs to CA3 during encoding and consolidation could be to promote plasticity at PP-CA3 and recurrent CA3 synapses in order to make them strong enough to trigger recall without DG inputs. Why not test that hypothesis (which would require testing pattern completion in the CA3 network)?

Again, these are important questions. We considered testing pattern completion in CA3, which is of course an extremely interesting problem that naturally arises in the present context. The main reason we did not proceed is because of a core conceptual difficulty, which is unresolved within the literature. Specifically, the present model and typical pattern completion models have different conceptions of time. In the present model time is ‘real’ time, i.e. the time that passes as the mouse traverses an arena. In pattern completion models such as Mishra et al. 2016, time is computational time, i.e. it is the time that passes as the network converges on an ‘attractor’ ensemble of cells.

Testing pattern completion in a way analogous to the previous literature would thus have entailed having to specify a computational theory about the relation of virtual and real time. Many possibilities are conceivable: i) a short virtual time convergence could have been simulated for every theta-cycle resulting in a vector of attractors; ii) a way to virtualize time, e.g. coding time point via spiking as a component of a single attractor coding for the whole

trajectory; iii) a CA3 attractor which has an explicit time dimension, which perhaps simply compresses real time. Ultimately, lacking an at least partially established or accepted way of dealing with this issue, it seemed that any potential solution we could have proposed would have been highly speculative and exceedingly difficult to defend. It also goes well beyond the key subject of our paper.

But before investigating the effect of recoded DG inputs on heterosynaptic plasticity (like the authors did), a more immediate question would have been to test the effect on homosynaptic plasticity. Do recoded DG inputs favor evasion from the feedforward inhibition thanks to short-term facilitation? And what about the role of the peculiar and mysterious non-NMDA-dependent LTP at MF synapses? do the recoded inputs help with that?

This appears to be a misunderstanding. STDP is a homosynaptic form of plasticity (between CA3 cells), which we agree is the most immediate question. Concerning the other suggestions, these are intriguing ideas, but they seem only very loosely associated with the core mechanism. The primary goal of our article is to introduce the novel concept of phase-to-rate recoding and analyse its underlying mechanism. We feel the addition of the CA3 model further adds to this, by exploring a key implication, which is directly suggested by the properties of the computation (namely increased synchrony in a phase-degraded rate code). While there are literally infinite possible paths for further analyses, it is not clear to us exactly which predictions arise from the mechanisms mentioned by the reviewer. Such analyses, while very interesting, thus seem beyond our present scope.

Finally, since the guiding thread of this study is the spatial phase code and that the authors widened the scope to CA3, it begs the question of how CA3 gets its phase-code back. Is CA3 recoding back some of the DG spatial rate info into phase through local computations? Or is it getting the phase info from EC, and if so, why is it not degraded like in DG? And, more generally, why would phase-coding be used in EC or CA? Why does the brain bother with phase coding if rate-coding + synchrony is all you need? Is the phase-info necessary to get synchrony in DG?

These are important questions, and we have now added a section in the discussion to address them (lines 575ff). Briefly, our current hypothesis is that the CA3 will receive a mostly intact phase-code from EC alongside the partially degraded DG phase code. Since the degradation of the phase code in DG resulted from the (experimentally determined) specifics of the feedback circuit, we see no reason to assume that this would generalize to CA3, which likely has weaker feedback inhibition. It is conceivable that the combination of a sparse, synchronized and decorrelated DG input with the phase-precessing PP input yields particular benefits in CA3. How exactly this might work clearly needs more study as we now explicitly state (line 573).

Why phase coding is used in general is of course also an intriguing question. The prevailing answer in the literature, as noted by reviewer two, is that it supports trajectory encoding. We believe our mechanism provides an additional answer, and that the function of phase coding may vary in a brain region and state dependent manner (there is no reason to assume there is only one single function). In short, we have demonstrated that the presence of the EC phase code improves the information content of a DG rate-code of given synchrony and sparsity. There may be other computations (similar to pattern separation) which require these properties, and our results suggest that upstream phase coding would be useful whenever this is the case. However, other computations (such as trajectory encoding) require other properties, so a claim that 'rate-coding + synchrony is all you need' seems clearly wrong.

I want to stress that I do not suggest that the authors should attempt to answer all the questions raised above with new modeling or analyses. They are merely suggestions to improve their story in the direction they see fit, and to discuss the questions that cannot be answered yet.

Thank you for this helpful clarification.

7 – Mechanism of phase-to-rate recoding in DG

Line 486-8 (“phase specific firing will be translated to spatially selective inhibition (Fig. 6B). GCs driven to spike in the early theta cycle will recruit inhibition leading to the suppression of GC spikes in the late theta cycle”): In the discussion, the authors tackle the question of how DG recodes phase info into rate. They make a perfectly sound hypothesis that would answer an important question. They should analyze their model to test whether it’s true.

Thank you for this valuable comment. We have now added new analyses to the previous discussion Fig. 6, where we trace the proposed mechanism in the data. Given the centrality of these points, we have moved the respective figure into the main results (it is now Fig. 5), and have added a dedicated results section (specifically lines 372ff).

Minor comments:

Line 24 (“question if”): The grammar feels off. I would replace “if” by “of whether”. Same issue in a few other instances (lines 78, 526, 531)

Line 31: challenge, no s

Line 40: add Mizuseki and Buzsaki 2009?

Line 66: Reference not listed in the bibliography.

Line 122 (“compare to Braganza 2020”): what should we compare?

Line 124: I just re-read Braganza 2020 and get what was done, but as a self-standing sentence, it could be more detailed.

Fig 3C and D: It took me a long time to understand why Fig 3C and 3D (full model) were different. Stating “EC” or “DG” in the figure would help the reader.

Line 249: measured

Line 282: “different” instead of “differential”

Line 376 “sensitivity analysis”: not sure what this means

Thank you for these comments. We have modified the manuscript to account for these errors and suggestions. Briefly, a ‘sensitivity analysis’ explores the ‘sensitivity’ of a finding to variations in input parameters (we have tried to make this more self-explanatory in the text).

REVIEWER COMMENTS

Reviewer #1 (Remarks to the Author):

The authors have fully and adequately addressed all my prior critique with extensive revisions including additional simulations, controls and figures. A couple of minor edits are suggested

1. There are several instances where it would help the reader and sentence structure to call out author name and year of references rather than the numbered reference. Please review the entire manuscript for this and revise as needed.
2. Lines 220-221. The authors indicate that “spatial information content is almost identical”, which is conceptually accurate but at odds with the statistically significant difference. Rephrasing to indicate that the statistical difference is not meaningful would improve clarity.
3. Line 571: “conjunction of thus functionally differentiated” should read “conjunction of these functionally differentiated”

Reviewer #2 (Remarks to the Author):

The authors have engaged with the issues I raised in the first review and I am sympathetic to the idea that discrepancies between aspects of activity in the model and in vivo measurements are not sufficiently serious to preclude publication. There will inevitably be some discrepancies between simplified models and in-vivo data, and the as they point out in the revised manuscript, a strength of their proposed mechanism is that it does reproduce the apparently paradoxical in-vivo observation that the phase of peak firing in the DG precedes that in it's EC input.

Nonetheless, I do think there remain real discrepancies between their modeling and in vivo data, and these should be discussed head-on. Specifically, from Mizuseki et al 2009 it appears that:

- The overall firing rate of the principal cell population in DG is not more strongly theta modulated than that in EC (Mizuseki Figure 3A). See also that the resultant vector for preferred phases in Mizuseki fig 3C is longer for EC than that for DG. This is not the case in the model.

- DG principal cells show theta phase precession that appear to be at least as strong if not stronger in DG as that in EC2 (Mizuseki Figure 7). Though phase precession in DG is not explicitly analysed in the paper, this in vivo finding does not appear consistent with the narrow theta-phase range of DG spiking in the model.

Reviewer #3 (Remarks to the Author):

The manuscript is much improved and even more pleasant to read than the original submission. The authors addressed a large number of concerns, adding new analyses to successfully show the robustness of their conclusions against different sources of noise, to better understand the relationship with DG sparsity and to demonstrate the mechanism of phase-to-rate recoding in DG. The claims of the paper are overall well-supported. However, I have two major remaining concerns: 1) the EC inputs used in the study do not seem to match the current knowledge on DG-targeting MEC stellate cells, and 2) the degradation of the spatial phase code in DG is only indirectly assessed with classifiers that are not guaranteed to be optimal and that are hard to interpret. Addressing those concerns would greatly improve the clarity and impact of the paper (see below for suggestions).

Main comments

1) Line 115: I understand why the analysis biases phase R_{in} to low values. Nevertheless, I suspect a wider range can be explored and it would be valuable to evaluate pattern separation for larger R_{in} values more relevant to episodic memory theory. To increase input similarity, maybe keep most of a pair of trajectories the same and have the trajectories deviate only for small portions?

2) realism of MEC inputs:

Fig 1c-d: The theta-phase preference of EC inputs is not easy to see in those figures, it would help the reader to add a third inset in 1c to show a zoom on how the peak rate coincides with theta.

Line 153: Fig 2a shows a somewhat flat distribution, i.e. a weak theta modulation, with a small peak around the trough of a theta cycle. Why is it so broad when EC2 cells have been shown to have a strong modulation? Rev-Fig 3 provided in the rebuttal confirms a weaker theta-modulation than real MEC layer 2 neurons.

Line 159: This indeed resembles Misuzeki and Buzsaki 2009. But 1) I think the “early theta phase” and “late theta phase” terminology (brought back in Fig 5) is a bit misleading. It's a cyclical phenomenon so DG does not really precede EC, DG can just as well be described as following EC inputs from the preceding cycle (admittedly with a long delay, which is what puzzled Misuzeki and Buzsaki originally), 2) more importantly, there is more recent data on EC layer 2 phase modulation, and the recordings in Misuzeki2009 seemed to have been dominated by EC2 pyramidal cells, which target CA1, rather than DG-targeting ECII stellate cells that actually show a phase-preference at the peak of theta and not the trough like shown in Fig 2a (see Tang and Brecht 2014 + Rowland and Moser 2018). So, the apparent

paradox, highlighted again in the discussion (line 579), has actually already been resolved experimentally: DG-targeting EC cells do spike “just before” DG (see Valero and de la Prida 2019).

[Incidentally, those studies also show that stellate MEC cells targeting DG are in majority not Grid Cells. Even though Grid Cells are still a major input to DG, the other cell types can carry spatial info and show theta-modulation that would interact with the computation investigated.]

Also, the range of phase precession in individual stellate cells seems perhaps more constrained than in the model shown in Fig 1, as it spans only half of the cycle or less on average (see Ebbesen and Brecht 2016)

Overall, I think several limitations of the design of the EC inputs should at least be noted if not corrected. It would be good to put Rev-Fig3 figure in the manuscript, adding comparisons to papers more recent than Misuzeki2009 on EC theta-modulation and precession (see above), to show how the model fits or doesn't fit what is currently known of DG-targeting MEC neurons. Of course, since the broad phase distribution of EC inputs is so crucial to the phase-to-rate recoding mechanism (Fig 5), it would be best to check if results hold when constraining modeled MEC inputs to look more like stellate cells.

3) LEC inputs:

I appreciate that the authors included counter-cyclical EC inputs to model LEC as in Desmukh and Knierim 2010.

More details on these input patterns are needed in the methods: are they non-homogeneous poisson spiking with a phase modulation but no spatial info? Is the phase-modulation profile entirely reproduced from Desmukh et al 2010? If LEC inputs don't carry any spatial info in the model (which is not clear in vivo), it should be stated in the result section for clarity. Are LEC and MEC contacting the same cells?

What is the effect on Pattern Separation (with R_{in} defined over MEC and LEC input patterns and not just MEC)? Actually, using very or perfectly similar LEC inputs (that would model contextual similarity) while varying the trajectory, could help test pattern separation on a wider range of input similarities, for the phase code in particular (see comment 1).

Of course, if one considers that MEC stellate cells fire at the peak as mentioned above, the LEC phase modulation is not countercyclical anymore, and the main difference between MEC and LEC becomes the rate and phase space representations (and potentially differences in the strength of the theta modulation). Desmukh2010's data were not segregated by layers, but assuming that theta-modulation is homogenous across celltypes in LEC, would LEC inputs participate in the phase-to-rate recoding in that case? Or would they impair it because of a lack of precession?

Fig S5: I'm confused by the x-axis values that go up to 6π . Shouldn't the modulation be over 2π ? Shouldn't Fig S5aLeft be the same as in Fig2a?

Line 252: with the preceding LEC analysis, it should be explicit here that it's a *spatial* rate code that is measured. LEC phase-to-rate recoding could happen too in a different latent neural space.

4) DG computations on the spatial phase code + decoding analyses

To make sure it makes sense to average the phase values for 100ms bins to build phase vectors, please show the variance of spike phases per bin. The authors did show some variability in Fig 5 and Fig S7 but I did not see a clear quantification of that variance. If the variance is low, taking an average makes complete sense, there is little loss of information, but if the variance is high, the mean conveys little information about the actual phase of spikes in a bin. In that case, working on the instantaneous phase per spike would make more sense (cf Tingley and Buzsaki 2018). It's not necessarily clear what would be the threshold to deem the variance too high, but, in any case, showing a quantification of the variability would be helpful for the reader to assess the robustness of the results.

More importantly: In the first round, a couple reviewers were concerned about the choice of classifiers (perceptron and tempotron), as they only offer a lower bound and indirect assessment of phase information, and because they are hard to interpret (in part because they conflate the question of spatial information content and separability of trajectories). I don't think those issues have been clearly resolved, although the claims are now more careful. I think it would greatly improve the clarity and impact of the paper if the question "to which extent spatial phase coding is altered in the DG?" could be answered directly, by computing the spatial info in bits like the authors do for the rate code, and by decoding directly the position of the animal from the phase-code rather than classify the trajectories. The authors ask this very question on line 262-3. What about using the analysis tools developed by Tingley and Buzsaki (2018)? They adapted Olipher and Fenton 2003's spatial information metric to phase vectors, and use different decoding approaches (maximum correlations, bayesian,...) to directly decode position. Their whole approach is easy to interpret and quite relevant to the current study, as they show how CA1's rate+phase spatial code is transformed into a rate-independent spatial phase code in a downstream region.

5) CA3 model

Line 474: Panel D in Figure 6 is rather small, it's hard to see what happens when inhibition gets close to 0.

What are error bars? 95% CI?

My interpretation of the analysis is a bit different from the authors'. A difference between pyramidal rates remains when you decrease inhibition, but the effect size decreases nonetheless. So, the effect of DG inputs on CA3 ff inhib does partially drive CA3 recruitment. Which is not surprising: as expected from the model architecture, CA3 recruitment is mediated both by direct GC inputs and indirectly via DG-mediated feedforward inhib.

The point the authors want to make with this analysis is not clear to me.

Line 495-505: the introductory sentence is not super clear. Add "DG" before "synchrony"? I think this paragraph would be clearer if the authors said "CA3 firing rates" rather than "CA3 recruitment" and make clear the idea that DG synchrony, in addition to increasing CA3 rates, affects the timing of CA3 spiking, increasing the cross-correlations between CA3 neurons, which in turn obviously helps with STDP that needs pre and postsynaptic activity to happen close in time. This conclusion is very intuitive, but the authors could actually show the ISI and cross-correlation histograms of CA3 pyramidal neurons in different conditions (although it's hard to make those measures completely independent of rates).

Lines 435-47: It should be noted that no MF-CA3 short-term dynamics (which would be highly relevant to the effect of DG synchrony during short timescales like theta cycles) and no MF-CA3 long-term plasticity are modeled.

Similarly, the fact that "to isolate the plasticity effects of DG inputs, CA3 spiking is exclusively driven by GCs and not affected by recurrent CA3 inputs." (line 745-6) should be highlighted in the main text. This is motivated in the methods to help interpretation, but this design is very disconnected from reality and thus, I feel that it actually makes it harder to generalize to the real hippocampus where efficacious CA3 recurrent connections likely have non-trivial interactions with the rest of the system that could possibly introduce delays impacting STDP.

The general concern here is that, if you strip down CA3 of all the physiology that makes interesting computations and non-trivial interactions between different components of the network possible, the conclusion that synchronized inputs are better at recruiting disconnected LIF neurons and synchronizing them is not very surprising. I personally would rather see how a more realistic CA3 network behaves and just compare between rate recoded (i.e. synchronized) DG inputs vs altered DG inputs to infer how phase-to-rate recoding impacts CA3. But this position is debatable, so if the authors prefer to address those issues with textual edits to warn the reader, I think that's fair too.

6) Discussion

Line 602-8: As pointed out by the authors, feedback inhibition is a ubiquitous motif. If it consistently impairs phase info, as they suggest, how does phase info arise in some/most regions? When it arises, is it always in spite of the feedback inhibition? That seems a strong statement. Relatedly, are there examples of network where feedback inhibition is not so prominent with good phase information transmission?

I think the paper by Tingley and Buzsaki 2018 would fit well here in that discussion of other recoding mechanisms, as the phase code is preserved in the lateral septum, but not the rate-code. I'd be curious to know if the septum's local inhibitory circuit fits with some ideas developed in the present study.

Minor comments:

Line 117: The end of the sentence is not super clear. By "longer period", you mean longer than a theta cycle, correct?

Line 133: wrong letter. Should be referring to panel d, not c.

Fig 5c-d: It might be clearer to plot as a function of space rather than time here. The theta phase color code provides the relevant time info

Fig 5d: Arrows are not super clear, especially in the feedback inhib x shuffle condition. I see the authors' point, but maybe a bracket or circle would be more effective?

Line 434: For clarity, I think it should read "DG phase-to-rate recoding mediates improved at CA3 recurrent synapses", so that it is clear that the recoding corresponds to DG inputs and that it helps STDP at a non-DG class of synapses (i.e. heterosynaptically).

Line 436: should specify STDP at which synapse, and add a reference.

Point to point response to reviewer comments.

Reviewer #1 (Remarks to the Author):

The authors have fully and adequately addressed all my prior critique with extensive revisions including additional simulations, controls and figures. A couple of minor edits are suggested

1. There are several instances where it would help the reader and sentence structure to call out author name and year of references rather than the numbered reference. Please review the entire manuscript for this and revise as needed.

We have revised as advised.

2. Lines 220-221. The authors indicate that “spatial information content is almost identical”, which is conceptually accurate but at odds with the statistically significant difference. Rephrasing to indicate that the statistical difference is not meaningful would improve clarity.

Done (now line 235ff)

3. Line 571: “conjunction of thus functionally differentiated” should read “conjunction of these functionally differentiated”

Done. (now line 627)

Thank you for the helpful comments.

Reviewer #2 (Remarks to the Author):

The authors have engaged with the issues I raised in the first review and I am sympathetic to the idea that discrepancies between aspects of activity in the model and in vivo measurements are not sufficiently serious to preclude publication. There will inevitably be some discrepancies between simplified models and in-vivo data, and the as they point out in the revised manuscript, a strength of their proposed mechanism is that it does reproduce the apparently paradoxical in-vivo observation that the phase of peak firing in the DG precedes that in its EC input.

Nonetheless, I do think there remain real discrepancies between their modeling and in vivo data, and these should be discussed head-on. Specifically, from Mizuseki et al 2009 it appears that:

- The overall firing rate of the principal cell population in DG is not more strongly theta modulated than that in EC (Mizuseki Figure 3A). See also that the resultant vector for preferred phases in Mizuseki fig 3C is longer for EC than that for DG. This is not the case in the model.
- DG principal cells show theta phase precession that appear to be at least as strong if not stronger in DG as that in EC2 (Mizuseki Figure 7). Though phase precession in DG is not explicitly analysed in the paper, this in vivo finding does not appear consistent with the narrow theta-phase range of DG spiking in the model.

Thank you for these comments. We have added several additional analyses to further clarify the relation of our model to available *in vivo* data (also to address several related issues raised by Reviewer 3: novel supplementary figures 3,7,8,11-13). These suggest that the model overall well

captures the *in vivo* situation. However, it is of course true that some discrepancies between a simplified model and *in vivo* data are inevitable, and we now more explicitly state and discuss these).

Specifically, we note that (line 672ff)

- i) “*in vivo*, EC cells are more strongly theta-modulated than DG cells” and that
- ii) “phase precession appears at least as pronounced in DG as in EC (a finding potentially at odds with the presently suggested degradation of phase information in DG).”

Additional analysis of literature revealed potential reasons for these discrepancies (discussed in lines 675ff).

Concerning i), recent research suggests that many EC cells recorded in Mizuseki et al.¹ were likely CA1-projecting pyramidal cells which do not project to DG^{2,3}. Incidentally, these cells are more strongly theta-modulated than DG projecting stellate cells² and would lead to an overestimation of mean EC modulation. A lower modulation of GCs *in vivo* may derive from non-modelled additional inputs (line 681).

Concerning ii), it remains unclear to which degree the DG cells recorded in Mizuseki et al.¹ reflected mossy cells, rather than GCs, since naïve extracellular recordings are known to be biased towards the more frequently firing mossy cells^{4,5} (Mizuseki included only cells with at least 50 spikes, which systematically biases analysis against sparsely firing GCs). Notably, mossy cells are less strongly theta-modulated than GCs⁵, and so might lead to an underestimate of phase-range (though phase precession strength was not analysed in Senzai and Buzsaki 2017⁵). Finally, we would like to note that R^2 , the quantity used by Mizuseki et al. to measure phase precession strength, should theoretically be independent of phase-range (the measure normalizes to spread in both x and y). Overall, literature data seems consistent with a comparatively narrow theta-distribution for GCs, (e.g. Skaggs 1996⁶: A ‘lack of activity at the end of the theta-cycle was characteristic of GCs’), including plots within Mizuseki et al (the previous Rev. Fig. 3, which is now shown as Supplementary fig 3c).

We thus believe these discrepancies can be plausibly explained and do not undermine our findings.

Reviewer #3 (Remarks to the Author):

The manuscript is much improved and even more pleasant to read than the original submission. The authors addressed a large number of concerns, adding new analyses to successfully show the robustness of their conclusions against different sources of noise, to better understand the relationship with DG sparsity and to demonstrate the mechanism of phase-to-rate recoding in DG. The claims of the paper are overall well-supported. However, I have two major remaining concerns: 1) the EC inputs used in the study do not seem to match the current knowledge on DG-targeting MEC stellate cells, and 2) the degradation of the spatial phase code in DG is only indirectly assessed with classifiers that are not guaranteed to be optimal and that are hard to interpret. Addressing those concerns would greatly improve the clarity and impact of the paper (see below for suggestions).

Thank you for these valuable comments. We have made substantial further revisions and added additional analysis, to address the numerous valuable comments (we added 6 novel supplementary figures and integrated new analyses into 3 supplements, see below for details).

Most pertinently:

On 1) We have carefully reviewed the respective literature and show additional analysis (novel Suppl. Fig. 2), finding that apparent differences are largely due to differences in analysis routines (see below). Where differences persist, we more clearly alert the reader and discuss them.

On 2) We have added several additional analyses, most pertinently of positional phase information following Tingley and Buzsaki 2018, directly showing that phase information is partially, but not fully, degraded in DG (novel Suppl. Fig. 8).

Main comments

1) Line 115: I understand why the analysis biases phase R_{in} to low values. Nevertheless, I suspect a wider range can be explored and it would be valuable to evaluate pattern separation for larger R_{in} values more relevant to episodic memory theory. To increase input similarity, maybe keep most of a pair of trajectories the same and have the trajectories deviate only for small portions?

Our first impulse was exactly the same. Specifically, we measured similarity for *fully identical* trajectories (distance = 0, but with different Poisson seeds). However, we found that this led to correlations indistinguishable from the 0.5cm case (this is now shown in Supplementary Fig. 7e). In this sense, we capture the maximal behaviorally plausible similarity, given the constraint of a Poisson process. While it would of course be possible to manipulate the Poisson process further to obtain higher similarities, careful consideration suggested that the biological motivation to do so is questionable. The reason is the already mentioned methodological point about analysis-time window, which we now try to make clearer (lines 117ff):

“Note that the presently reported maximal input correlation values are misleadingly low due to a necessary methodological idiosyncrasy, namely the need to measure correlations based on 100 ms (θ) time-windows. Such short time-windows are known to reduce *measured* rate-correlation values several-fold^{22,32}. Indeed, identical spike trains (those of our similar trajectories) lead to correlations of $R_{rate}=0.4$ when assessed with 100 ms time-windows and $R_{rate}=0.85$ when assessed with 2s time-windows. In other words, the actual underlying data covers a similar range as previous studies^{22,32} when compared appropriately, i.e. based on similar time-windows (note that, to the best of our knowledge there are no previous studies assessing phase code correlation). Indeed, ‘behaviorally’ identical trajectories (distance = 0 cm) led to input correlations indistinguishable from the shown maximal input correlation (distance = 0.5cm, Supplementary Fig. 7), suggesting we cover the behaviorally plausible range (given the constraint of a Poisson process). “

Note that we also increased similarity slightly by adding an identical LEC input (novel Supplementary Fig. 7e-f, see below).

2) realism of MEC inputs:

Fig 1c-d: The θ -phase preference of EC inputs is not easy to see in those figures, it would help the reader to add a third inset in 1c to show a zoom on how the peak rate coincides with θ .

We have tried to make this clearer and added another inset into Fig1c.

Line 153: Fig 2a shows a somewhat flat distribution, i.e. a weak θ modulation, with a small peak around the trough of a θ cycle. Why is it so broad when EC2 cells have been shown to

have a strong modulation? Rev-Fig 3 provided in the rebuttal confirms a weaker theta-modulation than real MEC layer 2 neurons.

In our view, the literature on identified stellate cells in EC2 is consistent with a weak overall modulation (e.g. 0.2 in Ray et al., 2014²).

The literature cited below by the Reviewer suggests that the Mizuseki data likely overestimate the modulation of grid-cell inputs to DG: As shown in Rowland et al 2018³, Fig4C or Ray et al., 2014², Fig4G, the strongly modulated cells constituting the peak in EC2 were likely EC2 pyramidal cells, since these are i) more strongly theta-modulated² and ii) match the theta-timing of the peak in Mizuseki et al.³. Yet EC2 pyramids do not project to DG^{2,9}. Additionally, Rowland et al 2018³, find that non-grid cells in EC are more strongly theta-modulated than grid cells.

We have integrated Rev-Fig 3 into Supplementary Fig. 3, where we now note these caveats.

Line 159: This indeed resembles Mizuseki and Buzsaki 2009. But 1) I think the “early theta phase” and “late theta phase” terminology (brought back in Fig 5) is a bit misleading. It's a cyclical phenomenon so DG does not really precede EC, DG can just as well be described as following EC inputs from the preceding cycle (admittedly with a long delay, which is what puzzled Mizuseki and Buzsaki originally), 2) more importantly, there is more recent data on EC layer 2 phase modulation, and the recordings in Mizuseki2009 seemed to have been dominated by EC2 pyramidal cells, which target CA1, rather than DG-targeting ECII stellate cells that actually show a phase-preference at the peak of theta and not the trough like shown in Fig 2a (see Tang and Brecht 2014 + Rowland and Moser 2018). So, the apparent paradox, highlighted again in the discussion (line 579), has actually already been resolved experimentally: DG-targeting EC cells do spike “just before” DG (see Valero and de la Prida 2019).

Thank you for this comment. We have added a clarification (line172), where we state “Note that the ‘early’ versus ‘late’ theta terminology used throughout this paper is a narrative convenience - since theta is a cyclical phenomenon ‘late’ theta grid cells can also drive ‘early’ theta GCs, though in the present case this would imply implausibly long synaptic delays (>50ms, compare Mizuseki et al., 2009)¹.”

We also now alert the reader to the likely confound of EC pyramidal cells in the discussion, where we added (662ff): “It should be noted that differences in EC cell types and projections provide an additional, though not mutually exclusive, explanation^{2,3,9} (Supplementary Fig. 3c).”

[Incidentally, those studies also show that stellate MEC cells targeting DG are in majority not Grid Cells. Even though Grid Cells are still a major input to DG, the other cell types can carry spatial info and show theta-modulation that would interact with the computation investigated.]

We now more explicitly alert the reader to this and motivate why we nevertheless focus on grid cell inputs, stating that (line 664):

“Nevertheless, it must be emphasized that our model reflects a dramatic simplification compared to the *in vivo* circuit. For instance, *in vivo* GCs receive numerous other inputs which may or may not themselves contain time codes. Indeed, while there are some discrepancies in the literature¹⁰⁻¹², recent research suggests that perhaps only 25% of DG-projecting EC stellate cells are grid cells³. We presently chose to nevertheless focus our analysis to these inputs, because only they allowed to specify a well-defined time code (addressing the role of additional inputs in various robustness analyses).”

Also, the range of phase precession in individual stellate cells seems perhaps more constrained than in the model shown in Fig 1, as it spans only half of the cycle or less on average (see Ebbesen and Brecht 2016)

Thank you for this extremely valuable comment. At first sight this indeed seems like a discrepancy, but at closer inspection, it revealed itself as a methodological artifact:

We quantified the phase-range of our grid-cells analogous to Ebbesen and Brecht, 2016¹³ or Reifenstein et al. 2016¹⁴, who both use the same method. Specifically, precession *within single trials* was assessed by fitting a regression line, and phase range was estimated as the y-range of the resulting regression. We obtain an average phase range of $128 \pm 65^\circ$ (mean \pm std), similar to the ranges reported in these studies 169° in Reifenstein et al. (2016) and 127.5° in Ebbesen et al. (2016). We have added the novel Supplementary figure 2, to allow direct comparison to these studies and reveal the origin of the seeming discrepancy:

1. A regression fit to a single trial^{13,14} will lead to systematically lower estimates than suggested by cumulated data¹⁵, because the cumulated data will contain more samples on the far x-range. The reason is that smaller x-ranges (of subsamples) directly translate to smaller y-ranges for any non-null slope.
2. The presence of noise spikes or non-perfect field-traversals will tend to lead to an underestimation of the regression slope, and by implication phase-range. By 'non-perfect field traversal' we mean e.g. a case where the mouse traverses a field at that field's periphery. For instance, Reifenstein et al. (2016) show an example where the phase slope is close to zero and actually slightly positive (in their Fig. 1B, a screenshot is pasted below). This could be due to noise-spikes, or perhaps the chaining of two adjacent fields which are cut at strange angles. We suspect such cases were excluded, since including e.g. a negative phase range would dramatically depress the estimated median (though we could not find any indication that such cases were excluded in either Reifenstein et al. or Ebbesen et al., or if they were, by which criterion). Note that our estimate of $128 \pm 65^\circ$ is based only on cases where the center of the grid field is traversed but did not contain an exclusion criterion for noise-spikes. The stated procedure in e.g. Ebbesen et al. was simply to detect periods of elevated activity separated by $>250\text{ms}$ inactivity, which would include the cases shown in Fig S2b.

Novel Supplementary figure 2. Single trial phase precession. To allow comparison to identified EC stellate cells^{13,14}, we computed single trial phase precession for individual grid cells of our model. **a** 40 random examples of phase precession plots of individual grid cells for a specific trajectory. For each cell, data are averaged over 20 Poisson seeds. Note, that even though every cell is defined as a perfectly phase precessing grid cell, many instances don't reflect the typical phase precession pattern (dashed frames). The reason is that for any physiologically plausible population of grid cells with diverse grid spacings, orientations and phase-offsets¹⁶, there will be numerous cases where the center of a grid field is not traversed (compare Reifenstein et al., 2016¹⁴ Fig. 1B). **b** Ten examples of linear fits to estimate phase range based on single trials (Poisson seeds), analogous to Ebbesen et al. (2016)¹³ or Reifenstein et al. (2016)¹⁴. Fits were made to single field traversals (20 cm corresponds to 1s) **c** Resulting estimates of single trial phase range for our model and the literature (as data on the distribution was unavailable we plot only the reported medians). Note that we included only cases where the center of the grid field is traversed, excluding cases that would lead to an artifactual depression of the estimated phase range (dashed frames in **a**).

Fig. 1B, copied from Reifenstein 2016.

Overall, I think several limitations of the design of the EC inputs should at least be noted if not corrected. It would be good to put Rev-Fig3 figure in the manuscript, adding comparisons to papers more recent than Misuzeki2009 on EC theta-modulation and precession (see above), to show how the model fits or doesn't fit what is currently known of DG-targeting MEC neurons. Of course, since the broad phase distribution of EC inputs is so crucial to the phase-to-rate recoding mechanism (Fig 5), it would be best to check if results hold when constraining modeled MEC inputs to look more like stellate cells.

Fortunately, the closer comparison of our data with the more recent papers revealed that our modeled grid cells do closely resemble those described by Ebbesen et al. and Reifenstein et al.. As partially quoted above, we have made several modifications to the manuscript to make this clearer, show the data, and discuss the more recent literature and remaining discrepancies (e.g. line 88, 651ff, 671ff).

3) LEC inputs:

I appreciate that the authors included counter-cyclical EC inputs to model LEC as in Desmukh and Knierim 2010.

More details on these input patterns are needed in the methods: are they non-homogeneous poisson spiking with a phase modulation but no spatial info? Is the phase-modulation profile entirely reproduced from Desmukh et al 2010? If LEC inputs don't carry any spatial info in the model (which is not clear in vivo), it should be stated in the result section for clarity. Are LEC and MEC contacting the same cells?

That is exactly correct. We simulated non-homogeneous Poisson trains with phase modulation but no spatial info. The modulation profile was approximately matched to Desmukh by using a smoothed saw-tooth profile. We have made clearer in the results that these LEC inputs carry no spatial information (line 266), and have added an LEC input that contains spatial information (see below). We have also added a dedicated methods section detailing how LEC and noise analyses were implemented (lines 772ff).

What is the effect on Pattern Separation (with R_{in} defined over MEC and LEC input patterns and not just MEC)? Actually, using very or perfectly similar LEC inputs (that would model contextual similarity) while varying the trajectory, could help test pattern separation on a wider range of input similarities, for the phase code in particular (see comment 1).

We have now added an 'identical LEC input' in which the exact same set of LEC spike-trains coincides with varying MEC inputs onto GCs, just as the reviewer suggested (novel

Supplementary Fig. 7). This models high contextual similarity which is encoded in LEC phases as well as rates. As expected, this further increases input correlations, but the pattern separation effect remains robust (novel Supplementary Fig. 7e). The phase-to-rate recoding effect also remained robust (novel Supplementary Fig. 7c-d).

Of course, if one considers that MEC stellate cells fire at the peak as mentioned above, the LEC phase modulation is not countercyclical anymore, and the main difference between MEC and LEC becomes the rate and phase space representations (and potentially differences in the strength of the theta modulation). Desmukh2010's data were not segregated by layers, but assuming that theta-modulation is homogenous across celltypes in LEC, would LEC inputs participate in the phase-to-rate recoding in that case? Or would they impair it because of a lack of precession?

While interesting, we feel an analysis of this question would likely remain quite speculative since it would require additional assumptions. We would like to point to the 'noise robustness analysis' which contains no theta-modulation, and thus partially addresses this point (suggesting the fundamental findings are robust to *coincident* input from other sources).

Fig S5: I'm confused by the x-axis values that go up to 6π . Shouldn't the modulation be over 2π ? Shouldn't Fig S5aLeft be the same as in Fig2a?

Thank you for this comment, the values were not in π but in radians. We have corrected the axis labels.

Line 252: with the preceding LEC analysis, it should be explicit here that it's a **spatial** rate code that is measured. LEC phase-to-rate recoding could happen too in a different latent neural space.

We have amended this (now line 272) "These results suggest that spatial *phase-codes* in EC are in part converted to spatial *rate-codes* in DG".

4) DG computations on the spatial phase code + decoding analyses

To make sure it makes sense to average the phase values for 100ms bins to build phase vectors, please show the variance of spike phases per bin. The authors did show some variability in Fig 5 and Fig S7 but I did not see a clear quantification of that variance. If the variance is low, taking an average makes complete sense, there is little loss of information, but if the variance is high, the mean conveys little information about the actual phase of spikes in a bin. In that case, working on the instantaneous phase per spike would make more sense (cf Tingley and Buzsaki 2018). It's not necessarily clear what would be the threshold to deem the variance too high, but, in any case, showing a quantification of the variability would be helpful for the reader to assess the robustness of the results.

We computed the bin-wise phase variance and now plot their distribution in Supplementary Fig1f (see below). Mean phase variance within a theta-cycle was $21 \pm 10^\circ$ for EC grid and $2 \pm 3^\circ$ (mean \pm std) for DG-granule cells. In our view, given these low variances, the mean is a useful approximation.

Supplementary Fig. 1f. Phase variance of spikes within individual theta cycles. Each variance assessed for all spikes of 20 Poisson seeds within a theta-bin. It was calculated for all bins of 40 random cells.

More importantly: In the first round, a couple reviewers were concerned about the choice of classifiers (perceptron and tempotron), as they only offer a lower bound and indirect assessment of phase information, and because they are hard to interpret (in part because they conflate the question of spatial information content and separability of trajectories). I don't think those issues have been clearly resolved, although the claims are now more careful. I think it would greatly improve the clarity and impact of the paper if the question "to which extent spatial phase coding is altered in the DG?" could be answered directly, by computing the spatial info in bits like the authors do for the rate code, and by decoding directly the position of the animal from the phase-code rather than classify the trajectories. The authors ask this very question on line 262-3. What about using the analysis tools developed by Tingley and Buzsaki (2018)? They adapted Olipher and Fenton 2003's spatial information metric to phase vectors, and use different decoding approaches (maximum correlations, bayesian,...) to directly decode position. Their whole approach is easy to interpret and quite relevant to the current study, as they show how CA1's rate+phase spatial code is transformed into a rate-independent spatial phase code in a downstream region.

Thank you for this excellent suggestion. We have now adapted the 'positional phase information' measure developed by Tingley and Buzsaki (2018) and use it to explicitly address the question to which degree phase info in the DG is degraded (novel Supplementary Fig. 8, Lines 354ff). This analysis confirms that phase information is partially, but not fully, degraded in DG due to feedback inhibition.

5) CA3 model

Line 474: Panel D in Figure 6 is rather small, it's hard to see what happens when inhibition gets close to 0.

What are error bars? 95% CI?

My interpretation of the analysis is a bit different from the authors'. A difference between pyramidal rates remains when you decrease inhibition, but the effect size decreases nonetheless. So, the effect of DG inputs on CA3 ff inhib does partially drive CA3 recruitment. Which is not surprising: as expected from the model architecture, CA3 recruitment is mediated both by direct GC inputs and indirectly via DG-mediated feedforward inhib. The point the authors want to make with this analysis is not clear to me.

The error bars represent sd (we have added this to the legend). We agree, that this behavior is expected, and there is no particular point we want to make with the panel d analysis per se. However, the panel does answer the question if the difference in pyramidal cell recruitment is driven merely by changes in interneuron recruitment, which we thought was worth addressing. Furthermore, as seen in panels g-l, the plasticity enhancing effect of phase-to-rate recoding (our main variable of interest) is sensitive to the strength of feedforward inhibition. Showing a plausible range of inhibition and its effect on firing rates thus seems indicated.

Line 495-505: the introductory sentence is not super clear. Add "DG" before "synchrony"? I think this paragraph would be clearer if the authors said "CA3 firing rates" rather than "CA3 recruitment" and make clear the idea that DG synchrony, in addition to increasing CA3 rates, affects the timing of CA3 spiking, increasing the cross-correlations between CA3 neurons, which in turn obviously helps with STDP that needs pre and postsynaptic activity to happen close in time. This conclusion is very intuitive, but the authors could actually show the ISI and cross-correlation histograms of CA3 pyramidal neurons in different conditions (although it's hard to make those measures completely independent of rates).

Thank you for these suggestions. We have added 'DG' before 'synchrony' and have changed 'CA3 recruitment' to 'CA3 pyramidal firing rates' (now line 514ff).

Lines 435-47: It should be noted that no MF-CA3 short-term dynamics (which would be highly relevant to the effect of DG synchrony during short timescales like theta cycles) and no MF-CA3 long-term plasticity are modeled.

Similarly, the fact that "to isolate the plasticity effects of DG inputs, CA3 spiking is exclusively driven by GCs and not affected by recurrent CA3 inputs." (line 745-6) should be highlighted in the main text. This is motivated in the methods to help interpretation, but this design is very disconnected from reality and thus, I feel that it actually makes it harder to generalize to the real hippocampus where efficacious CA3 recurrent connections likely have non-trivial interactions with the rest of the system that could possibly introduce delays impacting STDP.

The general concern here is that, if you strip down CA3 of all the physiology that makes interesting computations and non-trivial interactions between different components of the network possible, the conclusion that synchronized inputs are better at recruiting disconnected LIF neurons and synchronizing them is not very surprising. I personally would rather see how a more realistic CA3 network behaves and just compare between rate recoded (i.e. synchronized) DG inputs vs altered DG inputs to infer how phase-to-rate recoding impacts CA3. But this position is debatable, so if the authors prefer to address those issues with textual edits to warn the reader, I think that's fair too.

The choice between simplicity/interpretability and complexity/realism is of course always difficult when modeling, and there is arguably no ideal trade-off. In other words, we perfectly understand the reviewers concern. We have thus now added an additional 'extended CA3 model' (lines 551 to 563, Supplementary Fig. 13) in which we have incorporated several of the key features mentioned above. Specifically, we added MF facilitation and the recurrent CA3 input, which in turn necessitated the addition of an additional homeostatic mechanism to prevent runaway excitation (synaptic scaling). In this extended model, the key findings remained robust.

Of course, more realism could always be added, but we feel that this would necessitate extensive additional robustness and validity analyses, which would in our view exceed a reasonable scope for the present paper or question. We have amended the text noting these limitations (lines 666ff, 684ff)

Ultimately, the observation that increased DG synchrony leads to improved CA3 cell recruitment and STDP remains the main point of relevance for the present paper. It i) illustrates a likely

function of phase to rate recoding in the hippocampus which ii) is likely to generalize to potential other instances of the phenomenon.

6) Discussion

Line 602-8: As pointed out by the authors, feedback inhibition is a ubiquitous motif. If it consistently impairs phase info, as they suggest, how does phase info arise in some/most regions? When it arises, is it always in spite of the feedback inhibition? That seems a strong statement. Relatedly, are there examples of network where feedback inhibition is not so prominent with good phase information transmission?

I think the paper by Tingley and Buzsaki 2018 would fit well here in that discussion of other recoding mechanisms, as the phase code is preserved in the lateral septum, but not the rate-code. I'd be curious to know if the septum's local inhibitory circuit fits with some ideas developed in the present study.

Thank you for this comment. We now discuss this question in more detail in lines 629-648. First, it indeed appears that there is no documented local inhibitory circuit in lateral septum¹⁷ (LS), though it should be noted, that subsequent research does not substantiate Tingley and Buzsaki's finding of the absence of spatial rate coding in LS¹⁸. Second, we clarify that *we do not take our results to imply the necessary degradation of time coding through feedback inhibition*. Instead, the degradation of phase information will depend on a number of factors such as 'the precise spatial and temporal properties of inhibition, the overall strength of inhibition, and the type and oscillatory structure of incoming temporal codes.' (line 647)

We also clarify that "the presence of phase-to-rate recoding is ultimately logically independent of the degradation of the original phase code - it requires only that upstream phase-codes improve downstream rate-codes." (line 283ff)

Minor comments:

Line 117: The end of the sentence is not super clear. By "longer period", you mean longer than a theta cycle, correct?

Correct. More specifically, we meant 'more typical periods to measure rate correlations'. These can range from seconds⁷ to minutes¹⁹. We now state the specific measurement time-windows and respective correlation values obtained for identical spike-trains, namely our highly similar spike-trains (line 121ff).

Line 133: wrong letter. Should be referring to panel d, not c. **Corrected**

Fig 5c-d: It might be clearer to plot as a function of space rather than time here. The theta phase color code provides the relevant time info

Changed axis label to 'Location (cm)'

Fig 5d: Arrows are not super clear, especially in the feedback inhib x shuffle condition. I see the authors' point, but maybe a bracket or circle would be more effective?

Changed it to ovals.

Line 434: For clarity, I think it should read “DG phase-to-rate recoding mediates improved at CA3 recurrent synapses”, so that it is clear that the recoding corresponds to DG inputs and that it helps STDP at a non-DG class of synapses (i.e. heterosynaptically).

Changed it. (now line 479)

Line 436: should specify STDP at which synapse, and add a reference.

Done

Thank you very much for the extensive, detailed, and highly insightful reviews. We feel they have substantially improved the manuscript in terms of both content and presentation.

REVIEWERS' COMMENTS

Reviewer #2 (Remarks to the Author):

The authors have addressed the points I raised in the last review and I support publication of the manuscript.

Reviewer #3 (Remarks to the Author):

The authors have addressed all my concerns. I am grateful for all their hard work. This is a very good paper!

REVIEWERS' COMMENTS

Reviewer #2 (Remarks to the Author):

The authors have addressed the points I raised in the last review and I support publication of the manuscript.

Reviewer #3 (Remarks to the Author):

The authors have addressed all my concerns. I am grateful for all their hard work. This is a very good paper!

Author response:

Thank you very much.